# Improved Variational Inference in Discrete VAEs using Error Correcting Codes

## Abstract

Despite significant advancements in deep probabilistic models, effective learning of low-dimensional discrete latent representations remains challenging. This paper introduces a novel method to improve variational inference in discrete latent variable models by employing Error-Correcting Codes (ECCs) to add redundancy to the latent representations, later exploited by the variational approximated posterior to provide more accurate estimates, thereby reducing the variational gap. Drawing inspiration from ECCs used in digital communications and data storage, we demonstrate proof-of-concept using a Discrete Variational Autoencoder (DVAE) with binary latent variables and block repetition codes. We then extend it to a hierarchical structure inspired by polar codes, in which some latent bits are more robustly protected than others. Our approach significantly enhances generation quality, data reconstruction, and uncertainty calibration compared to the uncoded DVAE, even when trained with tighter bounds such as the Importance Weighted Autoencoder (IWAE) objective. In particular, we demonstrate superior performance on MNIST, FMNIST, CIFAR10, and Tiny ImageNet datasets. The general approach of integrating ECCs into variational inference is compatible with existing techniques to boost variational inference, such as importance sampling or Hamiltonian Monte Carlo. We also formulate the properties that ECCs need to possess to be effectively used for improved discrete variational inference.

## 1 Introduction

Discrete latent space models seek to represent data using a finite set of features. Recent progress in generative models has increasingly favored these representations, as they are well-suited for datasets characterized by naturally discrete hidden states. However, effective learning of low-dimensional discrete latent representations is technically challenging. Vector Quantized-Variational Autoencoders (VQ-VAEs) (Van Den Oord et al., 2017; Razavi et al., 2019) stand out as solutions for this problem but rely on a non-probabilistic autoencoder, which does not provide uncertainty quantification in the latent space (as further discussed in Appendix L). To fit a fully probabilistic Variational Autoencoder (VAE) model (Kingma & Welling, 2013), a common approach considers either Concrete (Maddison et al., 2017) or Gumble-Softmax (Jang et al., 2017) approximations to sample from a discrete latent distribution in a reparameterizable manner (Ramesh et al., 2021; Lievin et al., 2020). However, this approach leads to instabilities since the gradient variance is sensitive to the temperature that controls these approximations. The DVAE in Rolfe (2016); Vahdat et al. (2018b;a), augments the binary latent representations with a set of continuous random variables, pairing each bit with a continuous counterpart where reparameterization can be done in a more stable manner after marginalizing the latent bits. The key distinction between the DVAE (Rolfe, 2016) and the DVAE++ (Vahdat et al., 2018b) lies in their smoothing transformations: while Rolfe (2016) introduces spike-and-exponential transformations, Vahdat et al. (2018b) uses overlapping exponential distributions. These overlapping transformations are generalized in Vahdat et al. (2018a), enabling tighter variational bounds. We demonstrate the effectiveness of our method over a simplified version of the DVAE++ (Vahdat et al., 2018b).

This work presents a novel method to improve variational inference and representation learning in generative models with discrete latent variables. In particular, for a latent variable model, we argue that one should use ECCs to introduce redundancy into the latent sample before the reconstruction decoder network processes it to generate the data. The variational approximation to the true posterior

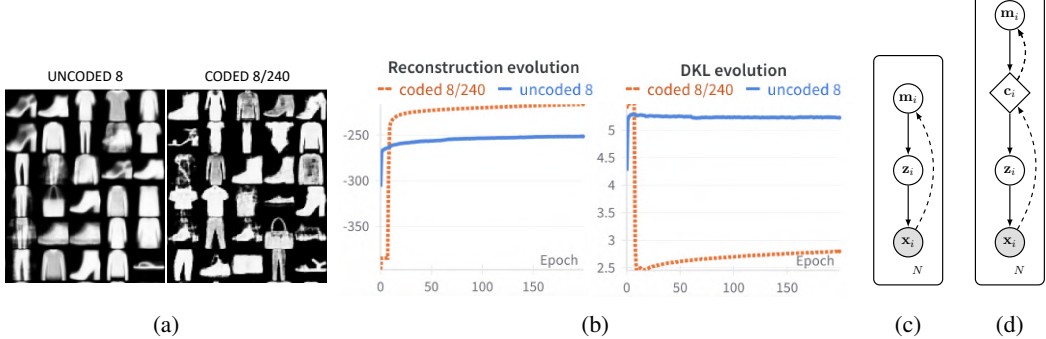

Figure 1: **Comparison between uncoded and coded DVAE models** with 8 latent bits, where the uncoded models are identified by the number of latent bits, and the coded models by their code rate. Fig. (a) presents uncurated generation examples for FMNIST. Fig. (b) illustrates the evolution of the reconstruction and regularization terms of the ELBO loss. Fig. (c) depicts the graphical model of the uncoded DVAE, while Fig. (d) shows the graphical model of the coded DVAE.

distribution can then exploit the added redundancy to provide more accurate estimates, reducing, in turn, the variational gap to the data likelihood.

Our approach is based on well-known digital communications and data storage techniques where information is protected with ECCs before transmission/storage to reduce the overall error rate during recovery. For different datasets, our results demonstrate that, compared to the uncoded DVAE, the DVAE with ECCs (Coded-DVAE) achieves superior generation quality, better data reconstruction, and critically calibrated uncertainty in the latent space. In Fig. 1, we highlight some representative results for both MNIST and FMNIST data sets. We note that the use of ECCs is a general design approach that is perfectly compatible with state-of-the-art techniques for improved variational inference, such as importance sampling (Burda et al., 2016; Thin et al., 2021) or Hamiltonian Monte Carlo (Wolf et al., 2016; Caterini et al., 2018). In summary, our main contributions are:

- We provide proof-of-concept results demonstrating that training deep generative models can be improved by ECC techniques, an idea that, to the best of our knowledge, is completely novel in the literature.

- We formulate a coded version of DVAE using block repetition codes. We show that encoding/decoding of the block repetition code can be efficiently done with linear complexity.

- We show that Coded-DVAE improves reconstruction, generation, and uncertainty calibration in the latent space when compared to the uncoded case using the same latent dimension, even when the uncoded DVAE is trained with tighter bounds such as the IWAE objective (Burda et al., 2016).

- We discuss the generalization of this method to other coding schemes and introduce a hierarchical structure, inspired by polar codes (Arikan, 2009), that effectively separates high-level information from finer details.

- Through an extensive ablation study, we show that the enhancement in performance is not attributed to the increased dimensionality introduced by the redundancy from the ECC.

## 2 OUR BASELINE: THE UNCODED DVAE

This section introduces a simplified version of the *uncoded* DVAE (Rolfe, 2016; Vahdat et al., 2018b;a), serving as the foundational model upon which the subsequent aspects of our work are constructed. Let $X = \{x_0, \ldots, x_N\}$ denote a collection of unlabelled data, where $x_i$ represents a K-dimensional feature vector. While Rolfe (2016), Vahdat et al. (2018b) and Vahdat et al. (2018a) use Boltzmann machine priors, we consider a generative probabilistic model characterized by a simple low-dimensional binary latent variable $m \in \{0,1\}^M$ comprising independent and identically

distributed (i.i.d.) Bernoulli components $p(\boldsymbol{m}) = \prod_{j=1}^{M} p(m_j) = \prod_{j=1}^{M} \text{Ber}(\nu)$. Since backpropagation through discrete variables is generally not possible, a smoothing transformation of these binary variables is introduced. While the smoothing transformations proposed in Rolfe (2016) are limited to spike-and-X type of transformations, Vahdat et al. (2018b) show better results by using truncated exponential distributions:

$$p(\boldsymbol{z}|\boldsymbol{m}) = \prod_{j=1}^{M} p(z_j|m_j), \quad p(z_j|m_j) = \frac{e^{-\beta(z_j - m_j)}}{Z_\beta}, \tag{1}$$

for $m_j \in \{0, 1\}$, $z_j \in [0, 1]$, and $Z_\beta = (1 - e^{-\beta})/\beta$. The parameter $\beta$ serves as an inverse temperature term, similar to the one in the Gumbel-Softmax relaxation (Jang et al., 2017). Given the simplicity of the defined binary prior, the complexity of the model is primarily determined by the likelihood function $p_{\boldsymbol{\theta}}(\boldsymbol{x}|\boldsymbol{z}) = p(f_{\boldsymbol{\theta}}(\boldsymbol{z}))$, where the likelihood is a Neural Network (NN) (referred to as the decoder) with parameter set $\boldsymbol{\theta}$.

**Variational family and inference**

Following Rolfe (2016), we assume an amortized variational family of the following form:

$$q_{\boldsymbol{\eta}}(\boldsymbol{m}, \boldsymbol{z}|\boldsymbol{x}) = q_{\boldsymbol{\eta}}(\boldsymbol{m}|\boldsymbol{x})p(\boldsymbol{z}|\boldsymbol{m}), \quad q_{\boldsymbol{\eta}}(\boldsymbol{m}|\boldsymbol{x}) = \prod_{j=1}^{M} \text{Ber}(g_{j,\boldsymbol{\eta}}(\boldsymbol{x})), \tag{2}$$

where $g_{\boldsymbol{\eta}}(\boldsymbol{x})$ represents a parameterized function; here, a NN (referred to as the encoder) with parameter set $\boldsymbol{\eta}$. Inference is achieved by maximizing the Evidence Lower Bound (ELBO), which can be expressed as

$$\log p(\boldsymbol{x}) \geq \int q_{\boldsymbol{\eta}}(\boldsymbol{m}, \boldsymbol{z}|\boldsymbol{x}) \log \left( \frac{p_{\boldsymbol{\theta}}(\boldsymbol{x}, \boldsymbol{z}, \boldsymbol{m})}{q_{\boldsymbol{\eta}}(\boldsymbol{m}, \boldsymbol{z}|\boldsymbol{x})} \right) d\boldsymbol{m} d\boldsymbol{z} = \mathbb{E}_{q_{\eta}(\boldsymbol{m}, \boldsymbol{z}|\boldsymbol{x})} \log \left( \frac{p_{\boldsymbol{\theta}}(\boldsymbol{x}|\boldsymbol{z})p(\boldsymbol{z}|\boldsymbol{m})p(\boldsymbol{m})}{q_{\boldsymbol{\eta}}(\boldsymbol{m}|\boldsymbol{x})p(\boldsymbol{z}|\boldsymbol{m})} \right)$$

$$= \mathbb{E}_{q_{\eta}(\boldsymbol{m}, \boldsymbol{z}|\boldsymbol{x})} \log p_{\boldsymbol{\theta}}(\boldsymbol{x}|\boldsymbol{z}) - \mathcal{D}_{KL}\big(q_{\boldsymbol{\eta}}(\boldsymbol{m}|\boldsymbol{x}) || p(\boldsymbol{m})\big), \tag{3}$$

where the first term corresponds to the reconstruction of the observed data and the second term is the Kullback-Leibler (KL) Divergence between the variational family and the binary prior distribution, which acts as a regularization term. This can be computed in closed form as $\mathcal{D}_{KL}\big(q_{\boldsymbol{\eta}}(\boldsymbol{m}|\boldsymbol{x}) || p(\boldsymbol{m})\big) = \sum_{j=1}^{M} \left[ q_j \log \frac{q_j}{\nu} + (1 - q_j) \log \frac{1-q_j}{1-\nu} \right]$, where $q_j = q_{\boldsymbol{\eta}}(m_j = 1|\boldsymbol{x})$. The reconstruction term needs to be approximated via Monte Carlo. Since $p_{\boldsymbol{\theta}}(\boldsymbol{x}|\boldsymbol{z})$ does not depend on the binary latent variable $\boldsymbol{m}$, we can marginalize the posterior distribution as

$$q_{\boldsymbol{\eta}}(\boldsymbol{z}|\boldsymbol{x}) = \prod_{j=1}^{M} q_{\boldsymbol{\eta}}(z_j|\boldsymbol{x}), \quad q_{\eta}(z_j|\boldsymbol{x}) = \sum_{k=0}^{1} q_{\eta}(m_j = k|\boldsymbol{x})p(z_j|m_j = k). \tag{4}$$

As shown in Vahdat et al. (2018b), the corresponding inverse Cumulative Density Function (CDF) is given by

$$F_{q_{\boldsymbol{\eta}}(z_j|\boldsymbol{x})}^{-1}(\rho) = -\frac{1}{\beta} \log \left( \frac{-b + \sqrt{b^2 - 4c}}{2} \right), \tag{5}$$

where $b = \big(\rho + e^{-\beta}(q_j - \rho)\big)/(1 - q_j) - 1$ and $c = -[q_j e^{-\beta}]/(1 - q_j)$. The equation 5 is a differentiable function that converts a sample $\rho$ from an independent uniform distribution $\mathcal{U}(0, 1)$ into a sample from $q_{\eta}(\boldsymbol{z}|\boldsymbol{x})$. Thus, we can apply the reparameterization trick to sample from the latent variable $\boldsymbol{z}$ and optimize the ELBO with respect to the model's parameters.

## 3 IMPROVING INFERENCE BY ADDING REDUNDANCY TO LATENT VECTORS

In the DVAE framework (Rolfe, 2016; Vahdat et al., 2018b;a), the authors use Boltzmann machines as priors instead of the independent prior $p(\boldsymbol{m})$ presented in Section 2. While these complex priors increase the model's flexibility and can produce competitive results, our objective is to enhance inference through model design by maintaining the simpler independent prior. This approach would improve interpretability and encourage the model to learn independent components in the latent space, which is essential for capturing potentially disentangled representations.

VAEs (Kingma & Welling, 2013) are often viewed as lossy compression models, where the goal is to minimize reconstruction error while imposing regularization through a prior distribution. However, our approach is better understood from a generative standpoint. We first sample a latent vector $\boldsymbol{m}$, generate an observation $\boldsymbol{x}$, and focus on minimizing the error rate when recovering $\boldsymbol{m}$ from $\boldsymbol{x}$. Achieving this requires the variational approximation to be sufficiently accurate. In fields where reliable data transmission or storage is important, introducing ECCs is a well-established approach to reduce the error rate when estimating a discrete source $\boldsymbol{m}$ transmitted through a noisy channel with output $\boldsymbol{x}$. Estimating $\boldsymbol{m}$ from $\boldsymbol{x}$ implies approximating the true and unknown posterior distribution $p(\boldsymbol{m}|\boldsymbol{x})$ with a proposed $q_{\boldsymbol{\eta}}(\boldsymbol{m}|\boldsymbol{x})$. The gap between $q_{\boldsymbol{\eta}}(\boldsymbol{m}|\boldsymbol{x})$ and $p(\boldsymbol{m}|\boldsymbol{x})$ is precisely the variational gap. We propose employing ECCs to safeguard $\boldsymbol{m}$ with controlled and known redundancy that can be leveraged by the variational posterior $q_{\boldsymbol{\eta}}(\boldsymbol{m}|\boldsymbol{x})$ by design. This way, it is possible to reduce the mistakes committed when comparing $\boldsymbol{m}$ with samples drawn from $q_{\boldsymbol{\eta}}(\boldsymbol{m}|\boldsymbol{x})$, obtaining a tighter approximation to the true posterior $p(\boldsymbol{m}|\boldsymbol{x})$, therefore reducing the gap to optimal inference.

ECCs play a crucial role in information theory and digital communications by enabling reliable data transmission over unreliable channels (Moon, 2005). They introduce redundancy into the transmitted data, allowing the receiver to detect errors and, in many cases, correct them without retransmission. In his seminal work, Shannon (Shannon, 1948) demonstrated the arbitrarily reliable communication is possible through error correction. Our approach builds on the idea that the generative model in Fig. 1c can be conceptualized as a communication system, where the bits sampled from $p(\boldsymbol{m})$ undergo continuous modulation into $\boldsymbol{z}$ and are then transmitted through a nonlinear communication channel (in this setting, the decoder NN) characterized by the input/output response $p_{\boldsymbol{\theta}}(\boldsymbol{x}|\boldsymbol{z}) = p(f_{\boldsymbol{\theta}}(\boldsymbol{z}))$. In this scenario, the complexity of the channel is essential since it is necessary to account for the intricate nature of the data at its output (e.g., complex images). Following this idea, the process of inference via $q_{\boldsymbol{\eta}}(\boldsymbol{m}|\boldsymbol{x})$ can be thought of as deciphering the latent variable $\boldsymbol{m}$ given the observed data $\boldsymbol{x}$, where the encoder NN plays the role of the channel equalizer, trying to reverse the channel's effects without knowing the bit correlations from the ECC.

## 4 CODED DVAE

This section extends the previously described DVAE, introducing an ECC over $\boldsymbol{m}$. We refer to this model as coded DVAE. In ECCs, we augment the dimensionality of the binary latent space from $M$ to $D$ in a controlled and deterministic manner, where $R = M/D$ is the *coding rate*. An ECC is typically designed so that the $2^M$ possible codewords are separated as much as possible in the space of binary vectors of $D$ bits. This facilitates algorithms in detecting and/or correcting errors by searching for the nearest code word. A random choice of the codewords brings what is known as a *random block code* (Shannon, 1948). While they are known to be very robust and amenable to theoretical analysis, their lack of structure makes them computationally intractable since we have to rely on codeword enumeration during the encoding/decoding process. In Appendix M, we include the formulation of a random code's encoding/decoding process within the DVAE model.

Instead, we adopt a much simpler linear coding scheme, namely repetition codes. In a repetition code, each bit of the original message $\boldsymbol{m}$ is repeated multiple times to create the encoded message $\boldsymbol{c}$. Intuitively, the more times an information bit is repeated, the better it is protected. Our experiments consider uniform $(M, D)$ repetition codes where all bits are repeated $L$ times, resulting in codewords of dimension $D = ML$ and a coding rate of $R = 1/L$. Note that repetition codes represent a special case of linear ECCs since each codeword can be deterministically computed by multiplying a binary vector $\boldsymbol{m}$ by an $M \times D$ *generator matrix* $\mathbf{G}$, such that $\boldsymbol{c} = \boldsymbol{m}^T \mathbf{G}$, where $u$-th row, with $u = 1, \ldots, M$, has entries equal to one at columns $L(u-1)+1, L(u-1)+2, \ldots, Lu$, and zero elsewhere. For example, for $M = 3$ and $L = 2$, the generator matrix of the $(3, 6)$ repetition code is

$$\mathbf{G} = \begin{bmatrix} 1 & 1 & 0 & 0 & 0 & 0 \\ 0 & 0 & 1 & 1 & 0 & 0 \\ 0 & 0 & 0 & 0 & 1 & 1 \end{bmatrix}. \tag{6}$$

The generative process of the coded DVAE follows similarly to the uncoded case, and it is represented in Fig. 1d. We assume the same prior distribution $p(\boldsymbol{m})$, but in this case the samples $\boldsymbol{m}$ are

deterministically encoded using $\mathbf{G}$. Now, the smoothing $\boldsymbol{z}$ transformations are defined over $\boldsymbol{c}$

$$p(\boldsymbol{z}|\boldsymbol{c}) = \prod_{j=1}^{D} p(z_j|c_j), \quad p(z_j|c_j) = \frac{e^{-\beta(z_j - c_j)}}{Z_\beta}, \tag{7}$$

for $z_j \in [0, 1]$, $c_j \in \{0, 1\}$ and $Z_\beta = (1 - e^{-\beta})/\beta$. The likelihood $p(\boldsymbol{x}|\boldsymbol{z})$ is again of the form $p_{\boldsymbol{\theta}}(\boldsymbol{x}|\boldsymbol{z}) = p(f_{\boldsymbol{\theta}}(\boldsymbol{z}))$. Note that, compared to the uncoded case, we have a larger input dimensionality to the decoder NN $f_{\boldsymbol{\theta}}(\boldsymbol{z})$. When comparing uncoded vs. coded DVAEs, the structure of the decoder NN $f_{\boldsymbol{\theta}}(\boldsymbol{z})$ (detailed in Appendix C) is equal in both cases except for the first Multilayer Perceptron (MLP) layer that attacks the input $\boldsymbol{z}$. Therefore, if a rate $R = 1/L$ repetition code is used, the number of additional parameters of the $f_{\boldsymbol{\theta}}(\boldsymbol{z})$ NN is given by $(L-1) \times h$, where $h$ is the dimension of the first hidden space of $f_{\boldsymbol{\theta}}(\boldsymbol{z})$.

**Variational family and inference**

The repetition code introduces correlations between the bits in $\boldsymbol{c}$ that we will exploit to obtain an improved variational bound. We again assume a variational family factorizing as

$$q_{\boldsymbol{\eta}}(\boldsymbol{m}, \boldsymbol{z}|\boldsymbol{x}) = q_{\boldsymbol{\eta}}(\boldsymbol{m}|\boldsymbol{x})p(\boldsymbol{z}|\boldsymbol{c}) \tag{8}$$

where $q_{\boldsymbol{\eta}}(\boldsymbol{m}|\boldsymbol{x}) = \prod_{u=1}^{M} q_{\boldsymbol{\eta}}(m_u|\boldsymbol{x})$ is computed in two steps. First, we construct an encoder NN $\mathbf{g}_{\boldsymbol{\eta}}(\boldsymbol{x})$ similar to that of equation 2, that retrieves the probabilities of the bits in $\boldsymbol{c}$ from $\boldsymbol{x}$ without exploiting the correlations introduced by the repetition code:

$$q_{\boldsymbol{\eta}}^u(\boldsymbol{c}|\boldsymbol{x}) = \prod_{j=1}^{D} \text{Ber}(g_{j,\boldsymbol{\eta}}(\boldsymbol{x})), \tag{9}$$

where the $u$ superscript serves as a reminder that this posterior does not exploit the redundancy introduced by the ECC.

Now, we utilize the known redundancy introduced by the ECC to constrain the solution of $q_{\boldsymbol{\eta}}^u(\boldsymbol{c}|\boldsymbol{x})$, given that each bit from $\boldsymbol{m}$ has been repeated $L$ times to create $\boldsymbol{c}$. To do so, we follow a *soft decoding* approach, where the marginal posteriors of the information bits are derived from the marginal posteriors of the encoded bits, exploiting the repetition code's known structure. In the case of repetition codes, we compute the all-are-zero and the all-are-ones products of probabilities of the bits in $\boldsymbol{c}$ that are copies of the same message bit and renormalize as

$$q(m_u = 1|\boldsymbol{x}) = \frac{1}{Z} \prod_{j=L(u-1)+1}^{Lu} g_{j,\boldsymbol{\eta}}(\boldsymbol{x}) \doteq \frac{g_{u,\boldsymbol{\eta}}^+(\boldsymbol{x})}{Z}, \tag{10}$$

$$q(m_u = 0|\boldsymbol{x}) = \frac{1}{Z} \prod_{j=L(u-1)+1}^{Lu} (1 - g_{j,\boldsymbol{\eta}}(\boldsymbol{x})) \doteq \frac{g_{u,\boldsymbol{\eta}}^-(\boldsymbol{x})}{Z}, \tag{11}$$

for $u = 1, \ldots, M$ and $Z = \left(g_{u,\boldsymbol{\eta}}^+(\boldsymbol{x}) + g_{u,\boldsymbol{\eta}}^-(\boldsymbol{x})\right)^{-1}$. This approach can be seen as a soft majority voting strategy, enabling the recovery of the original information vector even if some bits in the inferred encoded word are corrupted. All operations in equation 10 preserve the gradients concerning the parameters in the encoder $\mathbf{g}_{\boldsymbol{\eta}}(\boldsymbol{x})$. We implement them in the log domain for stability.

When compared to the uncoded case, as in the likelihood term, we consider the same NN structure for the encoder $\mathbf{g}_{\boldsymbol{\eta}}(\boldsymbol{x})$ where both cases only differ in the last MLP layer. The additional overhead in the coded cases requires $(L-1) \times h'$ parameters in the last layer, where $h'$ is the dimension at the output of the last layer. In Appendix J.2, we conduct an ablation study on the number of trainable parameters to demonstrate that the improvement in performance does not stem from this increase in the number of parameters, but rather from the incorporation of the ECC in the latent space.

**Soft encoding for efficient reparameterization**

Given the variational family in equation 8, the ELBO matches the expression in equation 3. However, the reparameterization trick in equation 5 requires independent bits, which is not the case in $\boldsymbol{c}$. To efficiently circumvent this issue during training, we employ a soft encoding approach. With soft encoding, a marginal probability is computed for each bit in the codeword $\boldsymbol{c}$, taking into account

the structure of the ECC and the marginal probabilities of the information bits. For a repetition code, this involves simply replicating the posterior probabilities $q_{\boldsymbol{\eta}}(\boldsymbol{m}|\boldsymbol{x}) = \prod_{u=1}^{M} q_{\boldsymbol{\eta}}(m_u|\boldsymbol{x})$ for each copy of the same information bit. Hence, we treat the bits in $\boldsymbol{c}$ as independent but distributed according to $q_{\boldsymbol{\eta}}(\boldsymbol{m}|\boldsymbol{x})$. The algorithm in Appendix B shows the training pseudo-code.

When marginalizing $\boldsymbol{c}$ using the soft encoding marginals, we disregard potential correlations between the coded bits. For instance, with repetition codes, sampling from the marginals could produce inconsistent bits, leading to an invalid codeword. However, since we do not sample the coded bits during training but instead propagate their marginal probabilities, we consider this approximation to have minimal negative impact. In fact, it can be seen as a form of probabilistic dropout, which enhances robustness during training. It is important to note that when sampling from the generative model in test time, we use hard bits encoded into valid codewords, yielding visually appealing samples, indicating that our training approach is reliable.

**Related work**

While the use of deep neural networks and generative models in digital communications problems have been profusely reported in recent years (see Ye et al. (2024), Chen et al. (2024), Guo et al. (2022), Wu et al. (2023), and Shen et al. (2023) for representative examples), the use of ECC techniques as a design tool in machine learning is scarce. The most prominent example is Dietterich & Bakiri (1995), where the authors address multiclass learning problems via ECCs. In Aldaghri et al. (2021), the authors proposed using linear codes for applications that may require removing the trace of a sample from the system, e.g., a user requests their data to be deleted or corrupted data is discovered. They address a regression problem by introducing a coded learning protocol that employs linear encoders to divide the training data before the learning phase. More recently, in Xue et al. (2024) the authors introduced ECCs to improve code-to-code translation using transformers.

## 5 EXPERIMENTS

This section empirically evaluates the DVAE and its coded counterpart using repetition codes. We show results on reconstruction and generation tasks. In particular, we display results for MNIST (Deng, 2012), FMNIST (Xiao et al., 2017), CIFAR10 (Krizhevsky et al., 2009), and Tiny ImageNet (Le & Yang, 2015) datasets. The selection of these relatively simple datasets is deliberate to aid in a clearer understanding of the behaviors of various configurations. Additionally, we compared the coded model to the uncoded DVAE trained using the IWAE objective (Burda et al., 2016), as presented in Appendix H. All experimental results were obtained using the same architecture, which is detailed in Appendix C. When introducing the repetition code, we only modify the encoder's output layer and the decoder's input layer to adapt the architecture to the augmented dimension.

### 5.1 RECONSTRUCTION

We first evaluate the model's performance of reconstructing data by examining its *uncoded* and *coded* versions across different configurations, varying the number of information bits and code rates. The introduction of the repetition code led to improved reconstruction and smaller KL values, indicating that the posterior latent features are disentangled and less correlated. In Appendices D, E, F and G, we show the behavior of the ELBO loss function for all the models and datasets.

**Image reconstruction quality**.   In the table included in Fig. 2, we first quantify the quality of the reconstructions in FMNIST by measuring the Peak Signal-To-Noise Ratio (PSNR) in the test set. The results for the rest of the datasets are provided in sections D, E, F and G of the Appendix. In all the cases, the coded models yield higher PSNR values than their uncoded counterparts, indicating a superior performance in reconstruction. This improvement is also evident by visual inspection of Fig. 2, where the coded models exhibit a greater ability to capture details in the images for the same latent dimension. We observe a general improvement in PSNR as we increase the number of information bits, i.e., as we augment the latent dimensionality of the model. This increase in the number of available latent vectors provides greater flexibility, enabling the models to capture the underlying structure of the data more effectively. We also observe a general improvement in PSNR as we decrease the code rate, i.e., as we add more redundancy. Note that adding redundancy does not increase the model's flexibility, since the information bits determine the number of latent vectors. However, coded models yield more accurate and detailed reconstructions.

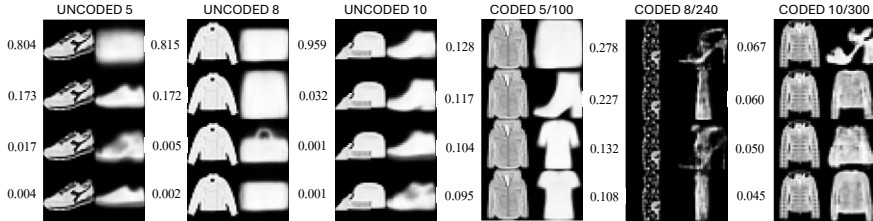

| Model | PSNR | Acc | Conf. Acc | Entropy |
|---|---|---|---|---|
| uncoded 5 | 14.477 | 0.536 | 0.536 | 0.237 |
| coded 5/50 | 16.241 | 0.647 | 0.700 | 1.899 |
| coded 5/80 | 16.624 | 0.688 | 0.748 | 2.180 |
| coded 5/100 | 16.702 | 0.700 | 0.757 | 2.256 |
| uncoded 8 | 15.598 | 0.594 | 0.595 | 0.467 |
| coded 8/80 | 17.318 | 0.750 | 0.816 | 2.905 |
| coded 8/160 | 17.713 | 0.783 | 0.831 | 3.637 |
| coded 8/240 | 17.861 | 0.799 | **0.893** | 4.000 |
| uncoded 10 | 16.000 | 0.644 | 0.648 | 0.659 |
| coded 10/100 | 17.694 | 0.790 | 0.850 | 3.879 |
| coded 10/200 | 18.009 | 0.814 | 0.871 | 4.609 |
| coded 10/300 | **18.111** | **0.817** | 0.870 | 5.076 |

Figure 2: **Reconstruction performance over the test set in FMNIST**. The figure at the left shows an example of reconstructed test images obtained with different model configurations. Observe that more details are visualized as we increase the number of bits in the latent space and decrease the coding rate. The table at the right includes reconstruction metrics. Acc is the semantic accuracy and Conf. Acc the confident semantic accuracy. Entropy is the average entropy of $q_{\boldsymbol{\eta}}(\boldsymbol{m}|\boldsymbol{x})$ in the test set.

Figure 3: **Example of erroneous reconstructions in FMNIST** using the 4 most-probable a posteriori latent vectors. The first column in each image shows the original input to the model, while the second column displays the reconstructions. The $q_{\boldsymbol{\eta}}(\boldsymbol{m}|\boldsymbol{x})$ probability is indicated in each row.

**Semantic accuracy**.    As the PSNR operates at the pixel level, it does not account for the *semantic* errors committed by the model. For example, if the model incorrectly reconstructs a nine instead of a four in the MNIST dataset, the PSNR may still yield a large value due to the similarity between the two images. Nonetheless, this would represent a severe failure in correct reconstruction of the intended class. Therefore, we additionally evaluate the reconstruction accuracy, ensuring that the model successfully reconstructs images within the same class as the original ones. For this purpose, we trained an image classifier for each dataset and compared the reconstructed images' predicted labels against the originals' ground truth labels. Additionally, we provide a *confident* reconstruction accuracy. While the reconstruction accuracy is computed across the entire dataset partitions, for the *confident* accuracy, we only consider those images projected into a latent vector with a probability exceeding 0.4.[1] Results for FMNIST are detailed in the table included in Fig.2, and corresponding results for MNIST are available in Table 2 within Appendix E. In light of the results, we can conclude that introducing an ECC in the model allows for latent spaces that better capture the semantics of the images while employing the same number of latent vectors, significantly outperforming the uncoded models in terms of accuracy in all the cases.

**Posterior uncertainty calibration**.    Finally, also in the table included in Fig.2, we report the average entropy of the variational posterior $q_{\boldsymbol{\eta}}(\boldsymbol{m}|\boldsymbol{x})$ over the test set. The low entropy observed in the uncoded models suggests a low uncertainty when the model projects data points into the latent space, which could be advantageous if the model consistently assigned high probability to the correct latent vectors. However, the semantic accuracy results demonstrate this is not true in the

---

[1]Namely, we do not count errors when the Maximum a Posteriori (MAP) value of $q_{\boldsymbol{\eta}}(\boldsymbol{m}|\boldsymbol{x})$ is below 0.4.

| Model | BER | WER | LL train | LL test |
|---|---|---|---|---|
| uncoded 5 | 0.051 | 0.195 | -266.157 | -267.703 |
| coded 5/50 | 0.011 | 0.046 | -239.379 | -241.882 |
| coded 5/80 | 0.008 | 0.039 | -227.550 | -232.992 |
| coded 5/100 | 0.010 | 0.049 | -238.206 | -241.404 |
| | | | | |
| uncoded 8 | 0.089 | 0.384 | -247.964 | -249.880 |
| coded 8/80 | 0.021 | 0.144 | -227.550 | -232.992 |
| coded 8/160 | 0.027 | 0.189 | -228.585 | -235.819 |
| coded 8/240 | 0.037 | 0.231 | -231.679 | -238.459 |
| | | | | |
| uncoded 10 | 0.142 | 0.622 | -242.842 | -244.997 |
| coded 10/100 | 0.040 | 0.321 | -222.011 | -230.772 |
| coded 10/200 | 0.044 | 0.341 | -223.748 | -234.849 |
| coded 10/300 | 0.045 | 0.349 | -226.504 | -238.647 |

Figure 4: **Evaluation of generation in FMNIST.** The figure at the left shows an example of randomly generated, uncurated FMNIST images. The table at the right shows the quantitative results on the evaluation of the Bit Error Rate (BER), Word Error Rate (WER), and log-likelihood (LL).

uncoded model. In other words, the uncoded variational family projects images into the wrong class with high confidence. This indicates the uncertainty of the uncoded case is severely miscalibrated.

Coded models, on the other hand, improve semantic accuracy and present a larger entropy. This suggests that i) the coded DVAE is aware that multiple latent vectors might be related to the image class and ii) that the model posterior shows large uncertainties (high entropy) for certain images for which the model has not properly identified the class. We illustrate this in Fig. 3, where we show some images that were selected so that the MAP latent word from $q_{\eta}(m|x)$ induces class reconstruction errors. We display the reconstruction of the 4 most probable latent vectors and their corresponding probabilities. Observe that the uncoded model is confident no matter the reconstruction outcome while, in the coded posterior, the uncertainty is much larger. These results are also observed for MNIST, indicating that the posterior distribution in coded models exhibits a better uncertainty calibration. Note also that the increase in the number of latent bits (from 8 to 10) does not result in an excessive increase in the entropy despite the exponential growth of the number of vectors.

## 5.2 GENERATION

In this section, we evaluate the model for the image generation task. In Fig. 4, we show examples of randomly generated images using different model configurations in FMNIST. Results for the rest of the datasets are available in Appendices D, E, F, and G. These results are consistent with the ones obtained in reconstruction since we can observe that the coded models can generate more detailed and diverse images. Both uncoded and coded models generate more intricate and varied images with increased information bits. However, if the number of latent vectors becomes too large for the dataset's complexity, not all words in the codebook are specialized during model training. This leads to generation artifacts, images where different classes of objects are overlapped. A visual inspection of Fig. 4 suggests these artifacts are more frequent in the uncoded case. Note that, since we are dealing with discrete latent variables, we could simply detect and prune uninformative vectors.

**Accuracy metrics in generation**. The improved inference given by the repetition code can also be tested by generating images using the generative model and counting errors using the MAP solution of the variational distribution $q_{\eta}(m|x)$. The table included in Fig. 4 reports the BER and WER for FMNIST. As expected, at the same number of latent bits, the coded models significantly reduce both the BER and WER w.r.t. the uncoded case. Note also that the error rates grow with the number of latent bits, which is expected due to the increased complexity of the inference process. We may commit more errors by taking the MAP, but errors typically fall in consistent reconstructions (latent words that also reconstruct the same type of image), as the results presented in Section 5.1 indicated.

**Log-likelihood**. We additionally estimated the log-likelihood (LL). Results for FMNIST with different model configurations, estimated through importance sampling with 300 samples per observation, are presented in the table included in Fig. 4, please refer to Appendix K for further details.

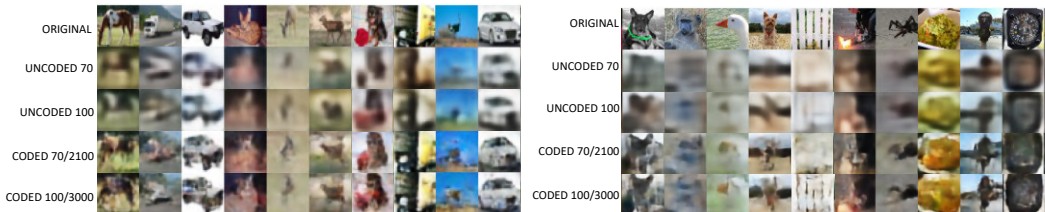

Figure 5: **Reconstruction results** for CIFAR10 (left) and Tiny ImageNet (right).

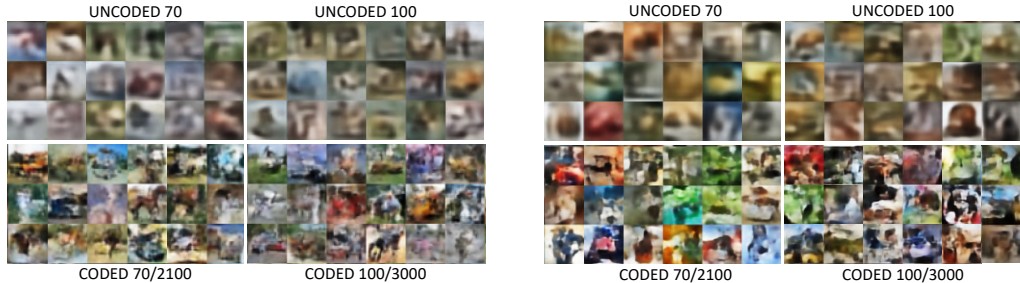

Figure 6: **Generation results** for CIFAR10 (left) and Tiny ImageNet (right).

We observe that coded models consistently outperform their uncoded counterparts for both train and test sets, aligning with the results previously presented. Thus, we can argue that the introduction of repetition codes in the definition of the model allows for an improved inference and tighter posterior approximation. We draw similar conclusions for other datasets.

We observe a general improvement in LL values as we increase the number of information bits, i.e., as we augment the latent dimensionality of the model and its flexibility. However, reducing the code rate does not lead to an improvement in log-likelihood. We argue that this might indicate overfitting of the decoder, as the LL deteriorates while reconstruction metrics improve. We must note that we use feed-forward networks at the decoder's input. However, this may not be appropriate for the correlations we present in our coded words. We might overcome this overfitting tendency by using an architecture that properly leverages these coded bits correlations.

## 5.3 ADDITIONAL RESULTS WITH CIFAR10 AND TINY IMAGENET

Since MNIST-like datasets are rather simple, it is difficult to assess the true gain in performance resulting from the introduction of ECCs proposed in our model. This section presents additional results using CIFAR10 and Tiny ImageNet, which contain colored images with more intricate shapes, patterns, and greater diversity than the previous datasets. We trained uncoded and coded models using different configurations to gain intuition regarding the effect of introducing the ECC. For a reference, in the case of the DVAE++ (Vahdat et al., 2018b), the authors needed 128 binary latent variables to achieve state-of-the-art performance in generation and reconstruction for this dataset. They employed a more intricate model than the one introduced in this study, featuring Boltzmann Machine priors. In Fig. 5 we show examples of reconstruction using different configurations of the model and in Fig. 6 we show examples of randomly generated images. Additional results are provided in Appendices F and G. The results are consistent with those presented in previous sections; but in this case, the difference in performance is even more pronounced. We observe that the uncoded DVAE cannot decouple spatial information from the images and project it in the latent space. Nevertheless, the coded DVAE shows particular promise for learning low-dimensional discrete latent representations in complex datasets. Note that we used a rather simple architecture as we want to focus on the gain obtained only by introducing ECCs in the latent space.

## 6 BEYOND REPETITION CODES

We have presented compelling proof-of-concept results that incorporating ECCs, like repetition codes, into DVAEs can improve performance. We believe this opens a new path for designing latent probabilistic models with discrete latent variables. Although a detailed analysis of the joint design of ECC and encoder-decoder networks is beyond the scope of this work, we will outline key properties that any ECCs must satisfy to be integrated within this framework.

- **Scalable hard encoding** $(m \to c)$. Our model requires hard encoding for generation once the model is trained. This process should have linear complexity in $M$.
- **Scalable soft encoding** $(p(m) \to p(c))$. Soft encoding is required during training for reparameterization. This process should also have linear complexity in $M$.
- **Scalable soft decoding** $(p(c) \to p(m))$. Our model employs soft-in soft-out (SISO) decoding during inference. This process should again be linearly complex in $M$.
- **Differentiability.** Both encoding and decoding processes must be differentiable w.r.t the inputs to enable gradient computation and backpropagation.

Since Shannon's seminal work (Shannon, 1948), researchers have developed effective ECC schemes that meet these properties, including state-of-the-art ECCs such as Low Density Parity Check (LDPC) codes (Gallager, 1962), or polar codes (Arikan, 2009). Efforts have also focused on developing efficient SISO decoders, such as the sum-product algorithm (Kschischang et al., 2001).

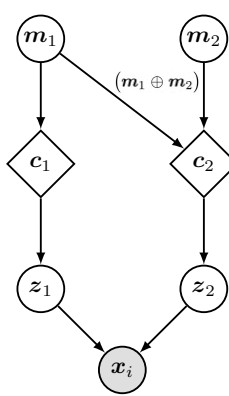

Figure 7: Graphical model of the hierarchical coded DVAE.

Inspired by polar codes (Arikan, 2009), we present a hierarchical coded DVAE with two layers of latent bits. In this model, the latent bits $m_1$ are encoded using a repetition code in the first layer, producing $c_1$ and $z_1$. Simultaneously, the vector $m_2$ is linearly combined with $m_1$ using modulo 2 operations $(m_1 \oplus m_2)$ and then encoded using a repetition code, yielding $c_2$ and $z_2$. Both soft vectors are concatenated and fed to the decoder NN to generate $x$. The model provides stronger protection for $m_1$, as it appears in both branches of the generative model. Inference follows a similar approach to the coded DVAE, incorporating the linear combination of $m_1$ and $m_2$ used in the second branch. This hierarchical structure allows the model to effectively separate high-level information from finer details, as we show in the results presented in Appendix I.

## 7 CONCLUSION

This paper presents the first proof-of-concept demonstration that safeguarding latent information with ECCs within deep generative models holds promise for enhancing overall performance. By integrating redundancy into the latent space, the variational family can effectively refine the inference network's output according to the structure of the ECC. Our findings underscore the efficacy of simple and efficient ECCs, like repetition codes, showcasing remarkable improvements over a lightweight version of the DVAE introduced in Vahdat et al. (2018b).

Furthermore, our work reveals numerous avenues for future research. Firstly, investigating decoder architectures capable of efficiently utilizing the correlations and structure introduced by the ECCs, in contrast to the feed-forward networks employed in this study. We also contemplate exploring more complex and robust coding schemes, conducting theoretical analyses aligned with Shannon's channel capacity and mutual information concepts to determine the fundamental parameters of the ECC needed to achieve reliable variational inference, exploring different modulations, and integrating these concepts into state-of-the-art models based on discrete representations.

## 8 REPRODUCIBILITY STATEMENT

Our supplementary materials and appendices contain all the necessary information to facilitate reproducibility. We provide the model's source code along with examples for training and evaluation.

Pseudo-codes outlining the training process are included in Appendices A, B, and M. Appendix C describes the encoder and decoder architectures used for the experiments, and Appendix N outlines the computational resources utilized for the experimental results. Furthermore, all experiments were conducted using widely known public datasets.

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

# Appendices

The following Appendices offer further details on the model architecture, implementation, and experimental setup. They also include additional results on the FMNIST (Xiao et al., 2017), MNIST (Deng, 2012), CIFAR10 (Krizhevsky et al., 2009), and Tiny ImageNet (Le & Yang, 2015) datasets, along with comparisons to uncoded models trained with the IWAE objective (Burda et al., 2016), and a description of the hierarchical coded DVAE. Given the length of the material, we have included a Table of Contents for easier navigation.

## A  UNCODED TRAINING ALGORITHM

The following pseudo-code describes the training process for the uncoded DVAE. It's important to note that the main difference from the training of the coded DVAE lies in the fact that the encoder directly outputs $q_{\boldsymbol{\eta}}^u(\boldsymbol{m}|\boldsymbol{x}_i)$, which is used to sample $\boldsymbol{z}$. Therefore, we skip the soft decoding and coding steps.

---

**Algorithm 1** Training the model with *uncoded* inference.

---

1: **Input:** training data $\boldsymbol{x}_i$.
2: **repeat**
3:   $q_{\boldsymbol{\eta}}^u(\boldsymbol{m}|\boldsymbol{x}_i) \leftarrow$ forward encoder $g_{\boldsymbol{\eta}}(\boldsymbol{x}_i)$
4:   $\boldsymbol{z} \leftarrow$ sample from equation 5
5:   $p_{\boldsymbol{\theta}}(\boldsymbol{x}|\boldsymbol{z}) \leftarrow$ forward decoder $f_{\boldsymbol{\theta}}(\boldsymbol{z})$
6:   Compute ELBO according to equation 3
7:   $\boldsymbol{\theta}, \boldsymbol{\eta} \leftarrow Update(ELBO)$
8: **until** convergence

---

## B  CODED TRAINING ALGORITHM

The following pseudo-code describes the training process for the coded DVAE. Here, we utilize soft decoding to leverage the added redundancy and retrieve the marginal posteriors of the information bits $\boldsymbol{m}$, correcting potential errors in $q_{\boldsymbol{\eta}}^u(\boldsymbol{c}|\boldsymbol{x}_i)$. We then apply the soft encoding technique to incorporate the structure of the code and sample $\boldsymbol{z}$ using the reparameterization trick as described in equation 5.

---

**Algorithm 2** Training the coded DVAE with repetition codes.

---

1: **Input:** training data $\boldsymbol{x}_i$, matrix $\mathbf{G}$.
2: **repeat**
3:   $q_{\boldsymbol{\eta}}^u(\boldsymbol{c}|\boldsymbol{x}_i) \leftarrow$ forward encoder $g_{\boldsymbol{\eta}}(\boldsymbol{x}_i)$
4:   $q_{\boldsymbol{\eta}}(\boldsymbol{m}|\boldsymbol{x}_i) \leftarrow$ soft decoding by aggregating $q_{\boldsymbol{\eta}}^u(\boldsymbol{c}|\boldsymbol{x}_i)$ according to equation 10
5:   $q_{\boldsymbol{\eta}}(\boldsymbol{c}|\boldsymbol{x}_i) \leftarrow$ repeat posterior bit probabilities $q_{\boldsymbol{\eta}}(\boldsymbol{m}|\boldsymbol{x}_i)$ according to $\mathbf{G}$
6:   $\boldsymbol{z} \leftarrow$ sample from equation 5
7:   $p_{\boldsymbol{\theta}}(\boldsymbol{x}|\boldsymbol{z}) \leftarrow$ forward decoder $f_{\boldsymbol{\theta}}(\boldsymbol{z})$
8:   Compute ELBO according to equation 3
9:   $\boldsymbol{\theta}, \boldsymbol{\eta} \leftarrow Update(ELBO)$
10: **until** convergence

---

## C  ARCHITECTURE

In this section, we detail the architecture used to obtain the experimental results with FMNIST and MNIST (28x28 gray-scale images). Note that across experiments we only modify the output layer of the encoder and the input layer of the decoder to adapt to the different configurations of the model. This modification leads to a minimal alteration in the total number of parameters. In Section J.2, we conduct an ablation study on the number of trainable parameters to show that the enhancement in performance is not attributed to the increased dimensionality introduced by redundancy.

For the additional CIFAR10 experiments, we change the input of the encoder and the output of the decoder to process the 32x32 color images. For the Tiny ImageNet experiments, we do the same to process 64x64 color images. The rest of the architecture remains unchanged.

These architectures are comprehensively described in the following subsections.

### C.1  ENCODER

The encoder NN consists of 3 convolutional layers followed by two fully connected layers. We employed Leaky ReLU as the intermediate activation function and a Sigmoid as the output activation, as the encoder outputs bit probabilities. The full architecture is detailed in Fig. 8.

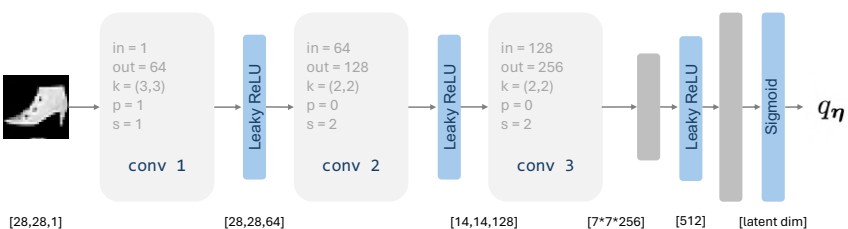

Figure 8: Block diagram of the **encoder architecture** for FMNIST and MNIST.

## C.2 DECODER

The decoder architecture is inspired by the one proposed in Schuster & Krogh (2023). It is composed of two fully connected layers, followed by transposed convolutional layers with residual connections and Squeeze-and-Excitation (SE) layers (Hu et al., 2018). We employed Leaky ReLU as the intermediate activation function and a Sigmoid as output activation, given that we consider datasets with gray-scale images. The complete architecture is detailed in Fig. 9.

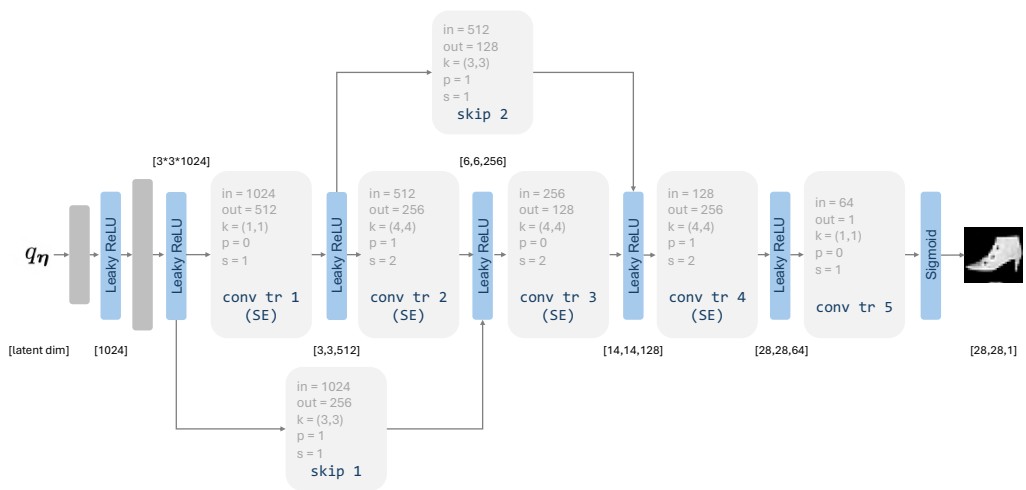

Figure 9: Block diagram of the **decoder architecture** for FMNIST and MNIST.

## D FMNIST RESULTS

In this section, we present supplementary results obtained with the FMNIST dataset.

### D.1 TRAINING

We present the evolution of the ELBO and its terms throughout the training process. The models were trained for 200 epochs using an Adam optimizer with a learning rate of $10^{-4}$, and a batch size of 128. Fig. 10 displays the results for configurations with 5 information bits, Fig. 11 for 8 information bits, and Fig. 12 for 10 information bits. The colors in all plots represent the various code rates.

Across all cases, coded models achieve superior bounds. The main differences in the ELBO come from the different performances in reconstruction. As we have observed in the different experi-

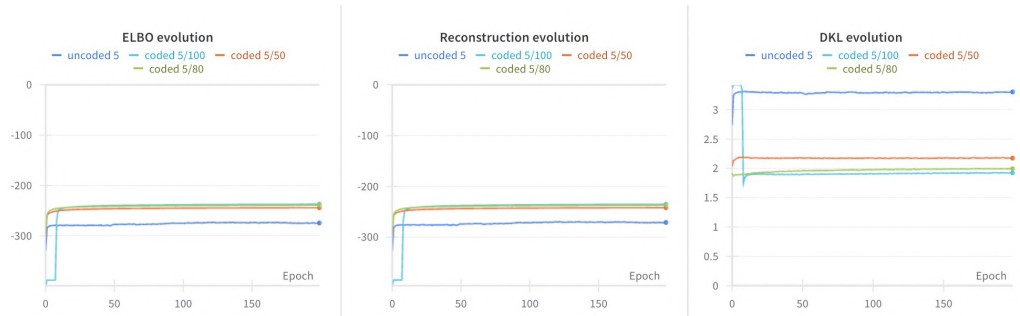

Figure 10: Evolution of the ELBO during training with 5 information bits on FMNIST.

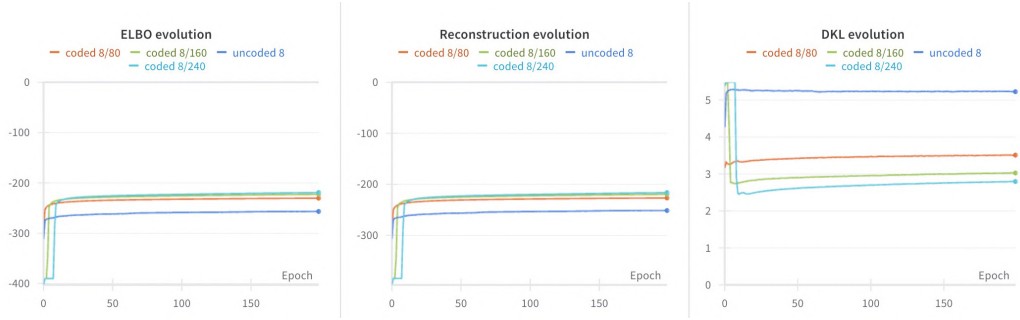

Figure 11: Evolution of the ELBO during training with 8 information bits on FMNIST.

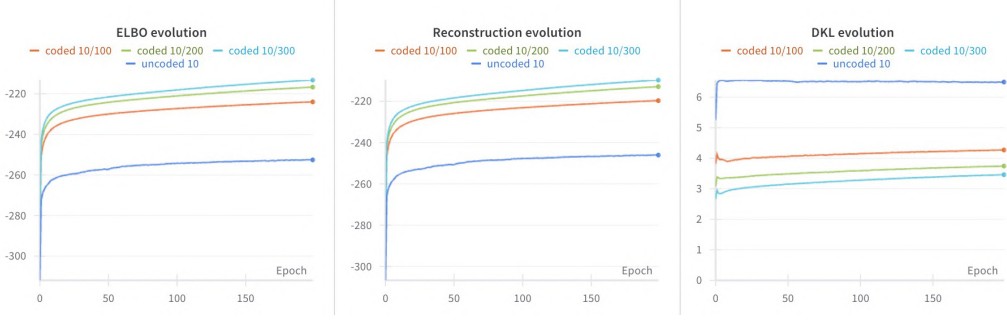

Figure 12: Evolution of the ELBO during training with 10 information bits on FMNIST.

ments, coded models are capable of generating more detailed images and accurate reconstructions. Introducing the repetition code also leads to smaller KL values, indicating that the posterior latent features are disentangled and less correlated.

We observe that, as we decrease the code rate, we obtain better bounds in general. Adding redundancy does not increase the model's flexibility, since the information bits determine the number of

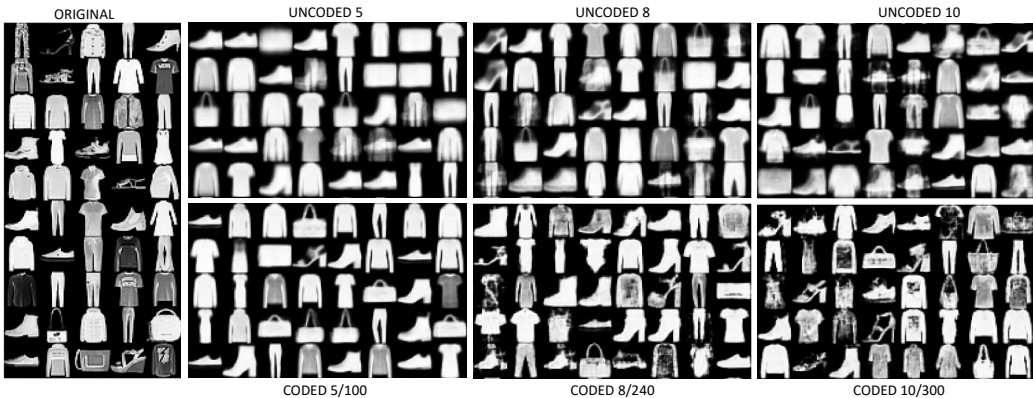

Figure 13: Example of randomly generated, uncurated images using different model configurations.

Table 1: Evaluation of reconstruction performance in FMNIST.

| Model | PSNR (train) | Acc (train) | Conf. Acc. (train) | PSNR (test) | Acc (test) | Conf. Acc. (test) |
|---|---|---|---|---|---|---|
| uncoded 5 | 14.490 | 0.541 | 0.541 | 14.477 | 0.536 | 0.536 |
| coded 5/50 | 16.375 | 0.656 | 0.702 | 16.241 | 0.647 | 0.700 |
| coded 5/80 | 16.824 | 0.694 | 0.751 | 16.624 | 0.688 | 0.748 |
| coded 5/100 | 17.001 | 0.708 | 0.760 | 16.702 | 0.700 | 0.757 |
| | | | | | | |
| uncoded 8 | 15.644 | 0.601 | 0.602 | 15.598 | 0.594 | 0.595 |
| coded 8/80 | 17.877 | 0.769 | 0.842 | 17.318 | 0.750 | 0.816 |
| coded 8/160 | 18.828 | 0.807 | 0.878 | 17.713 | 0.783 | 0.831 |
| coded 8/240 | 19.345 | 0.831 | 0.921 | 17.861 | 0.799 | **0.893** |
| | | | | | | |
| uncoded 10 | 16.053 | 0.650 | 0.652 | 16.000 | 0.644 | 0.648 |
| coded 10/100 | 18.827 | 0.813 | 0.885 | 17.694 | 0.790 | 0.850 |
| coded 10/200 | 19.937 | 0.846 | 0.897 | 18.009 | 0.814 | 0.871 |
| coded 10/300 | **20.529** | **0.855** | **0.907** | **18.111** | **0.817** | 0.870 |

latent vectors. However, the introduction of ECCs in the model allows for latent spaces that better capture the structure of the images while employing the same number of latent vectors.

## D.2 RECONSTRUCTION AND GENERATION

In this section, we augment the results presented in the main text, including outcomes obtained with the training dataset in Table 1. We include again the results obtained in the test to facilitate comparison. The results remain consistent across the two data partitions, and the analysis conducted for the test set also applies to training data.

In all the cases, the coded models yield higher PSNR values than their uncoded counterparts, indicating a superior performance in reconstruction. We observe a general improvement in PSNR as we increase the number of information bits (i.e., as we augment the latent dimensionality of the model) and decrease the code rate (i.e., as we introduce more redundancy).

As we discussed in the main text, the PSNR does not account for the semantic errors committed by the model. Therefore, we additionally report the semantic accuracy and the *confident* semantic accuracy. While the reconstruction accuracy is computed across the entire dataset partitions, for the confident accuracy, we only consider those images projected into a latent vector with a probability exceeding $0.4$. We observe that coded models better capture the semantics of the images while employing the same number of latent vectors, significantly outperforming the uncoded models in terms of accuracy in all the cases.

In Fig 13 we include additional examples of randomly generated images using different model configurations. We observe that coded models can generate more detailed and diverse images than their uncoded counterparts.

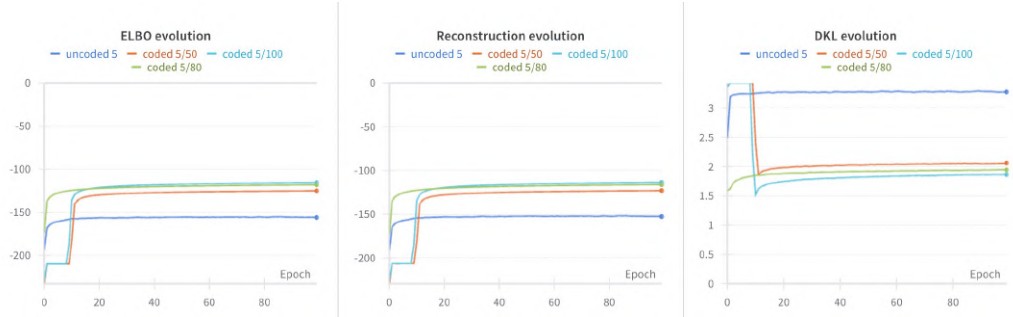

Figure 14: Evolution of the ELBO during training with 5 information bits on MNIST.

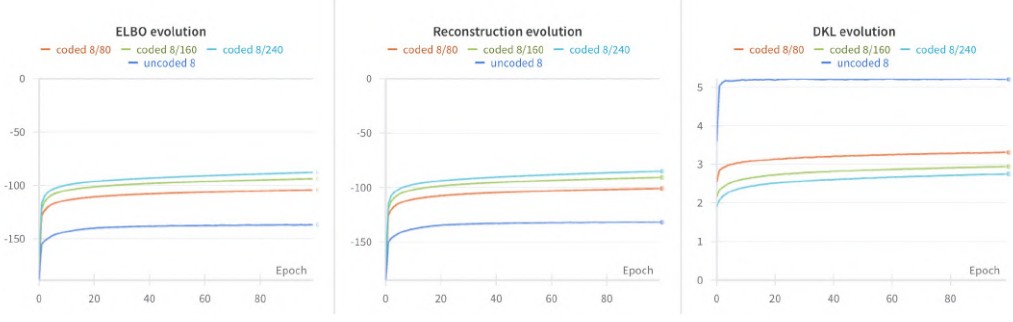

Figure 15: Evolution of the ELBO during training with 8 information bits on MNIST.

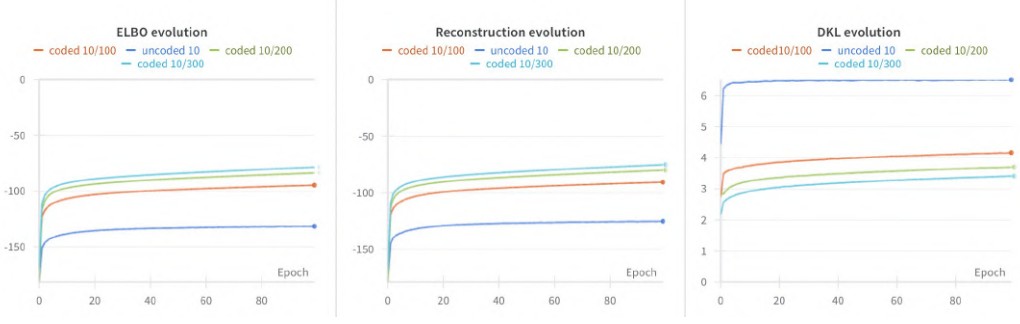

Figure 16: Evolution of the ELBO during training with 10 information bits on MNIST.

# E MNIST RESULTS

In this section, we report the results obtained with the MNIST dataset.

## E.1 TRAINING

We present the evolution of the ELBO and its terms throughout the training process. The models were trained for 100 epochs using an Adam optimizer with a learning rate of $10^{-4}$, and a batch size of 128. Fig. 14 displays the results for configurations with 5 information bits, Fig. 15 for 8 information bits, and Fig. 16 for 10 information bits. The colors in all plots represent the various code rates.

The results are consistent with the ones obtained for FMNIST. Across all the configurations, coded models achieve superior bounds. The main differences in the ELBO come from the different performances in reconstruction. As we have observed across the different experiments, coded models are capable of better capturing the structure of the data, generating more detailed images and accurate reconstructions.

We observe that, as we decrease the code rate, we obtain better bounds in general. Adding redundancy does not increase the model's flexibility, since the information bits determine the number of latent vectors. However, the introduction of ECCs in the model allows for latent spaces that better capture the structure of the images while employing the same number of latent vectors.

## E.2 RECONSTRUCTION

We first evaluate the model's performance in reconstructing data by examining its *uncoded* and *coded* versions across different configurations, varying the number of information bits and code rates. All the results obtained with MNIST are consistent with those presented in the main text for FMNIST.

In Table 2 we quantify the quality of the reconstructions measuring the PSNR in both training and test sets. In all the cases, coded models yield higher PSNR values, indicating a superior performance in reconstruction. This improvement is also evident through visual inspection of Fig. 17, where the coded models better capture the details in the images. As in FMNIST, we observe a general improvement of the PSNR as we increase the number of information bits and decrease the code rate.

As we discussed in the main text, the PSNR does not account for the *semantic* errors committed by the model. Therefore, we additionally evaluate the reconstruction accuracy, ensuring that the model successfully reconstructs images within the same class as the original ones. We also provide a *confident* reconstruction accuracy, for which we do not count errors when the MAP value of $q(\boldsymbol{m}|\boldsymbol{x})$ is below 0.4. In light of the results, we argue that introducing ECCs in the model allows for latent spaces that better capture the semantics of the images while employing the same number of latent vectors, outperforming the uncoded models in all the cases.

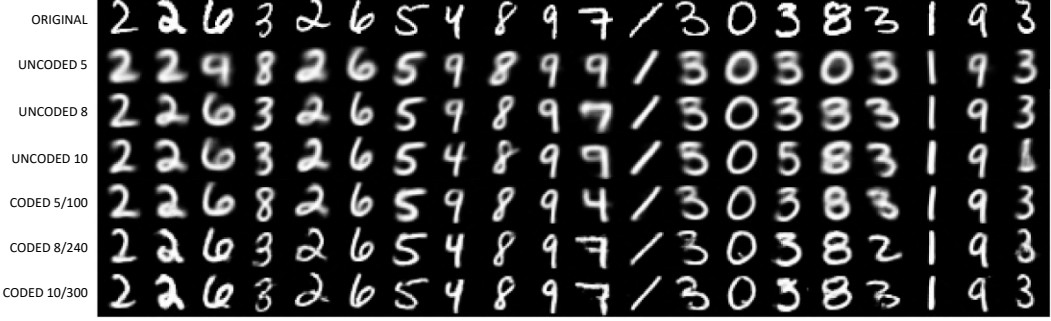

Figure 17: Example of reconstructed test images obtained with different model configurations. Observe that more details are visualized as we increase bits in the latent space and decrease the coding rate.

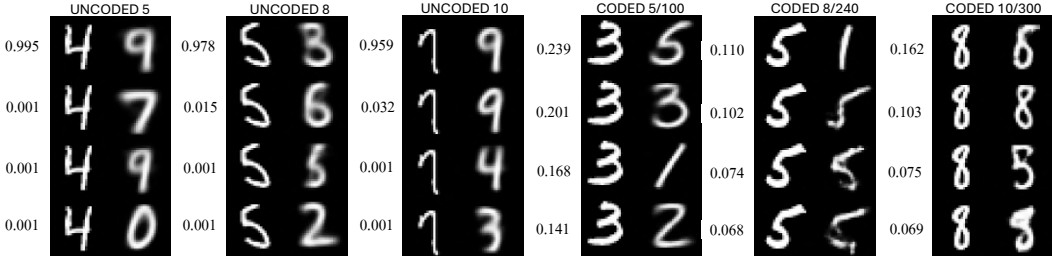

Figure 18: Example of erroneous reconstructions in MNIST using the 4 most-probable words (a posterior). The first column in each image shows the original input to the model, while the second column displays the reconstructions. The a posteriori probability of the word used for reconstruction is indicated in each row.

Table 2: Evaluation of reconstruction performance in MNIST.

| Model | PSNR (train) | Acc (train) | Conf. Acc. (train) | PSNR (test) | Acc (test) | Conf. Acc. (test) | Entropy |
|---|---|---|---|---|---|---|---|
| uncoded 5 | 13.483 | 0.702 | 0.703 | 13.483 | 0.701 | 0.702 | 0.277 |
| coded 5/50 | 14.983 | 0.887 | 0.923 | 14.888 | 0.887 | 0.920 | 2.073 |
| coded 5/80 | 15.436 | 0.899 | 0.936 | 15.263 | 0.895 | 0.929 | 2.237 |
| coded 5/100 | 15.590 | 0.905 | 0.931 | 15.352 | 0.898 | 0.924 | 2.382 |
| uncoded 8 | 14.530 | 0.860 | 0.864 | 14.490 | 0.860 | 0.868 | 0.513 |
| coded 8/80 | 16.878 | 0.937 | 0.964 | 16.042 | 0.912 | 0.947 | 3.105 |
| coded 8/160 | 18.108 | 0.957 | 0.974 | 16.497 | 0.927 | 0.951 | 3.645 |
| coded 8/240 | 19.984 | 0.967 | 0.978 | 16.688 | 0.936 | 0.957 | 3.881 |
| uncoded 10 | 14.879 | 0.888 | 0.891 | 14.816 | 0.887 | 0.890 | 0.636 |
| coded 10/100 | 17.584 | 0.945 | 0.972 | 16.795 | 0.928 | **0.968** | 4.080 |
| coded 10/200 | 20.060 | 0.973 | 0.977 | 16.863 | 0.932 | 0.944 | 4.411 |
| coded 10/300 | **21.083** | **0.979** | **0.984** | **17.114** | **0.941** | 0.945 | 4.810 |

We also report the average entropy of the variational posterior over the test set in Table 2. If we analyze the entropy together with the semantic accuracy, we can argue that coded VAE is aware that multiple vectors might be related to the same image class, and that the posterior shows larger uncertainties for images for which the model has not properly identified the class. We illustrate this argument in Fig. 18, where we show some images selected so that the MAP latent word of $q(\boldsymbol{m}|\boldsymbol{x})$ induces class reconstruction errors. We show the reconstruction of the 4 most probable latent vectors and their corresponding probabilities. Observe that the uncoded model is confident no matter the reconstruction outcome, while in the coded posterior, the uncertainty is much larger.

Table 4 shows the log-likelihood values obtained for the MNIST dataset with various model configurations. Coded models consistently outperform their uncoded counterparts for both the training and test sets, consistent with the findings observed using the FMNIST dataset.

E.3 GENERATION

In this section, we evaluate the model in the image generation task. In Fig. 19, we show examples of randomly generated images using different model configurations in MNIST. These results are consistent with the ones obtained in reconstruction, and with the ones obtained for FMNIST, as we observe that the coded models can generate more detailed and diverse images.

The improved inference provided by the repetition code can also be tested by generating images using the generative model and counting errors using the MAP solution of the variational posterior distribution. Table 3 reports the BER and WER for MNIST. Remarkably, for the same number of latent bits, coded models reduce both the BER and WER w.r.t. the uncoded case. Note also that the

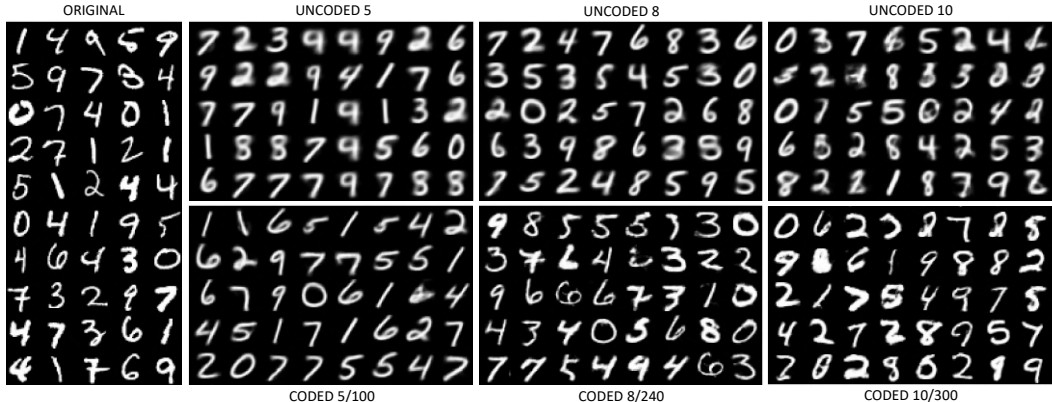

Figure 19: Example of randomly generated, uncurated images using different model configurations.

Table 3: Evaluation of the BER, WER in MNIST.

| Model | BER | WER |
|---|---|---|
| uncoded 5 | 0.002 | 0.008 |
| coded 5/50 | 0.007 | 0.034 |
| coded 5/80 | 0.004 | 0.021 |
| coded 5/100 | 0.009 | 0.045 |
| | | |
| uncoded 8 | 0.015 | 0.071 |
| coded 8/80 | 0.020 | 0.147 |
| coded 8/160 | 0.021 | 0.160 |
| coded 8/240 | 0.023 | 0.167 |
| | | |
| uncoded 10 | 0.057 | 0.373 |
| coded 10/100 | 0.030 | 0.258 |
| coded 10/200 | 0.034 | 0.282 |
| coded 10/300 | 0.041 | 0.331 |

Table 4: Evaluation of the log-likelihood (LL) in MNIST.

| Model | LL (train) | LL (test) |
|---|---|---|
| uncoded 5 | -149.049 | -148.997 |
| coded 5/50 | -117.979 | -119.094 |
| coded 5/80 | -114.911 | -116.639 |
| coded 5/100 | -115.189 | -117.200 |
| | | |
| uncoded 8 | -127.079 | -127.555 |
| coded 8/80 | -96.554 | -104.692 |
| coded 8/160 | -96.014 | -107.436 |
| coded 8/240 | -97.316 | -111.312 |
| | | |
| uncoded 10 | -120.594 | -121.332 |
| coded 10/100 | -92.545 | -99.373 |
| coded 10/200 | -86.072 | -106.249 |
| coded 10/300 | -88.904 | -110.799 |

error rates grow with the number of latent bits, but this is expected due to the increased complexity of the inference process.

# F    CIFAR10 RESULTS

In this section, we provide additional results using the CIFAR10 dataset with different model configurations.

## F.1    TRAINING

We present the evolution of the ELBO and its terms throughout the training process. The models were trained for 300 epochs using Adam optimizer with a learning rate of $10^{-4}$, and a batch size of 128. Fig. 20 displays the results for configurations with 70 information bits, Fig. 21 for 100 information bits, and Fig. 22 for 130 information bits. The colors in all plots represent the various code rates.

Across all the configurations, coded models achieve superior bounds. The main differences in the ELBO come from the different performances in reconstruction. As we have observed across the different experiments, coded models are capable of better capturing the structure of the data, generating more detailed images and accurate reconstructions.

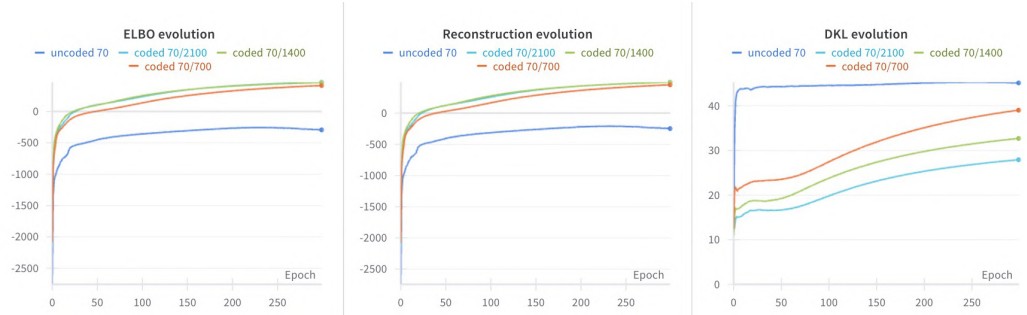

Figure 20: Evolution of the ELBO during training with 70 information bits on CIFAR10.

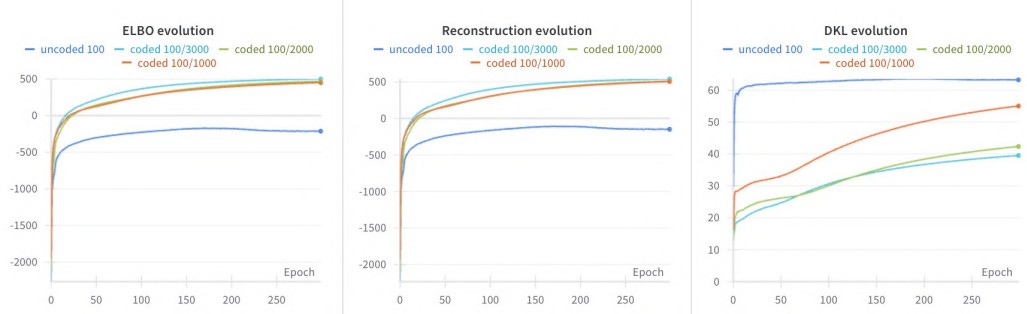

Figure 21: Evolution of the ELBO during training with 100 information bits on CIFAR10.

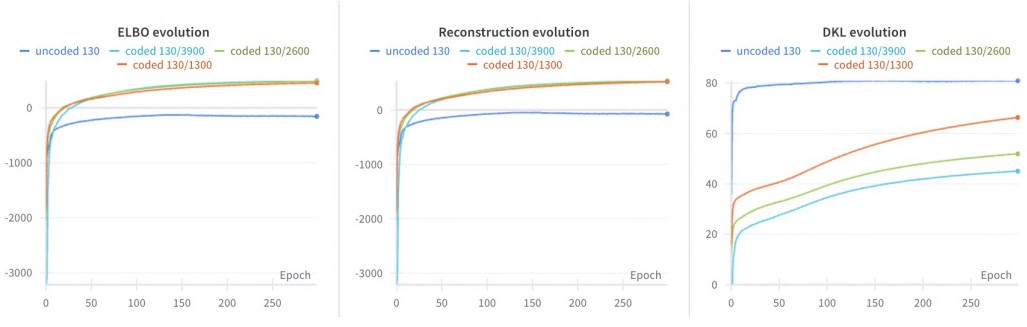

Figure 22: Evolution of the ELBO during training with 130 information bits on CIFAR10.

In the coded case, we do not observe significant differences in the obtained bounds as we increase the number of information bits and reduce the code rate. However, the difference is notable if we compare the coded and uncoded models. Adding redundancy does not increase the model's flexibility, since the information bits determine the number of latent vectors. However, the introduction of ECCs in the model allows for latent spaces that better capture the structure of the images while employing the same number of latent vectors.

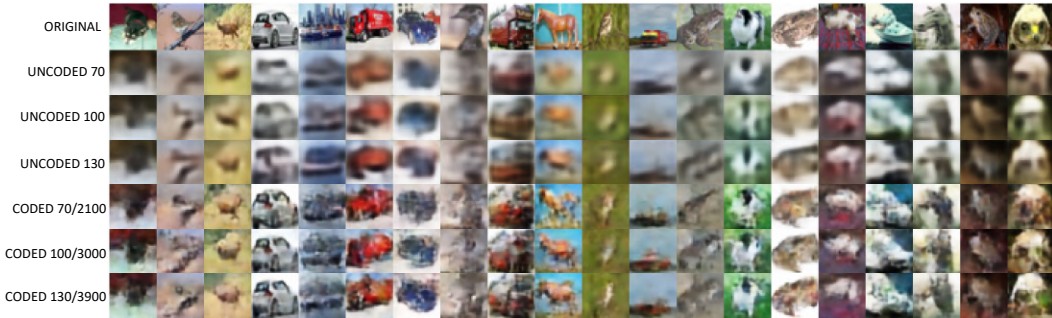

Figure 23: Example of reconstructed test images obtained with different model configurations. Observe that more details are visualized as we increase bits in the latent space and introduce redundancy.

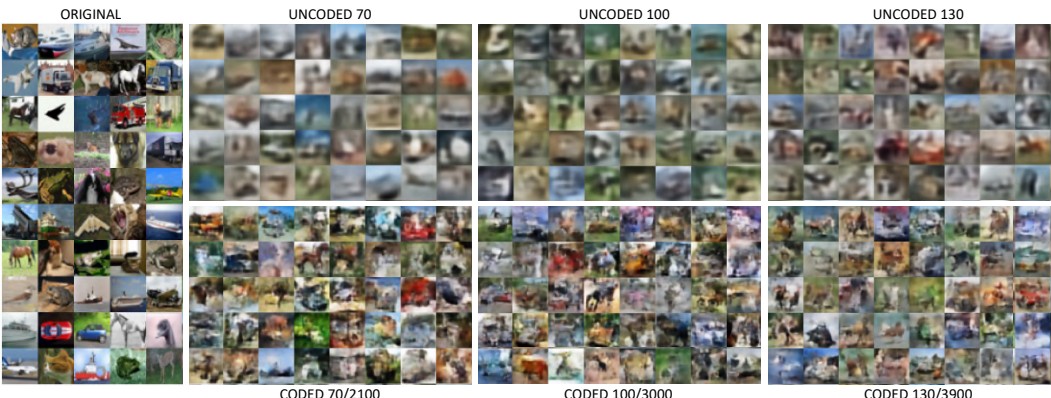

Figure 24: Example of randomly generated, uncurated images using different model configurations.

It's important to note that we are currently using feed-forward networks at the decoder's input. However, this approach may not be suitable for the correlations present in our coded words. Utilizing an architecture capable of effectively leveraging these correlations among the coded bits could potentially enable us to better exploit the introduced redundancy.

## F.2 RECONSTRUCTION AND GENERATION

We first evaluate the model's performance in reconstructing data by examining its *uncoded* and *coded* versions across different configurations, varying the number of information bits and code rates.

In Table 5 we quantify the quality of the reconstructions measuring the PSNR in both training and test sets. Coded models yield higher PSNR values in train, and similar values in test, although the coded models with lower rates outperform the rest of the configurations. However, the improvement in reconstruction is evident through visual inspection of Fig. 23, where the coded models better capture the details in the images. We observe that the coded model yields images that better resemble the structure of the dataset, while the uncoded DVAE cannot decouple spatial information from the images and project it in the latent space.

We hypothesize that to adequately model complex images, transitioning to a hierarchical structure may be necessary. This would allow for the explicit modeling of both global and local information. However, despite employing this rather simple model, we observe that coded configurations outperform their uncoded counterparts in capturing colors and textures.

We also evaluate the model in the image generation task. In Fig. 24, we show examples of randomly generated images using different model configurations in CIFAR10. These results are consistent with the ones obtained in reconstruction, as we observe that the coded models can generate more

Table 5: Evaluation of reconstruction performance in CIFAR10 with different model configurations.

| Model | PSNR (train) | PSNR (test) |
|---|---|---|
| uncoded 70 | 17.985 | 17.596 |
| coded 70/700 | 23.790 | 17.731 |
| coded 70/1400 | 24.555 | 18.008 |
| coded 70/2100 | 25.551 | 18.401 |
| uncoded 100 | 18.509 | 18.334 |
| coded 100/1000 | 24.754 | 18.229 |
| coded 100/2000 | 24.866 | 18.927 |
| coded 100/3000 | 25.646 | 18.920 |
| uncoded 130 | 18.951 | 18.758 |
| coded 130/1300 | 25.007 | 18.887 |
| coded 130/2600 | 25.460 | 19.416 |
| coded 130/3900 | **25.515** | **19.292** |

Table 6: Evaluation of the BER, WER, and FID in CIFAR10 with different model configurations.

| Model | BER | WER | FID |
|---|---|---|---|
| uncoded 70 | 0.162 | 1.000 | 177.524 |
| coded 70/700 | 0.101 | 1.000 | 104.977 |
| coded 70/1400 | 0.088 | 0.999 | 104.078 |
| coded 70/2100 | 0.090 | 0.999 | 102.795 |
| uncoded 100 | 0.182 | 1.000 | 172.063 |
| coded 100/1000 | 0.123 | 1.000 | 107.887 |
| coded 100/2000 | 0.114 | 1.000 | 101.182 |
| coded 100/3000 | 0.138 | 1.000 | 107.287 |
| uncoded 130 | 0.197 | 1.000 | 164.138 |
| coded 130/1300 | 0.144 | 1.000 | 109.905 |
| coded 130/2600 | 0.164 | 1.000 | 110.250 |
| coded 130/3900 | 0.185 | 1.000 | 108.561 |

detailed and diverse images. Additionally, we obtained the Fréchet Inception Distance (FID) score using the test set and 10k generated samples. For this, we used the implementation available at `https://github.com/mseitzer/pytorch-fid`. We can observe that the coded models significantly reduced the FID score in all the cases compared to their uncoded counterparts.For the coded models, we do not observe a clear influence of the code rate on the quality of the generations.

# G    TINY IMAGENET RESULTS

In this section, we provide additional results using the Tiny ImageNet dataset with different model configurations.

## G.1    TRAINING

We present the evolution of the ELBO and its terms throughout the training process. The models were trained for 300 epochs using an Adam optimizer with a learning rate of $10^{-4}$, and a batch size of 128. Fig. 25 displays the results for configurations with 70 information bits, Fig. 26 for 100 information bits, and Fig. 27 for 130 information bits. The colors in all plots represent the various code rates.

As in the rest of the datasets, coded models achieve superior bounds across all the configurations. The main differences in the ELBO come from the different performances in reconstruction. As we have observed across the different experiments, coded models are capable of better capturing the structure of the data, generating more detailed images and accurate reconstructions.

In the coded case, we do not observe significant differences in the obtained bounds as we increase the number of information bits and reduce the code rate. However, the difference is notable if we compare the coded and uncoded models. Adding redundancy does not increase the model's flexibility, since the information bits determine the number of latent vectors. However, the introduction of ECCs in the model allows for latent spaces that better capture the structure of the images while employing the same number of latent vectors.

It's important to note that we are currently using feed-forward networks at the decoder's input. However, this approach may not be suitable for the correlations present in our coded words. Utilizing an architecture capable of effectively leveraging these correlations among the coded bits could potentially enable us to better exploit the introduced redundancy.

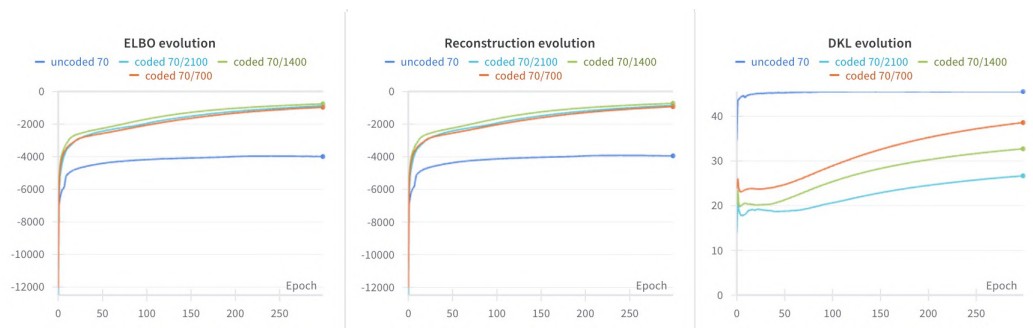

Figure 25: Evolution of the ELBO during training for the configurations with 70 information bits.

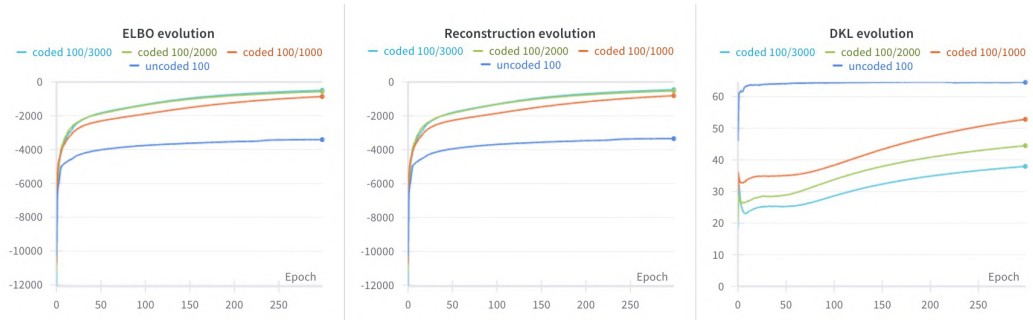

Figure 26: Evolution of the ELBO during training for the configurations with 100 information bits.

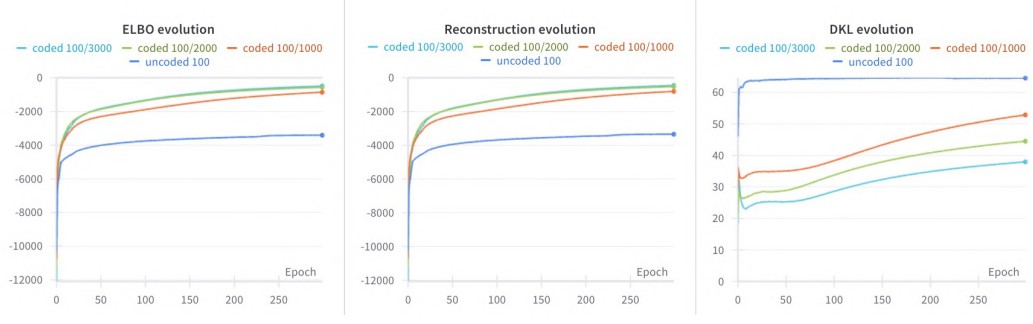

Figure 27: Evolution of the ELBO during training for the configurations with 130 information bits.

## G.2 RECONSTRUCTION AND GENERATION

We first evaluate the model's performance in reconstructing data by examining its *uncoded* and *coded* versions across different configurations, varying the number of information bits and code rates.

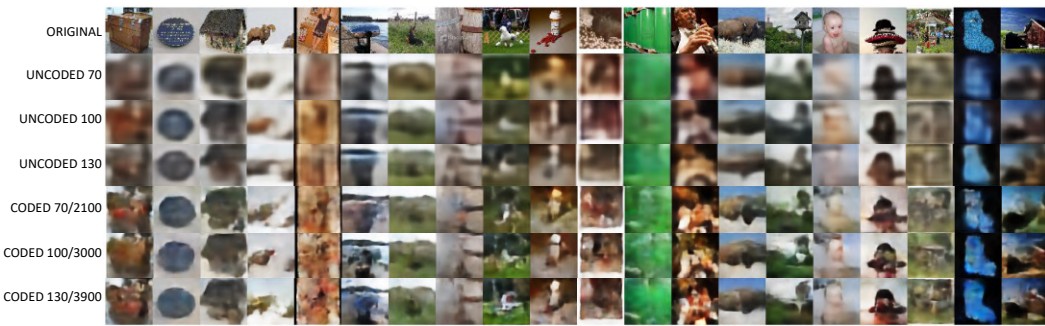

Figure 28: Example of reconstructed test images obtained with different model configurations. Observe that more details are visualized as we increase the bits in the latent space and introduce redundancy.

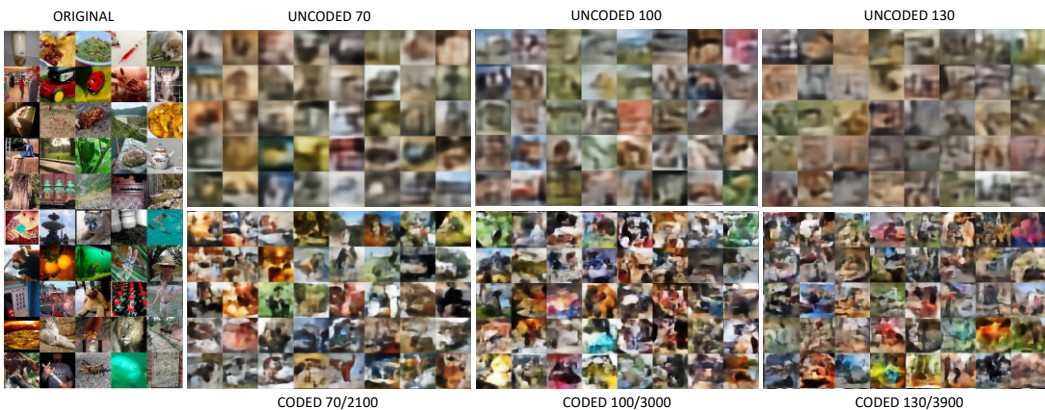

Figure 29: Example of randomly generated, uncurated images using different model configurations.

In Table 7 we quantify the quality of the reconstructions measuring the PSNR in both training and test sets. Coded models yield higher PSNR values in train, and similar values in test, although the coded models with lower rates outperform the rest of the configurations. However, the improvement in reconstruction is evident through visual inspection of Fig. 28, where the coded models better capture the details in the images. We observe that the coded model yields images that better resemble the structure of the dataset, while the uncoded DVAE cannot decouple spatial information from the images and project it in the latent space.

We hypothesize that to adequately model complex images, transitioning to a hierarchical structure may be necessary. This would allow for the explicit modeling of both global and local information. However, despite employing this rather simple model, we observe that coded configurations outperform their uncoded counterparts in capturing colors and textures.

We also evaluate the model in the image generation task. In Fig. 29, we show examples of randomly generated images using different model configurations in Tiny ImageNet. These results are consistent with the ones obtained in reconstruction, as we observe that the coded models can generate more detailed and diverse images. Additionally, we obtained the FID score using the test set and 10k generated samples. For this, we used the implementation available at https://github.com/mseitzer/pytorch-fid. We can observe that the coded models significantly reduced the FID score in all the cases compared to their uncoded counterparts.

For the coded models, we do not observe a clear influence of the code rate on the quality of the generations in CIFAR-10. However, in Tiny ImageNet, smaller code rates produce worse FID scores. We hypothesize that this may be due to the presence of artifacts in the generated images. Our experiments indicate that coded models with lower rates attempt to model fine details in images, which can lead to artifacts in generation.

Table 7: Evaluation of reconstruction performance in Tiny ImageNet with different model configurations.

| Model | PSNR (train) | PSNR (test) |
|---|---|---|
| uncoded 70 | 15.598 | 15.402 |
| coded 70/700 | 18.156 | 15.158 |
| coded 70/1400 | 18.396 | 15.419 |
| coded 70/2100 | 18.228 | 15.789 |
| uncoded 100 | 16.012 | 15.774 |
| coded 100/1000 | 18.298 | 15.677 |
| coded 100/2000 | 18.647 | 15.892 |
| coded 100/3000 | 18.729 | 16.167 |
| uncoded 130 | 16.278 | 16.009 |
| coded 130/1300 | 18.719 | 15.901 |
| coded 130/2600 | 18.818 | **16.329** |
| coded 130/3900 | **19.020** | 16.288 |

Table 8: Evaluation of the BER, WER, and FID in Tiny ImageNet with different model configurations.

| Model | BER | WER | FID |
|---|---|---|---|
| uncoded 70 | 0.143 | 1.000 | 265.474 |
| coded 70/700 | 0.096 | 0.998 | 171.993 |
| coded 70/1400 | 0.104 | 1.000 | 170.496 |
| coded 70/2100 | 0.096 | 0.998 | 176.245 |
| uncoded 100 | 0.164 | 1.000 | 234.358 |
| coded 100/1000 | 0.099 | 1.000 | 153.743 |
| coded 100/2000 | 0.097 | 1.000 | 162.889 |
| coded 100/3000 | 0.098 | 1.000 | 163.049 |
| uncoded 130 | 0.200 | 1.000 | 219.003 |
| coded 130/1300 | 0.129 | 1.000 | 165.064 |
| coded 130/2600 | 0.114 | 1.000 | 164.759 |
| coded 130/3900 | 0.128 | 1.000 | 170.603 |

# H IWAE RESULTS

One could draw a parallel between the coded DVAE with repetition codes and the well-known IWAE (Burda et al., 2016), but the two approaches are fundamentally different. In the IWAE, independent samples are drawn from the variational posterior and propagated independently through the generative model to obtain a tighter variational bound on the marginal log-likelihood. In our method, we jointly propagate the output of the ECC encoder through the generative model, obtaining a single prediction and exploiting the introduced known correlations in the variational approximation of the posterior. In the case of repetition codes, the ECC encoder outputs are repeated bits, or repeated probabilities in the case of soft encoding. However, our approach extends beyond repetition codes, opening a new field for improved inference in discrete latent variable models.

In this work, we specifically utilize the redundancy introduced by the repetition code to correct potential errors made by the encoder through a soft decoding approach, leading to a more accurate approximation of $p(\boldsymbol{m}|\boldsymbol{x})$ and an improved proposal for sampling. The results obtained in the coded DVAE case cannot be achieved by training the uncoded DVAE with the IWAE objective. The following results compare the uncoded IWAE model with the coded DVAE trained on FMNIST. While the uncoded model shows slight performance gains with an increasing number of IWAE samples (which improves the evidence lower bound), it still underperforms compared to the coded model. Furthermore, when using 20 and 30 IWAE samples, the metrics slightly declined compared to using 10 samples, likely due to overfitting, as we applied a common early stopping point.

Table 9: Comparison of the metrics obtained with our method and the uncoded DVAE trained with the IWAE objective.

| Model | BER | WER | Entropy | Acc. | Conf. Acc. | PSNR |
|---|---|---|---|---|---|---|
| uncoded 8 | 0.089 | 0.384 | 0.467 | 0.594 | 0.595 | 15.598 |
| uncoded 8 IWAE 10 samples | 0.063 | 0.372 | 1.309 | 0.617 | 0.640 | 14.282 |
| uncoded 8 IWAE 20 samples | 0.075 | 0.447 | 1.391 | 0.634 | 0.651 | 14.237 |
| uncoded 8 IWAE 30 samples | 0.074 | 0.438 | 1.564 | 0.619 | 0.641 | 13.757 |
| coded 8/80 | 0.021 | 0.144 | 2.905 | 0.750 | 0.816 | 17.318 |
| coded 8/160 | 0.027 | 0.189 | 3.637 | 0.783 | 0.831 | 17.713 |
| coded 8/240 | 0.037 | 0.231 | 4.000 | 0.799 | 0.893 | 17.861 |

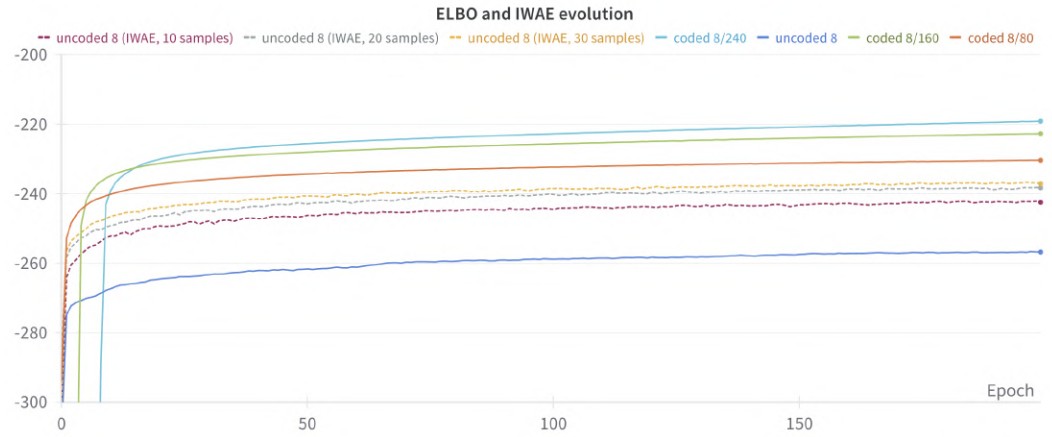

Figure 30: Evolution of the ELBO and the IWAE objectives for various configurations. Observe that the IWAE provides a tighter bound than the ELBO in the uncoded setting. However, coded models obtain even better bounds using the same number of samples/repetitions.

# I  HIERARCHICAL CODED DVAE RESULTS

Inspired by polar codes (Arikan, 2009), we present a hierarchical coded DVAE with two layers of latent bits, as illustrated in Figure 31. In this model, the latent bits $m_1$ are encoded using a repetition code in the first layer, producing $c_1$ and $z_1$. Concurrently, the bits in the second layer, $m_2$, are linearly combined with $m_1$ following $m_{1,2} = m_1 \oplus m_2$, considering a binary field or Galois field. The resulting vector is then encoded with another repetition code to produce $c_2$, which is subsequently modulated into $z_2$. Finally, both $z_1$ and $z_2$ are concatenated and passed through the decoder network to generate $x$. The model provides stronger protection for $m_1$, as it appears in both branches of the generative model. Inference follows a similar approach to the one employed in the coded DVAE, incorporating the linear combination of $m_1$ and $m_2$ used in the second branch.

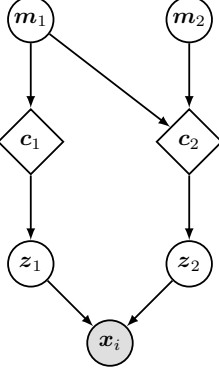

Figure 31: Graphical model of the hierarchical Coded VAE with two layers.

We adopt the same variational family as used in the standard Coded VAE; however, in this case, we incorporate both hierarchical levels, leading to

$$q_{\boldsymbol{\eta}}(\boldsymbol{m}, \boldsymbol{z}|\boldsymbol{x}) = q_{\boldsymbol{\eta}}(\boldsymbol{m}_1|\boldsymbol{x})q_{\boldsymbol{\eta}}(\boldsymbol{m}_2|\boldsymbol{x})p(\boldsymbol{z}_1|\boldsymbol{c}_1)p(\boldsymbol{z}_2|\boldsymbol{c}_2), \qquad (12)$$

where $q_{\boldsymbol{\eta}}(\boldsymbol{m}_1|\boldsymbol{x})$ is calculated following the same approach as in the coded DVAE with repetition codes, computing the all-are-zero and all-are-ones products of probabilities of the bits in $c_1$ that are copies of the same message bit. The posterior $q_{\boldsymbol{\eta}}(\boldsymbol{m}_2|\boldsymbol{x})$ considering both the encoder's output and the inferred posterior distribution $q_{\boldsymbol{\eta}}(\boldsymbol{m}_1|\boldsymbol{x})$. Note that in this case, the decoder outputs the

probabilities for both $c_1$ and $c_2$, with $c_2$ being the encoded version of the linear combination $m_{1,2} = m_1 \oplus m_2$. Consequently, we first obtain $q_{\eta}(m_{1,2}|x)$ following the same approach as in the coded DVAE with repetition codes, and determine $q_{\eta}(m_2|x)$ as

$$q_{\eta}(m_2|x) = \prod_{u=1}^{M} \text{Ber}(p_u), \tag{13}$$

$$p_u = q_{\eta}(m_{1,2,u} = 1|x)q_{\eta}(m_{1,u} = 0|x) + q_{\eta}(m_{1,2,u} = 0|x)q_{\eta}(m_{1,u} = 1|x). \tag{14}$$

After obtaining $q_{\eta}(m_1|x)$ and $q_{\eta}(m_2|x)$, we recalculate the posterior bit probabilities for the linear combination $q'_{\eta}(m_{1,2}|x)$ as

$$q'_{\eta}(m_{1,2}|x) = \prod_{u=1}^{M} \text{Ber}(q_u), \tag{15}$$

$$q_u = q_{\eta}(m_{1,u} = 1|x)q_{\eta}(m_{2,u} = 0|x) + q_{\eta}(m_{1,u} = 0|x)q_{\eta}(m_{2,u} = 1|x). \tag{16}$$

Next, we apply the soft encoding approach to incorporate the repetition code structure at both levels of the hierarchy. The posterior probabilities $q_{\eta}(m_1|x)$ are repeated to obtain $q_{\eta}(c_1|x)$, and the posterior probabilities $q_{\eta}(m_{1,2}|x)$ are repeated to produce $q_{\eta}(c_2|x)$. Utilizing the reparameterization trick from Eq. 5, we sample $z_1$ and $z_2$, concatenate them to form $z$, and pass this through the decoder to generate $p_{\theta}(x|z)$. The model is trained by maximizing the ELBO, given by

$$\text{ELBO} = \mathbb{E}_{q_{\eta}(m,z|x)} \log p_{\theta}(x|z) - \mathcal{D}_{KL}\big(q_{\eta}(m_1|x)||p(m_1)\big) - \mathcal{D}_{KL}\big(q_{\eta}(m_2|x)||p(m_2)\big), \tag{17}$$

where both $p(m_1)$ and $p(m_2)$ are assumed to be independent Bernoulli distributions with bit probabilities of $0.5$, consistent with the other scenarios.

We obtained results on the FMNIST dataset using a model with 5 information bits per branch and repetition rates of $R = 1/10$ and $R = 1/20$. In this case, we applied the same code rate to both branches, although varying code rates could be used to control the level of protection at each hierarchy level. Tables 10 and 11 present the metrics obtained for the different configurations. Specifically, Table 10 shows the overall metrics obtained with this structure, and Table 11 compares the error metrics across the two hierarchy levels. As expected, $m_2$ shows poorer error metrics compared to $m_1$, since the model provides more redundancy to $m_1$ incorporating it in both branches. Although the overall metrics and generation quality are somewhat similar to those of the coded DVAE with 10 information bits (see tables in Figures 2 and 4), the introduced hierarchy results in a more interpretable latent space. In this setup, $m_1$ captures global features (such as clothing types in the FMNIST dataset), while $m_2$ controls individual features, as we can observe in Figure 32, where we show examples of the model's generative outputs for fixed $m_1$ and random samples of $m_2$.

Table 10: Comparison of the obtained metrics for the coded DVAE with polar codes with different configurations, which we refer to as 'hierarchical coded DVAE'.

| Model | BER | WER | Acc | Conf. Acc | PSNR |
|---|---|---|---|---|---|
| hier. 5/50 | 0.099 | 0.400 | 0.753 | 0.800 | 17.130 |
| hier. 5/100 | 0.050 | 0.330 | 0.784 | 0.870 | 17.513 |

Table 11: Comparison of the obtained error metrics in the different hierarchy levels.

| Model | BER $m_1$ | WER $m_1$ | BER $m_2$ | WER $m_2$ |
|---|---|---|---|---|
| hier. 5/50 | 0.079 | 0.259 | 0.119 | 0.362 |
| hier. 5/100 | 0.026 | 0.110 | 0.075 | 0.287 |

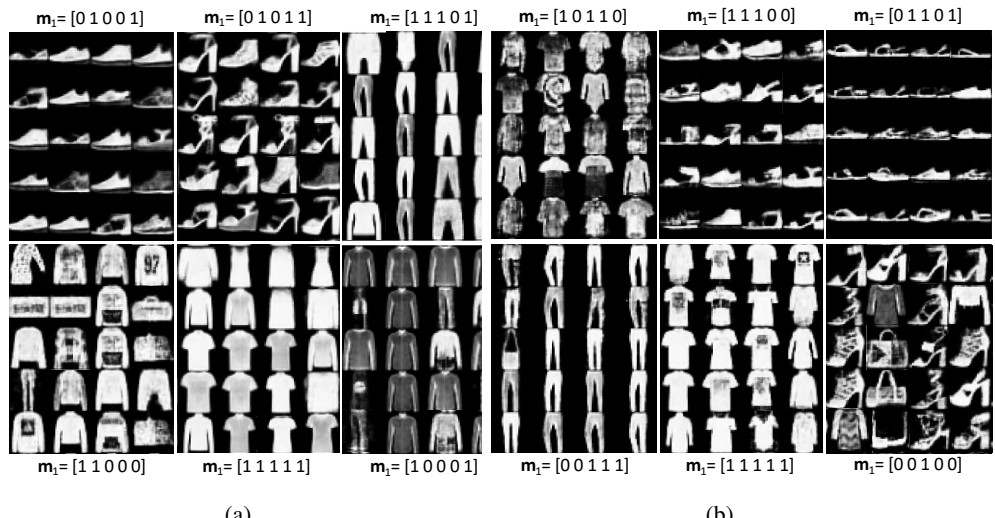

(a)    (b)

Figure 32: Examples of generated images using the hierarchical coded DVAE with (a) a 5/50 repetition code in each branch, and (b) a 5/100 repetition code in each branch. In all the examples provided, $m_1$ was fixed while $m_2$ was randomly sampled.

## J    ABLATION STUDY

In this section, we conduct ablation studies on the hyperparameter $\beta$ of the model, responsible for regulating the decay of exponentials in the smoothing transformation, as well as on the number of trainable parameters in the models.

### J.1    ABLATION STUDY ON THE HYPERPARAMETER $\beta$

Across all experiments, we have consistently configured the hyperparameter $\beta$, which controls the decay of exponentials in the smoothing transformation, to a value of 15. To illustrate its impact on the overall performance of the model, we conducted an ablation study on the value of this hyperparameter for both uncoded and coded cases.

The smoothing distribution employed for the reparameterization trick consists of two overlapping exponentials. The hyperparameter $\beta$ functions as a temperature term, regulating the decay of the distributions and, consequently, influencing the degree of overlapping. A lower $\beta$ value results in more overlapped tails, while a higher value leads to less overlapped distributions. A priori, we would like these distributions to be separated, allowing us to retrieve the true value of the bit and effectively use the latent structure of the model.

### J.1.1    CODED MODEL

We first evaluate the influence of the parameter $\beta$ in coded models. We take as a reference the coded model with 8 information bits and a rate $R = 1/30$, and train it using $\beta = 5, 10, 15, 20$. We assess the performance of the model in reconstruction and generation tasks. We observe the model is fairly robust, achieving similar performance across configurations in most metrics.

In Fig. 33 we show examples of reconstructed images using the different configurations to assess reconstruction through visual examination, and Table 12 contains the associated reconstruction metrics. All the configurations achieve similar performances, although the models trained with $\beta = 10$ and $\beta = 15$ seem to be the best configurations for this scenario. Larger values may result in unstable training and inferior performance.

Next, we evaluate the model in the image generation task. Fig. 34 contains examples of randomly generated images using the different configurations. Table 13 reports the obtained BER and WER, and Table 14 the estimated log-likelihood of the different values of $\beta$. The model trained with

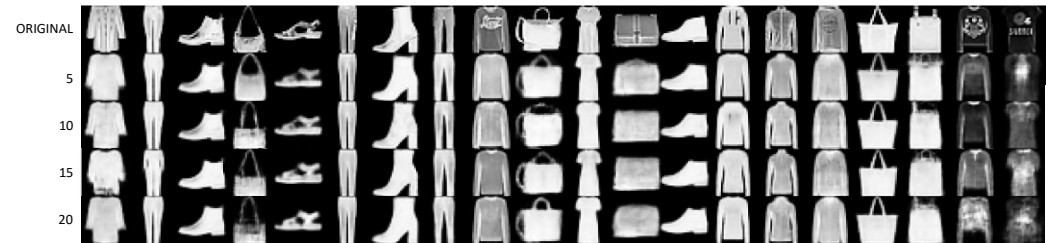

Figure 33: Example of reconstructed images obtained with different values of $\beta$ using the coded model with an 8/240 code.

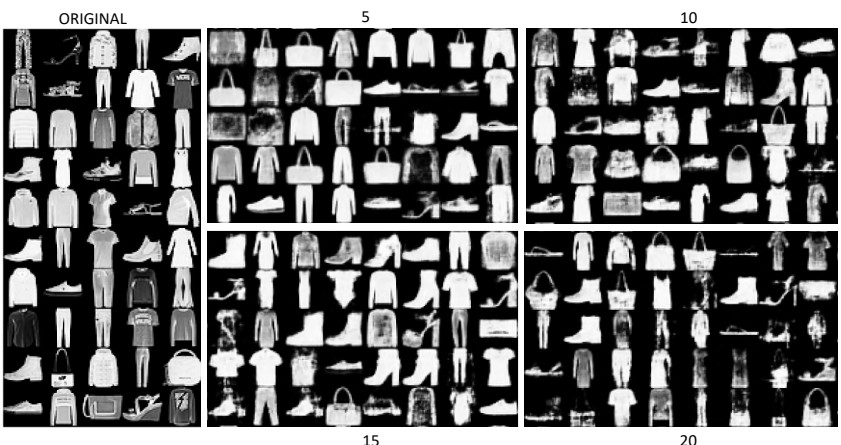

Figure 34: Example of randomly generated, uncurated images with different values of $\beta$ using the coded model with an 8/240 code.

Table 12: Evaluation of reconstruction performance in FMNIST with different values of $\beta$ using the coded model with an 8/240 code.

| Model | PSNR (train) | Acc (train) | Conf. Acc. (train) | PSNR (test) | Acc (test) | Conf. Acc. (test) | Entropy |
|---|---|---|---|---|---|---|---|
| 5 | 18.344 | 0.791 | 0.895 | 17.614 | 0.766 | 0.849 | 4.025 |
| 10 | 19.106 | 0.822 | 0.904 | 17.737 | 0.793 | 0.872 | 4.023 |
| 15 | 19.345 | 0.831 | 0.921 | 17.861 | 0.799 | 0.893 | 4.000 |
| 20 | 18.797 | 0.809 | 0.887 | 17.837 | 0.787 | 0.877 | 3.810 |

Table 13: Evaluation of the BER and WER in FMNIST with different values of $\beta$ using the coded model with an 8/240 code.

| Beta | BER | WER |
|---|---|---|
| 5 | 0.150 | 0.726 |
| 10 | 0.080 | 0.480 |
| 15 | 0.037 | 0.231 |
| 20 | 0.065 | 0.399 |

Table 14: Evaluation of the log-likelihood (LL) in FMNIST with different values of $\beta$ using the coded model with an 8/240 code.

| Beta | LL (train) | LL (test) |
|---|---|---|
| 5 | -228.448 | -234.629 |
| 10 | -229.379 | -237.495 |
| 15 | -231.679 | -238.459 |
| 20 | -229.627 | -235.927 |

$\beta = 15$ stands out in terms of error metrics, although achieves similar log-likelihood values as the model trained with $\beta = 10$. Again, these two configurations appear to be the most suitable in this scenario.

### J.1.2 UNCODED MODEL

We first evaluate the influence of the parameter $\beta$ in uncoded models. We take as a reference the coded model with 8 information bits and train it using $\beta = 5, 10, 15, 20$. We assess the performance of the model in reconstruction and generation tasks. We observe that the uncoded model is also robust, achieving similar performance across configurations.

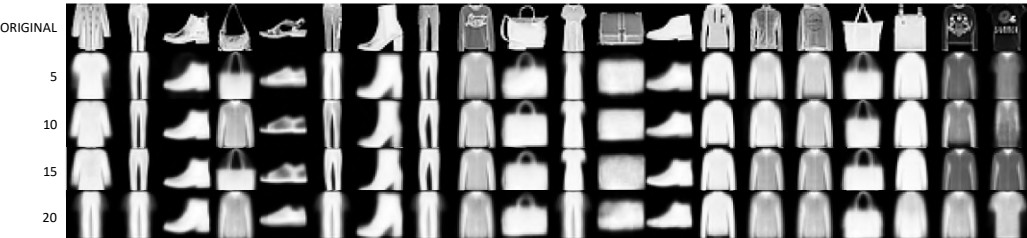

Figure 35: Example of reconstructed images obtained with different values of $\beta$ using an uncoded model 8 information bits.

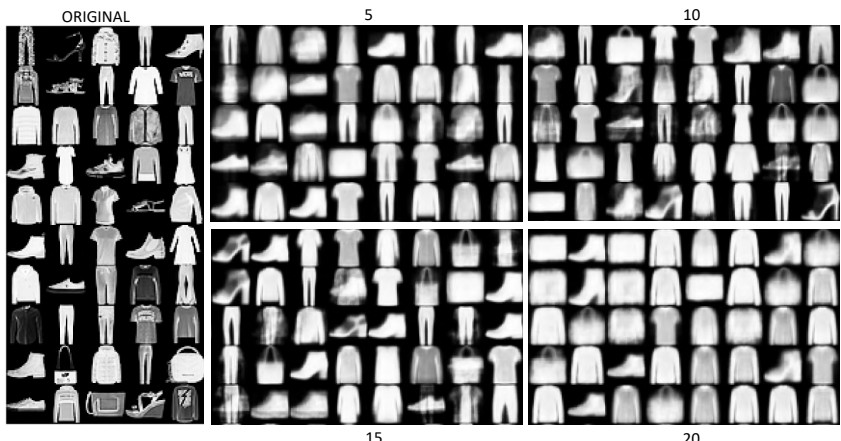

Figure 36: Example of randomly generated, uncurated images with different values of $\beta$ using an uncoded model 8 information bits.

Table 15: Evaluation of reconstruction performance in FMNIST with different values of $\beta$ using an uncoded model 8 information bits.

| Model | PSNR (train) | Acc (train) | Conf. Acc. (train) | PSNR (test) | Acc (test) | Conf. Acc. (test) | Entropy |
|---|---|---|---|---|---|---|---|
| 5 | 14.239 | 0.503 | 0.501 | 14.237 | 0.503 | 0.491 | 0.231 |
| 10 | 15.624 | 0.606 | 0.603 | 15.571 | 0.598 | 0.598 | 0.357 |
| 15 | 15.644 | 0.601 | 0.602 | 15.598 | 0.594 | 0.595 | 0.467 |
| 20 | 13.717 | 0.464 | 0.466 | 13.743 | 0.460 | 0.462 | 0.383 |

In Fig. 35 we show examples of reconstructed images using the different configurations to assess reconstruction through visual examination, and Table 15 contains the associated reconstruction metrics. All the configurations achieve similar performances, although the models trained with $\beta = 10$ and $\beta = 15$ seem to be the best configurations for this scenario. Larger values may result in unstable training and inferior performance, as we can clearly observe in this case.

Next, we evaluate the model in the image generation task. Fig. 36 contains examples of randomly generated images using the different configurations. Table 16 reports the obtained BER and WER, and Table 17 the estimated log-likelihood of the different values of $\beta$. The models trained with

Table 16: Evaluation of the BER and WER in FMNIST with different values of $\beta$ using an uncoded model 8 information bits.

| Beta | BER | WER |
|------|-------|-------|
| 5 | 0.203 | 0.852 |
| 10 | 0.086 | 0.384 |
| 15 | 0.089 | 0.384 |
| 20 | 0.278 | 0.939 |

Table 17: Evaluation of the log-likelihood (LL) in FMNIST with different values of $\beta$ using an uncoded model 8 information bits.

| Model | LL (train) | LL (test) |
|-------|------------|-----------|
| 5 | -256.431 | -257.983 |
| 10 | -247.507 | -249.460 |
| 15 | -247.964 | -249.880 |
| 20 | -272.460 | -273.554 |

$\beta = 15$ and $\beta = 10$ clearly outperform the other two in this task, generating more diverse and detailed images, and obtaining better error metrics and log-likelihood values.

### J.2 ABLATION STUDY ON THE NUMBER OF TRAINABLE PARAMETERS

A consistent architecture was employed across all experiments, which is detailed in Section C. However, since the introduction of the code alters the dimensionality of the latent space, it is necessary to adjust the encoder's output and the decoder's input. This results in an augmentation of the trainable parameters in the coded cases compared to their uncoded counterparts.

Given that a higher number of parameters usually results in better performance, we conducted an ablation study on the model's trainable parameters to confirm that the improved performance introduced by the coded models is not due to this factor. We adjusted the hidden dimensions of the encoder and decoder architectures to ensure both configurations (coded and uncoded) have roughly the same number of trainable parameters. We have conducted the ablation study using the uncoded model with 8 bits and the coded 8/240 model trained on FMNIST.

We adjusted the encoder's last hidden dimension and the decoder's first hidden dimension to equalize the parameter count between the uncoded and coded models. This adjustment was straightforward since the last layers of the encoder and the first layers of the decoder are feed-forward layers. We kept the latent dimension of the model unchanged, ensuring that the modification solely pertained to the neural network architecture.

Table 18: Parameter count.

| Model | # encoder parameters | # decoder parameters |
|-------|----------------------|----------------------|
| uncoded 8 | 6,592,008 | 19,341,185 |
| uncoded 8 adjusted | 6,717,538 | 19,581,035 |
| coded 8/240 | 6,711,024 | 19,578,753 |
| coded 8/240 adjusted | 6,583,174 | 19,332,871 |

#### J.2.1 EVALUATION

This section provides an empirical evaluation of the models trained with the adjusted parameter count, demonstrating that the enhanced performance observed in coded models does not result from an augmented number of trainable parameters. We found that the performance of the original and adjusted models is very similar, meaning that the conclusions drawn in the main text hold even in this scenario.

We first evaluate the reconstruction performance, measuring the PSNR and reconstruction accuracy in both train and test sets, which are included in Table 19. Then, we assess generation measuring the BER and WER, reported in Table 20. Finally, we compute the log-likelihood for train and test sets, shown in Table 21.

In terms of reconstruction, both the original and adjusted models exhibit very similar performance, observed in both reconstruction quality and accuracy. However, the adjusted models show slightly inferior results than the original ones. For the coded model, this might be attributed to reduced flexibility when decreasing the number of parameters. As for the uncoded model, the increased

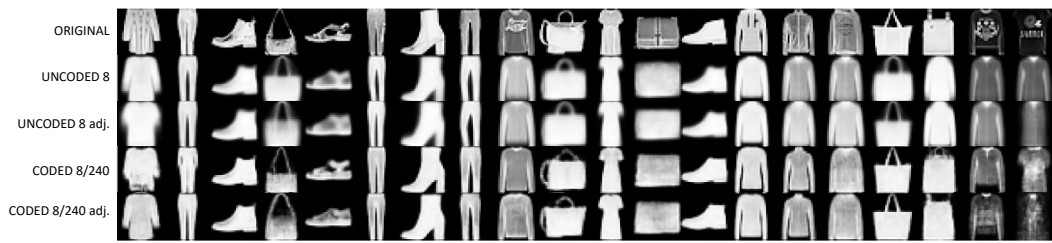

Figure 37: Example of reconstructed images obtained with different configurations.

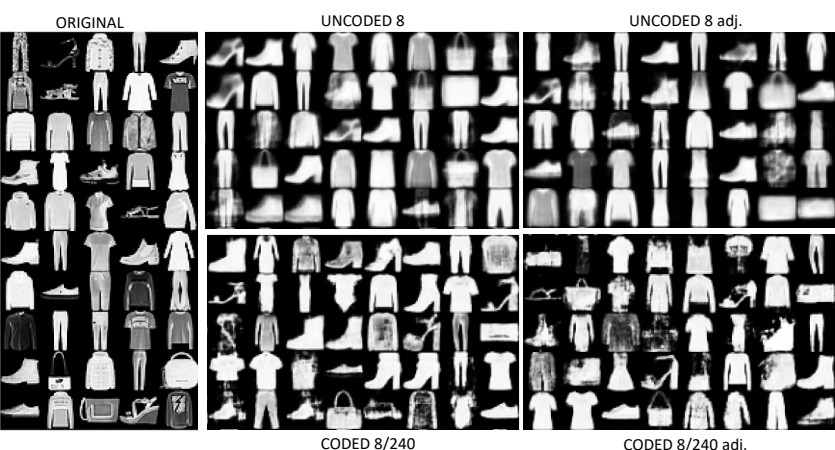

Figure 38: Example of randomly generated, uncurated images using different model configurations.

Table 19: Evaluation of reconstruction performance in FMNIST with the adjusted parameter count.

| Model | PSNR (train) | Acc (train) | Conf. Acc. (train) | PSNR (test) | Acc (test) | Conf. Acc. (test) | Entropy |
|---|---|---|---|---|---|---|---|
| uncoded 8 | 15.644 | 0.601 | 0.602 | 15.598 | 0.594 | 0.595 | 0.659 |
| uncoded 8 adj. | 15.530 | 0.586 | 0.586 | 15.491 | 0.581 | 0.580 | 0.449 |
| coded 8/240 | 19.345 | 0.831 | 0.921 | 17.861 | 0.799 | 0.893 | 4.609 |
| coded 8/240 adj. | 19.383 | 0.828 | 0.883 | 17.771 | 0.792 | 0.828 | 3.952 |

Table 20: Evaluation of the BER and WER in FMNIST with the adjusted parameter count.

| Model | BER | WER |
|---|---|---|
| uncoded 8 | 0.089 | 0.384 |
| uncoded 8 adj. | 0.125 | 0.561 |
| coded 8/240 | 0.037 | 0.231 |
| coded 8/240 adj. | 0.064 | 0.399 |

Table 21: Evaluation of the log-likelihood (LL) in FMNIST with the adjusted parameter count.

| Model | LL (train) | LL (test) |
|---|---|---|
| uncoded 8 | -247.964 | -249.880 |
| uncoded 8 adj. | -250.543 | -252.408 |
| coded 8/240 | -231.679 | -238.459 |
| coded 8/240 adj. | -229.283 | -238.302 |

complexity while maintaining the same low-dimensional latent vector may not provide enough expressiveness to leverage the added flexibility in the architecture, potentially causing the observed decrease in performance. We observe the same behavior when we evaluate the BER and WER.

Analyzing the results, especially the log-likelihood values shown in Table 21, we can argue that increasing the flexibility in the architecture does not necessarily lead to improved performance in this scenario. The coded models exhibit similar performance with both the original and adjusted parameter counts, consistently outperforming the uncoded models. These results indicate that the performance enhancement is attributed to the introduction of ECCs in the latent space, rather than differences in the architecture required to handle the introduced redundancy.

## K    EVALUATING LOG-LIKELIHOOD USING THE SOFT-ENCODING MODEL

The reparameterization trick introduced in equation 5 requires that bits are independent, a condition not met in $c$ once the code's structure is introduced. To address this issue, during training, we employ a *soft encoding strategy*. We assume the bits in $c$ are independent and equally distributed according to $q_{\eta}(m|x)$. Therefore, instead of directly repeating sampled bits in $m$ to obtain $c$ following $c = m^T G$, we repeat the posterior probabilities for the copies of the same bit and sample $z$ using the reparameterization trick in equation 5. Remark that, despite the soft encoding assumption during training, the generative results in Fig. 4, 6, 13, 19, 24, 24, and 29 are obtained through *hard encoding*. Namely, we sampled an information word $m$ and obtained $c$ by repeating its bits.

While the hard-encoding images are visually appealing, to evaluate the coded DVAE LL for a given image $x$, we have to leverage the soft-encoding model since it is unrealistic for a sample from the soft-encoding model to produce equally repeated bits. In the soft-coded model, we sample bit probabilities from a prior distribution that we model through the product of $M$ independent Beta distributions, and we use a proposal distribution model similar to the Vamp-prior in Tomczak & Welling (2018). Namely, a mixture model with components given by $q(m|x)$ for different training points.

In the main text and Section J, we report the log-likelihood values for different model configurations trained on the FMNIST dataset. Table 4 presents the results obtained for the MNIST dataset. Due to their simplicity, these datasets do not require high-dimensional latent spaces, and competitive results can be achieved with just 8 or 10 information bits. However, for more complex datasets like CIFAR10 or Tiny ImageNet, high-dimensional latent spaces are necessary to capture spatial information, colors, and textures. For these high-dimensional datasets, we were unable to obtain valid log-likelihood estimates because the number of samples needed for importance sampling to converge was too large. However, given the difference in the level of detail in the reconstructed and generated images between coded and uncoded models (see Sections F and G), coded models are expected to better approximate the true marginal likelihood of the data.

## L    CONNECTION TO PREVIOUS WORK ON VAEs AS SOURCE CODING METHODS

Effective learning of low-dimensional discrete latent representations is a technically challenging problem. In this work, we propose a novel method to improve inference in discrete VAEs within a fully probabilistic framework, introducing a new perspective on the inference problem. While VAEs have been analyzed in the literature using rate-distortion (RD) theory (Chen et al., 2022; Townsend et al., 2019; Van Den Oord et al., 2017), our approach stems from a different perspective, that is indeed compatible with all those works.

While VAEs are often viewed as lossy compression models (e.g. VAEs as source coding methods), our contribution is best understood from a generative perspective. We conceptualize the process of inference via as decoding the latent variable from the observed data. We sample a vector $m$, generate an image $x$, and seek to minimize the error rate in recovering $m$ from $x$. Achieving this requires the variational approximation $q(m|x)$ to closely align with the true posterior. Estimating $m$ from $x$ thus involves approximating the true posterior $p(m|x)$ with $q(m|x)$. We show that introducing redundancy in generation reduces error rates in estimating $m$, leading to better approximations of the latent posterior distribution. This improvement is reflected in the enhanced performance of our model, as demonstrated in our experimental results.

VQ-VAEs (Van Den Oord et al., 2017) are notable for effectively learning compressed discrete representations of data, and we believe it can be beneficial to highlight key differences between VQ-VAEs and our approach. A primary distinction lies in the latent space's structure and dimensionality. In image modeling, VQ-VAEs typically employ a latent matrix where indices correspond to codewords in a codebook. This design allows different codewords to capture specific patches of the original image, improving reconstruction and generation (the latter through an autoregressive prior on the latent representations). However, this matrix representation complicates interpretability, as each data point is represented by a grid of embeddings. In contrast, our method encodes the entire image into the latent space, rather than splitting it into patches. Another important difference is that

VQ-VAEs is based on a non-probabilistic encoder, where the output is mapped to the nearest latent code based on a distance metric. This deterministic mapping limits the model's ability to quantify uncertainty in the latent space. In contrast, our method uses a fully probabilistic framework that enables uncertainty quantification in the latent representations. Moreover, all operations in our method are differentiable, enabling seamless computation and backpropagation of gradients, allowing for an end-to-end training of the model. This also lets us use the reparameterization trick introduced in the DVAE++ (Vahdat et al., 2018b), which we found to be more stable than continuous relaxations like the Gumbel-Softmax, used in VQ-VAEs with stochastic quantization (Williams et al., 2020).

In RD theory, the focus is on compression within the latent space, typically analyzed from an encoder/decoder and reconstruction (distortion) perspective. RD theory sets theoretical limits on achievable compression rates and describes how practical models may diverge from these limits. Using RD practical compression methods, Townsend et al. (2019) demonstrates how asymmetric numeral systems (ANS) can be integrated with a VAE to improve its compression rate by jointly encoding sequences of data points, bringing performance closer to RD theoretical limits. Similarly, Chen et al. (2022) shows that a complex prior distribution in a VAE using an autoregressive invertible flow narrows the gap between the approximate posterior and the true posterior, thereby enhancing the overall performance of the VAE. We note that even using an independent prior, the hierarchical code structure outlined in Section 6 naturally decouples information across different latent spaces at various conceptual levels. Specifically, the most relevant information (class label) is encoded by the most protected latent space, while the other space captures fine-grained features. This effect is not straightforward to enforce through direct design of a more complex prior.

Instead, our method complements these efforts by introducing redundancy in the generative pathway to enhance variational inference, resulting in more accurate approximations of the latent posterior. Importantly, even when using a complex prior distribution, ECCs can still be leveraged to improve variational inference and boost overall model performance. In this paper, we demonstrate that our approach yields more robust models even with a simple, fixed independent prior, as evidenced by improved log-likelihood, generation quality, and reconstruction metrics. Moreover, in Section 6, we show that integrating a hierarchical ECC with the same independent prior leads to even greater performance gains.

## M  VARIATIONAL INFERENCE AT CODEWORD LEVEL

Here, we present an alternative variational family that is valid for any ECC, including random codes. We assume that we have a deterministic mapping of the form $c = C(m)$. We assume a variational family of the form

$$q_{\boldsymbol{\eta}}(\boldsymbol{c}, \boldsymbol{z}|\boldsymbol{x}) = q_{\boldsymbol{\eta}}(\boldsymbol{c}|\boldsymbol{x})p(\boldsymbol{z}|\boldsymbol{c}), \tag{18}$$

$$q_{\boldsymbol{\eta}}(\boldsymbol{c}|\boldsymbol{x}) \propto p(\boldsymbol{c})q_{\boldsymbol{\eta}}^{u}(\boldsymbol{c}|\boldsymbol{x}), \tag{19}$$

$$q_{\boldsymbol{\eta}}^{u}(\boldsymbol{c}|\boldsymbol{x}) = \prod_{j=1}^{M/R} \mathrm{Ber}(g_{j,\boldsymbol{\eta}}(\boldsymbol{x})), \tag{20}$$

where $g_{\boldsymbol{\eta}}(\boldsymbol{x})$ represents the output of the encoder with a parameter set denoted as $\boldsymbol{\eta}$. Note that $q_{\boldsymbol{\eta}}^{u}(\boldsymbol{c}|\boldsymbol{x})$ corresponds to the *uncoded* posterior, which we subsequently constrain using the prior distribution $p(\boldsymbol{c})$ over the code words to obtain the *coded* posterior $q_{\boldsymbol{\eta}}(\boldsymbol{c}|\boldsymbol{x})$. Then, the *coded* posterior distribution can be defined as a categorical distribution over the set of code words $C(\boldsymbol{m})$, which is given by

$$
\begin{aligned}
q_{\boldsymbol{\eta}}(\boldsymbol{c}|\boldsymbol{x}) &= \mathrm{Cat}\Big(\Big[\frac{1}{W}q_{\boldsymbol{\eta}}^{u}(\boldsymbol{c}_1|\boldsymbol{x}), \ldots, \frac{1}{W}q_{\boldsymbol{\eta}}^{u}(\boldsymbol{c}_{2^M}|\boldsymbol{x})\Big]\Big) \\
&= \frac{1}{W}\prod_{\boldsymbol{c}_i \in C(\boldsymbol{m})}\prod_{j=1}^{M/R} g_{j,\boldsymbol{\eta}}(\boldsymbol{x})^{c_{i,j}}(1 - g_{j,\boldsymbol{\eta}}(\boldsymbol{x}))^{(1-c_{i,j})},
\end{aligned} \tag{21}
$$

where $W = \sum_{\boldsymbol{c}_i \in C(\boldsymbol{m})} q_{\boldsymbol{\eta}}^{u}(\boldsymbol{c}_i|\boldsymbol{x})$ is a constant for normalization.

Inference is done by maximizing the ELBO, which can be expressed as

$$\text{ELBO} = \int q_{\boldsymbol{\eta}}(\boldsymbol{c}, \boldsymbol{z} | \boldsymbol{x}) \log \left( \frac{p_{\boldsymbol{\theta}}(\boldsymbol{x}, \boldsymbol{z}, \boldsymbol{c})}{q_{\boldsymbol{\eta}}(\boldsymbol{c}, \boldsymbol{z} | \boldsymbol{x})} \right) d\boldsymbol{c} d\boldsymbol{z}$$

$$= \mathbb{E}_{q_{\boldsymbol{\eta}}(\boldsymbol{c}, \boldsymbol{z} | \boldsymbol{x})} \log \left( \frac{p_{\boldsymbol{\theta}}(\boldsymbol{x} | \boldsymbol{z}) p(\boldsymbol{z} | \boldsymbol{c}) p(\boldsymbol{c})}{q_{\boldsymbol{\eta}}(\boldsymbol{c} | \boldsymbol{x}) p(\boldsymbol{z} | \boldsymbol{c})} \right) \tag{22}$$

$$= \mathbb{E}_{q_{\boldsymbol{\eta}}(\boldsymbol{c}, \boldsymbol{z} | \boldsymbol{x})} \log p_{\boldsymbol{\theta}}(\boldsymbol{x} | \boldsymbol{z}) - \mathcal{D}_{KL}\big(q_{\boldsymbol{\eta}}(\boldsymbol{c} | \boldsymbol{x}) || p(\boldsymbol{c})\big).$$

In this case, due to the inability to compute the KL Divergence in closed form, we approximate it via Monte Carlo, sampling from the categorical distribution $q_{\boldsymbol{\eta}}(\boldsymbol{c} | \boldsymbol{x})$. The reconstruction term also needs to be approximated via Monte Carlo. Since the use of channel coding introduces structural dependencies among the components of the vectors $\boldsymbol{c}$, we can no longer assume their independence. Consequently, the formulation of the smoothing transformation as independent mixtures introduced in the DVAE is no longer applicable. Hence, this approach involves sampling $\boldsymbol{c}'$ from the categorical distribution $q_{\boldsymbol{\eta}}(\boldsymbol{c} | \boldsymbol{x})$ and subsequently applying the transformation over the sampled word. Thus, we obtain a smooth transformation for each sample $\boldsymbol{c}'$ using the inverse CDFs of $p(z_j | c_j = 0)$ and $p(z_j | c_j = 1)$, which are given by

$$F^{-1}_{p(z_j | c'_j = 0)}(\rho) = -\frac{1}{\beta} \log \big(1 - \rho(1 - e^{-\beta})\big), \tag{23}$$

$$F^{-1}_{p(z_j | c'_j = 1)}(\rho) = \frac{1}{\beta} \log \big(\rho(1 - e^{-\beta}) + e^{-\beta}\big) + 1. \tag{24}$$

These are differentiable functions that convert samples $\rho$ from a uniform distribution $\mathcal{U}(0, 1)$ into a sample from $q_{\boldsymbol{\eta}}(\boldsymbol{z}, \boldsymbol{c} = \boldsymbol{c}' | \boldsymbol{x})$ following

$$q_{\boldsymbol{\eta}}(\boldsymbol{z}, \boldsymbol{c} = \boldsymbol{c}' | \boldsymbol{x}) =$$

$$= \prod_{j=1}^{M/R} \left[ F^{-1}_{p(z_j | c'_j = 1)}(\rho)^{c'_j} + F^{-1}_{p(z_j | c'_j = 0)}(\rho)^{(1 - c'_j)} \right]. \tag{25}$$

Thus, we can apply the reparameterization trick to obtain samples from the latent variable $\boldsymbol{z}$ and optimize the reconstruction term of the ELBO with respect to the parameters $\boldsymbol{\theta}$ of the decoder.

The KL Divergence term is approximated via Monte Carlo, drawing samples from $q_{\boldsymbol{\eta}}(\boldsymbol{c} | \boldsymbol{x})$. As it is not possible to backpropagate through discrete variables, we approximate the gradients with respect to the parameters of the encoder using the REINFORCE leave-one-out estimator Salimans & Knowles (2014); Kool et al. (2019), given by

$$\widehat{g}_{LOO} =$$

$$= \frac{1}{S-1} \left[ \sum_{s=1}^{S} f_{\boldsymbol{\eta}}\big(\boldsymbol{z}^{(s)}, \boldsymbol{c}^{(s)}\big) \bigtriangledown_{\boldsymbol{\eta}} \log q_{\boldsymbol{\eta}}\big(\boldsymbol{c}^{(s)} | \boldsymbol{x}\big) - \overline{f}_{\boldsymbol{\eta}} \sum_{s=1}^{S} \bigtriangledown_{\boldsymbol{\eta}} \log q_{\boldsymbol{\eta}}\big(\boldsymbol{c}^{(s)} | \boldsymbol{x}\big) \right], \tag{26}$$

where

$$f_{\boldsymbol{\eta}}(\boldsymbol{z}^{(s)}, \boldsymbol{c}^{(s)}) = \log \left( \frac{q_{\boldsymbol{\eta}}(\boldsymbol{c}^{(s)} | \boldsymbol{x})}{p_{\boldsymbol{\theta}}(\boldsymbol{x} | \boldsymbol{z}^{(s)}) p(\boldsymbol{c}^{(s)})} \right), \tag{27}$$

$$\overline{f}_{\boldsymbol{\eta}} = \frac{1}{S} \sum_{s=1}^{S} f_{\boldsymbol{\eta}}(\boldsymbol{z}^{(s)}, \boldsymbol{c}^{(s)}). \tag{28}$$

Defining a distribution over the codebook can seem intuitive, but scalability becomes challenging as the size of the codebook increases. The reason is that the posterior distribution must be evaluated for all codewords during both inference and test time. However, it can still provide a bound, enabling the utilization of more complex codes with theoretical guarantees that can outperform the previously proposed repetition codes.

---

**Algorithm 3** Training the model with inference at codeword level.

---

1: **Input:** training data $\boldsymbol{x}_i$, codebook.
2: **repeat**
3:    $q_{\boldsymbol{\eta}}^u(\boldsymbol{c}|\boldsymbol{x}_i) \leftarrow$ forward encoder $g_{\boldsymbol{\eta}}(\boldsymbol{x}_i)$
4:    $q_{\boldsymbol{\eta}}(\boldsymbol{c}|\boldsymbol{x}_i) \leftarrow$ evaluate $q_{\boldsymbol{\eta}}^u(\boldsymbol{c}|\boldsymbol{x}_i)$ over the codebook and normalize
5:    $\tilde{\boldsymbol{c}} \leftarrow$ sample from $q_{\boldsymbol{\eta}}(\boldsymbol{c}|\boldsymbol{x}_i)$
6:    $\boldsymbol{z} \leftarrow$ modulate $\tilde{\boldsymbol{c}}$
7:    $p_{\boldsymbol{\theta}}(\boldsymbol{x}|\boldsymbol{z}) \leftarrow$ forward decoder $f_{\boldsymbol{\theta}}(\boldsymbol{z})$
8:    Compute ELBO according to equation 22
9:    Compute encoder's gradients according to equation 26
10:   $\boldsymbol{\theta}, \boldsymbol{\eta} \leftarrow Update(ELBO)$
11: **until** convergence

---

## N    COMPUTATIONAL RESOURCES

All the experiments in this paper were conducted on a single GPU. Depending on availability, we used either a Titan X Pascal with 10GB of RAM, a Nvidia GeForce GTX with 10GB of RAM, or a Nvidia GeForce RTX 4090 with 24GB of RAM. Since training times varied significantly based on the hardware used, we were unable to provide comparable training times.

## O    CODED DVAE SCHEME

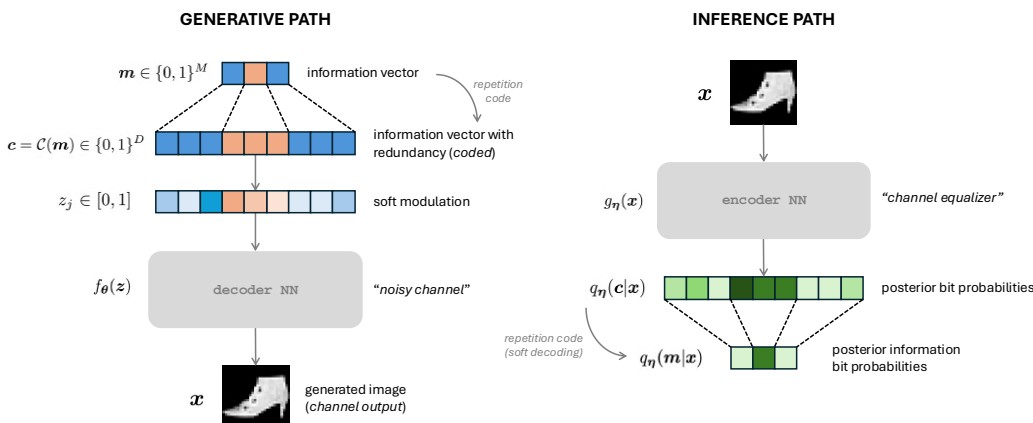

Figure 39: Graphic representation of the coded DVAE, illustrating both the generative and inference paths.

# P    HIERARCHICAL CODED DVAE SCHEME

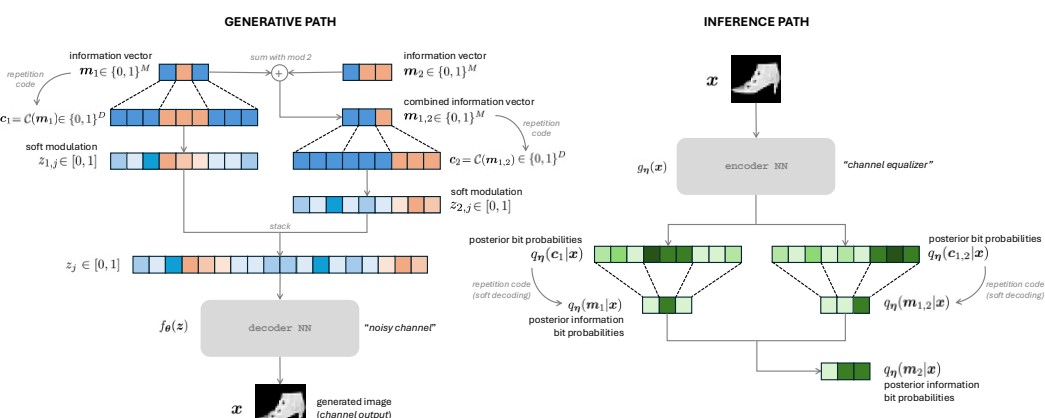

Figure 40: Graphic representation of the hierarchical coded DVAE, illustrating both the generative and inference paths.

