# OpenReview forum: "Improved Variational Inference in Discrete VAEs using Error Correcting Codes"
_ICLR.cc/2025/Conference — ICLR 2025 Conference Withdrawn Submission_

### Official Review · Reviewer_tR1Q · 2024-10-28

**Soundness:** 3
**Presentation:** 3
**Contribution:** 4
**Rating:** 5
**Confidence:** 3

**Summary:**

This paper proposes a novel approach to enhancing variational inference for discrete latent variable models. The method introduces redundancy in the latent space by leveraging error-correcting codes (ECC) from coding theory, resulting in improved performance. The proposed technique is applied to discrete variational autoencoders (DVAEs) with binary latent variables. It demonstrates improvements in generative quality, data reconstruction, and uncertainty calibration across datasets such as MNIST, Fashion MNIST, CIFAR-10, and Tiny ImageNet.

**Strengths:**

- The integration of ECC into variational inference is innovative, and the concept shows promise for applicability across various latent space models.
- The method is validated on diverse datasets, including MNIST, Fashion MNIST, CIFAR-10, and Tiny ImageNet, providing evidence of its generalizability.
- The paper comprehensively evaluates coded and uncoded models using various metrics, such as PSNR, log-likelihood, reconstruction quality, and uncertainty calibration.

**Weaknesses:**

- The paper appears to mix theoretical claims with empirical results. For instance, the statement, "This way, it is possible to reduce the mistakes committed when comparing $m$ with samples drawn from $q(m|x)$, obtaining a tighter approximation to the true posterior $ p(m|x)$," suggests a theoretical result. However, it is unclear if this has been formally proven. Additionally, there is no explicit theoretical proof showing that the inclusion of ECC improves generalization performance. Based on my review, such proof does not seem present in the paper.
- As Equation (9) indicates, using ECC increases the number of parameters due to the added redundancy. This corresponds to an effective increase in the dimensionality of the latent variables compared to the original model without ECC. In the numerical experiments, comparisons should be made with uncoded models with the same total number of latent dimensions. For example, in the case where $ c=10$, it would be more appropriate to compare against an uncoded model with $z=10$. Whether the observed performance improvements are due to the added parameters or the ECC itself remains unclear. The authors should clarify the number of parameters in the models and specify the dimensions used for $c$ and $z$. Based on my understanding, the notation "CODED 5/100" in Figures 4, 5, and 6 likely corresponds to $z=5$ and $c=100$, meaning the model has parameters associated with $c=100$. In this case, wouldn't the appropriate baseline be an uncoded model with $z=100$?
- The paper appears to exceed the page limit, which I understand to be 10 pages. Could you confirm whether the page count adheres to the conference guidelines?

**Questions:**

- Why is the parameter $c$ absent on the left-hand side of Equation (8)? Could you clarify the reasoning behind this formulation?
- How does the dependency on $\beta$ manifest in Equation (7)? It would be helpful to have a more detailed explanation of its role.
- If the transformations in Equations (10) and (11) are applied, can it still be demonstrated that the Evidence Lower Bound (ELBO) serves as a lower bound for the log-likelihood?

---

> ### Author Response · Authors · 2024-11-15
> **Response**
>
> **Reduced error rates in inference indicate a tighter posterior approximation**
>
> In this work, we conceptualize the process of inference via $q(\boldsymbol{m}|\boldsymbol{x})$ as decoding the latent variable $\boldsymbol{m}$ given the observed data $\boldsymbol{x}$. While VAEs are often viewed as lossy compression models that aim to minimize reconstruction error while regularizing the encoding with a prior, our contribution is best understood from a generative perspective. Here, we sample a vector $\boldsymbol{m}$, generate an image $\boldsymbol{x}$, and seek to minimize the error rate in recovering $\boldsymbol{m}$ from $\boldsymbol{x}$. Achieving this requires the variational approximation $q(\boldsymbol{m}|\boldsymbol{x})$ to closely align with the true posterior. Estimating $\boldsymbol{m}$ from $\boldsymbol{x}$ thus involves approximating the true posterior $p(\boldsymbol{m}|\boldsymbol{x})$ with $q(\boldsymbol{m}|\boldsymbol{x})$. The difference between $q(\boldsymbol{m}|\boldsymbol{x})$ and $p(\boldsymbol{m}|\boldsymbol{x})$ defines the variational bound, representing the “noise” in this process. By introducing structured redundancy through the ECC, we improve the posterior $q(\boldsymbol{m}|\boldsymbol{x})$, thereby reducing the variational gap. Our results show that this approach decreases error rates (as illustrated in Table in Figure 4) and enhances model performance overall. It's worth noting that the optimal decoder (the one that minimizes the error rate in the estimation of $\boldsymbol{m}$) is the one that maximizes $p(\boldsymbol{m}|\boldsymbol{x})$; any other decoder will yield higher error rates. Thus, a reduced bit error rate (BER) in inference indicates a closer approximation to the true posterior [1].

---

> > ### Author Response · Authors · 2024-11-15
> > **Response continuation**
> >
> > **The dimensionality of the latent space and the number of trainable parameters**
> >
> > The dimensionality of the latent space is determined by the number of information bits, corresponding to the dimensionality of the binary latent variable $\boldsymbol{m}$, as this defines the set of possible latent vectors for representing the information. In the coded approach, redundancy is introduced to safeguard the information in $\boldsymbol{m}$, generating a codeword $\boldsymbol{c}$. Note that $\boldsymbol{c}$ carries the same information as $\boldsymbol{m}$ since the encoding process is injective and adds redundancy deterministically without increasing flexibility. Thus, a fair comparison between coded and uncoded models must maintain the same number of information bits (i.e., equal dimensionality of $\boldsymbol{m}$).
> >
> > The notation “CODED 5/100” represents the code rate, which is defined as the ratio of information bits in $\boldsymbol{m}$ to the total number of bits in the resulting codeword $\boldsymbol{c}$. Therefore, this model should be compared with a 5-information bits baseline. Note that a 5/100 coded model has $2^5 = 32$ possible codewords to represent the information, while a 100-bit uncoded model has $2^{100}$ possible vectors.
> >
> > We used a consistent architecture across all experiments. However, for coded models, the encoder’s output and decoder’s input layers were adjusted to accommodate the dimensionality of the encoded vectors $\boldsymbol{c}$, resulting in a slight increase in **trainable parameters** compared to uncoded models (see Table 18 in Appendix J.2 for an example of the difference in parameter count between coded and uncoded models). Since models with more parameters may show better performance, we conducted an ablation study on the model’s trainable parameters (see Appendix J.2) to verify that the performance gains in coded models were not due to this difference. By adjusting the encoder’s last hidden dimension and the decoder’s first hidden dimension, we ensured that both coded and uncoded configurations had approximately the same number of trainable parameters (as indicated in Table 18), confirming that the improved performance is not attributed to the small increase in parameter count.
> >
> > However, we trained uncoded models on FMNIST using 50, 100, 200, and 300 bits to assess their performance. The results are available in our response to Reviewer 82yp. We observe that coded models with 10 information bits achieve performance that is comparable to, or even surpasses, that of uncoded models (as shown by the PSNR and Semantic Accuracy metrics), despite having latent dimensionalities that are 10, 20, or 30 times larger (note the column where we indicate the number of possible latent vectors).In generation, we observed that uncoded models with high-dimensional latent spaces fail to fully specialize all codebook entries during training, leading to generation artifacts (i.e. noisy images where different object classes overlap). In contrast, coded models do not exhibit such artifacts, demonstrating more effective use of the latent space. This further supports that the superior performance of coded models is not attributed to the difference in trainable parameters but rather to more efficient utilization of the latent space enabled by improved posterior inference.
> >
> > **Page limit**
> >
> > According to the ICLR 2025 Author Guide: “Authors are strongly encouraged to include a paragraph-long Reproducibility Statement at the end of the main text (before references) to discuss the efforts that have been made to ensure reproducibility. [...] **This optional reproducibility statement will not count toward the page limit, but should not be more than 1 page.**”
> >
> > **Why is the parameter $c$ absent on the left-hand side of Equation (8)? Could you clarify the reasoning behind this formulation?**
> >
> > This is simply a matter of notation. Since $\boldsymbol{c}$ is an injective, deterministic function of $\boldsymbol{m}$, the randomness in $\boldsymbol{c}$ is uniquely determined by $\boldsymbol{m}$, and the equation could be rewritten as $q_{\boldsymbol{\eta}}(\boldsymbol{m}, \boldsymbol{z}|\boldsymbol{x}) = q_{\boldsymbol{\eta}}(\boldsymbol{m}, \boldsymbol{z}|\boldsymbol{x}) p(\boldsymbol{z}|\boldsymbol{m}^T\boldsymbol{G})$. We will provide further clarification on this point in the final version of the paper.

---

> > > ### Author Response · Authors · 2024-11-15
> > > **Response continuation**
> > >
> > > **If the transformations in Equations (10) and (11) are applied, can it still be demonstrated that the Evidence Lower Bound (ELBO) serves as a lower bound for the log-likelihood?**
> > >
> > > In Eq. (10) and Eq. (11), we add an extra step in computing the marginal posteriors $q(m_i|\boldsymbol{x})$ by utilizing the known structure of the ECC (through soft decoding). In the coded case, the encoder network outputs the posterior bit probabilities for $\boldsymbol{c}$ rather than for $\boldsymbol{m}$. Since $\boldsymbol{c}$ is an injective, deterministic transformation of $\boldsymbol{m}$, we can easily compute the marginal posteriors $q(m_i|\boldsymbol{x})$ using the code’s structure, specifically via $G$. The ELBO expression remains unchanged after introducing the ECC; the only difference is in the calculation of the posterior bit probabilities in $q(\boldsymbol{m}|\boldsymbol{x})$. We simply propose a refined approximation to the posterior distribution, where any approximation yields a lower bound.
> > >
> > > **Conclusion**
> > >
> > > While we appreciate the reviewer’s acknowledgment of the innovative nature of our work—particularly the integration of ECC into variational inference and its potential applicability across various latent space models—we respectfully disagree with the assessment that our paper falls 'marginally below the acceptance threshold.' We believe have addressed all of the reviewer’s concerns and will make further updates to ensure that the points raised are even clearer. Given the novelty and potential impact of our work, we believe it merits a more favorable evaluation.
> > >
> > > **References**
> > >
> > > [1] Richardson, T., & Urbanke, R. (2008). Modern Coding Theory. Cambridge University Press.

---

> > > ### Comment · Reviewer_tR1Q · 2024-11-21
> > > **Response**
> > >
> > > >  In contrast, coded models do not exhibit such artifacts, demonstrating more effective use of the latent space.
> > >
> > > I am convinced that the performance improvement of your proposed method is not solely attributable to an increase in the number of learnable parameters. However, I still lack a qualitative understanding of why this encoding approach leads to improved performance. Therefore, at this stage, I can only regard it as an empirical heuristic method that performs well. While the idea is novel and interesting, I will maintain my current score for the reasons outlined above. If you can provide theoretical insights or quantitatively demonstrate the advantages of your approach, I will reconsider my score.
> > > Even without a theoretical basis, it is essential to experimentally analyze how encoding influence the distribution of the latent space and how this influence enables high accuracy to be achieved with a smaller number of trainable parameters.
> > >
> > > I understand well regarding the page limitations and the explanation of the equations. Thank you very much.

---

> > ### Comment · Reviewer_tR1Q · 2024-11-21
> > **Response**
> >
> > > By introducing structured redundancy through the ECC, we improve the posterior $q(m|x)$, thereby reducing the variational gap. Our results show that this approach decreases error rates (as illustrated in Table in Figure 4) and enhances model performance overall.
> >
> > Thank you for your response.
> > What I am concerned about is that it is unclear whether the reduction of the variational gap is a theoretical result or an experimental one.  If it is a theoretical result, I would appreciate it if you could include the proof in the appendix or elsewhere in the revised version.  If it is an experimental result, I believe it should be explicitly stated as an empirical finding.

---

> ### Author Response · Authors · 2024-11-22
> **Response**
>
> Thank you for your feedback! We will address your concerns by responding to each comment individually.
>
> >What I am concerned about is that it is unclear whether the reduction of the variational gap is a theoretical result or an experimental one. If it is a theoretical result, I would appreciate it if you could include the proof in the appendix or elsewhere in the revised version. If it is an experimental result, I believe it should be explicitly stated as an empirical finding.
>
> Our approach is definitely grounded in a theoretical base. In information theory, it is well established that “Maximum a Posteriori (MAP) decoding is the minimizer of the probability of error if all input vectors are equally probable” (we refer to Chapter 1.5 in [1]). This means that the solution that minimizes the error rate in estimating $\boldsymbol{m}$ from $\boldsymbol{x}$ is the one that maximizes $p(\boldsymbol{m}|\boldsymbol{x})$ w.r.t. $\boldsymbol{m}$. In variational inference, we do not have access to the true posterior $p(\boldsymbol{m}|\boldsymbol{x})$ but to $q(\boldsymbol{m}|\boldsymbol{x})$, which we construct using the ECC constraints on top of the encoder NN output $g_{\boldsymbol{\eta}}(\boldsymbol{x})$.
>
> Consequently, a lower bit error rate (BER) in inference corresponds to a better approximation of the true posterior. While this result is well established in information theory, to the best of our knowledge, this is the first time it has been applied within the VAE framework. To validate our approach, we not only report lower BER values in the table presented in Figure 4 but also demonstrate a smaller gap to capacity, as previously shown to Reviewer 82yp. For completeness, we include the gap table here and will integrate it into the paper.
>
> | Model | LL train | ELBO train | Gap train| LL test | ELBO test | Gap test |
> |----------|------------|----------------|-------------|-----------|---------------|-------------|
> | uncoded 5 | -266.157 | -274.525 | 8.368 | -267.703 | -275.416 | 7.713 |
> | coded 5/50 | -239.379 | -244.286 | 4.907| -241.882 | -247.024 | 5.142 |
> | uncoded 8 | -247.964 | -256.792 | 8.828 | -249.880 | -259.049 | 9.169 |
> | coded 8/80 | -227.550 | -230.392| 2.842 | -232.992 | -236.924 | 3.932 |
> | uncoded 10 | -242.842 | -252.555 | 9.713 | -244.997 | -254.714 | 9.717 |
> | coded 10/100 | -222.011 | -223.970 | 1.959 | -230.772 | -234.206 | 3.434|

---

> ### Author Response · Authors · 2024-11-22
> **Response Continuation**
>
> > I still lack a qualitative understanding of why this encoding approach leads to improved performance.
>
> Following our previous comment, the key behind our method is that **we exploit the known correlations introduced by the ECC in inference to refine the output of the encoder NN** $g_{\boldsymbol{\eta}}(\boldsymbol{x})$, obtaining a better approximation to the posterior. The reason why this works, intuitively, is that once we introduce the ECC, the resulting codewords form an $m$-dimensional subspace of the larger $n$-dimensional space in which any two codewords $\boldsymbol{c}$ in the latent space are now separated by at least some minimum distance (i.e., it is a sphere packing). This structure, where $2^{n-m}$ of $n$-tuples are not valid codewords, allows a decoder to tolerate errors and attempt to identify the most likely codeword. For example in the case of a 5/50 repetition code, only $2^5$ of the $2^{50}$ possible combinations in the 50-dimensional space are valid (that is why introducing redundancy does not increase flexibility, since the number of available vectors remains the same). Here, the minimum distance is 50 and this large separation of codeword symbols in the latent space provides significant error correcting capability and coding gain (defined in information theory as the reduction in signal power for a target error rate over a system without coding). Indeed, given a specific channel and codeword weight distribution, one can theoretically characterize the probability of failure of the optimal decoder which must, in turn, reduce as a function of increasing minimum distance/improving weight spectrum [1].
>
> Given an observation $\boldsymbol{x}$, ECC decoding algorithms can efficiently (attempt to) find the codeword $\boldsymbol{c}$ to maximize the probability of $p(\boldsymbol{c}|\boldsymbol{x})$ over all $2^m$ codewords and thereby maximize the probability $p(\boldsymbol{m}|\\boldsymbol{x})$ of the estimate of $\boldsymbol{m}$, since $\boldsymbol{c}$ and $\boldsymbol{m}$ are deterministically and injectively related. Specifically, for repetition codes, error detection and correction can be achieved through a straightforward majority voting approach. Since each information bit is repeated $L$ times, the original value can be recovered by selecting the most frequent value among the repetitions. In our method, in the case of repetition codes, we employ a **soft majority voting strategy during inference** to compute the posterior probability of each information bit in a theoretically optimal (MAP) way. This is done by evaluating the all-ones and all-zeros probabilities for the $L$ repeated bits. It is important to note that, while our paper primarily focuses on repetition codes for simplicity, the proposed method can be extended to more advanced coding schemes, as discussed in Section 6. Transitioning to more robust error-correcting codes has the potential to further enhance performance, as indicated by our preliminary results using polar codes.
>
> > If you can provide theoretical insights or quantitatively demonstrate the advantages of your approach, I will reconsider my score.
>
> As mentioned in our previous response, it is a well-established result in information theory that the optimal decoder, which minimizes the error rate in estimating $\boldsymbol{m}$, is the one that maximizes $p(\boldsymbol{m}|\boldsymbol{x})$. Any other estimator will result in higher error rates [1]. Employing this mechanism here helps to improve the estimate of the posterior $q(\boldsymbol{m}|\boldsymbol{x})$. Estimating $\boldsymbol{m}$ from $\boldsymbol{x}$ involves approximating the true posterior $p(\boldsymbol{m}|\boldsymbol{x})$ distribution with $q(\boldsymbol{m}|\boldsymbol{x})$, our variational proposal. The closer these distributions are, the fewer errors we make when comparing $\boldsymbol{m}$ with samples from $q(\boldsymbol{m}|\boldsymbol{x})$. The gap between $q(\boldsymbol{m}|\boldsymbol{x})$ and $p(\boldsymbol{m}|\boldsymbol{x})$ is precisely the variational bound and can be considered as the “channel noise”. The structured redundancy introduced by the ECC, with large minimum distance, is exploited to improve the posterior $q(\boldsymbol{m}|\boldsymbol{x})$ and therefore reduce the variational gap to the true posterior. Our results confirm that this approach reduces error rates (see table included in Figure 4, for example), improves log-likelihoods (refer to the table in Figure 4), narrows the variational gap (see the table included in the previous comment), and enhances overall model performance.

---

> > ### Comment · Reviewer_tR1Q · 2024-11-25
> > **Response**
> >
> > Thank you very much for your detailed response.
> >
> > Your experimental results are indeed very intriguing, and I understand that your method achieves a closer approximation to the true posterior distribution. However, I would like to point out that a closer match between the true posterior and the variational posterior does not necessarily guarantee a reduction in generalization error. For instance, a weaker variational posterior could act as an implicit regularizer, potentially benefiting generalization.
> >
> > Furthermore, even without your proposed method, increasing the complexity of the variational posterior can generally reduce the gap between the variational posterior and the true posterior. What I am particularly interested in is a theoretical explanation of why your method improves generalization performance, beyond simply closing this gap. As such, I am not yet fully convinced that your proposed method has a definitive advantage in enhancing generalization.
> >
> > Taking all of these considerations into account, I will maintain my current score.

---

> > > ### Author Response · Authors · 2024-11-25
> > >
> > > Thank you for your response and for acknowledging the value of our experimental results despite your reservations.
> > >
> > > We would like to emphasize that the primary aim of this paper is to demonstrate that introducing ECCs into the discrete VAE framework enables a tighter posterior approximation and, therefore, helps to reduce the variational gap. Our experimental results support this claim, showing reductions in the variational gap and improved performance on both training and test sets, which in the end indicates improved inference.
> > >
> > > Of course, many factors influence the tightness of the variational approximation, including the complexity of the true posterior, the choice of the variational family, the architectures used to parameterize the distributions, and the sampling strategies for evaluating the ELBO objective. As the reviewer notes, alternative approaches (such as increasing the complexity of the variational posterior) could also address these factors and reduce the gap between the variational and true posterior distributions in both continuous and discrete settings.
> > >
> > > In this work, we present a **proof of concept** for a novel and simple method that offers an alternative perspective on the inference problem—one that, to the best of our knowledge, has not been explored in the literature. **We believe this contribution is valuable and worth sharing with the community**. This seminal work opens the door to further theoretical analysis, which is beyond the scope of this paper.
> > >
> > > To help this proof of concept make a meaningful contribution to the community—enabling us to build on the feedback received and explore a more theoretical direction in future work—we request that the reviewer, even if unable to grant an "accept" (8) due to reservations, consider raising the score to a (6), “marginally above the acceptance threshold”. We genuinely believe that, with all the evidence we have provided, this work is deserving of such consideration.

---

> > > > ### Author Response · Authors · 2024-11-28
> > > >
> > > > We kindly direct you to the general comment we have posted, as we address your concerns.
> > > >
> > > > As the discussion period is still ongoing, we would greatly appreciate any specific suggestions on actions we could take to further address your concerns and encourage you to consider raising your score after reviewing our comments.

---

> > > > > ### Comment · Reviewer_tR1Q · 2024-11-30
> > > > > **Response**
> > > > >
> > > > > Thank you for your response.
> > > > >
> > > > > I pointed out that using a more complex variational posterior distribution would reduce the variational gap.
> > > > > Therefore, while you claim that the redundancy you introduced can reduce the variational gap, this approach does not offer a clear advantage over other methods that achieve the same goal by increasing the complexity of the variational posterior.
> > > > >
> > > > > As such, **even though your "proof-of-concept" may be interesting, I cannot justify adjusting the score unless your proposed method demonstrates some quantitative benefits beyond simply reducing the variational gap**.
> > > > > At this stage, I am still trying to understand why the proposed method, reducing the variational gap, alone would lead to performance improvements.
> > > > >
> > > > > If you aim to establish the superiority of your method theoretically, you should provide evidence—through statistical analysis or other means—that your approach improves generalization performance or results in a smoother ELBO loss landscape beyond merely reducing the variational gap.
> > > > > Alternatively, if your argument is experimental, you should present additional quantitative evidence of the benefits of your method apart from the reduction in the variational gap.
> > > > >
> > > > > However, these changes will require substantial revisions, and I intend to maintain my current score.

---

> > > > > > ### Author Response · Authors · 2024-12-01
> > > > > >
> > > > > > We would like to remind the reviewer that reducing the variational gap is a common goal in VAE literature. The evidence lower bound (ELBO), which serves as the model's training objective, becomes tight when $q_{\eta}(\boldsymbol{m}|\boldsymbol{x}) = p(\boldsymbol{m}|\boldsymbol{x})$. Thus, an improved ELBO—or equivalently, a reduced variational gap (achieved through a tighter posterior approximation)—directly contributes to a better model (we would appreciate it if the reviewer could provide any references that suggest otherwise). As mentioned in our last general comment, various approaches in the literature have been proposed to reduce this gap to the true posterior distribution, thereby enhancing inference. A tighter posterior approximation allows the model to more effectively map data to the latent space, improving reconstruction and supporting the learning of structured latent spaces, which can, in turn, enhance generative performance. A clear example demonstrating that a tighter ELBO results in a better generative model is provided in the foundational IWAE paper [1]. Notably, in our experiments, we trained the uncoded model using the IWAE bound, but its performance remained significantly below that achieved with our proposed approach leveraging ECCs.
> > > > > >
> > > > > > Therefore, we argue that improved inference (reduced variational gap) is crucial in deep generative modeling, particularly for discrete representations where the state-of-the-art remains significantly limited. Our method offers a novel approach in this context. For a detailed discussion on how our contribution fits within the current state of the art, we kindly refer the reviewer to our last general comment.
> > > > > >
> > > > > > We would also like to reiterate that we are not claiming our method surpasses the state-of-the-art approaches discussed (including flexible variational posteriors). Rather, we present it as a novel alternative for achieving the same objective, offering a novel perspective on the inference problem with the potential to inspire future research. A key contribution of our work is that it provides an innovative approach inherently compatible with these existing methods. For example, consider the hierarchical extension of our method through the integration of polar codes, as detailed in Section 6.
> > > > > >
> > > > > > Regarding the quantitative evidence supporting the benefits of our method, as discussed in the general comment, we conducted a comprehensive empirical evaluation (which the reviewer acknowledged in their initial review). Our results show improvements on both the training and test sets (with training metrics detailed in the Appendix), indicating an overall enhancement in model fitting. We evaluated the model's reconstruction accuracy (using metrics such as PSNR, Semantic Accuracy, and Confident Semantic Accuracy), its generative performance (through visual examples and FID), and the tightness of the variational posterior (measured by the variational gap, log-likelihood, BER, and WER). These results are consistent across various datasets and model configurations, providing strong empirical evidence to support our claims. If the reviewer has specific suggestions for additional experimental results that we should include, we would be grateful for their input.
> > > > > >
> > > > > > **References**
> > > > > >
> > > > > > [1] Yuri Burda, Roger Grosse, and Ruslan Salakhutdinov (2016). Importance Weighted Autoencoders. In 4th International Conference on Learning Representations (ICLR).

---

> > > > > > > ### Comment · Reviewer_tR1Q · 2024-12-02
> > > > > > > **Response**
> > > > > > >
> > > > > > > Thank you for your response.
> > > > > > >
> > > > > > > It is generally understood that complex variational distributions can lead to degraded performance in representation learning and data generation due to overfitting and poor local minima. For instance, theoretical results in [1], [2], and prior studies cited therein have discussed these phenomena. Moreover, these issues are commonly referred to as *Posterior Collapse*.
> > > > > > >
> > > > > > > Your point is valid: when the dataset is sufficiently large, and training error can be nearly equated with generalization error, the optimization process may avoid poor local solutions and achieve results that minimize training error. In such scenarios, the smaller variational gap from the log-likelihood will invariably lead to better outcomes. However, in practice, training data is finite, training error is approximated using a limited dataset, and the loss landscape is both nonlinear and non-convex.
> > > > > > >
> > > > > > > While your idea is intriguing, I believe it is necessary to provide a theoretical advantage beyond just improving ELBO and to experimentally demonstrate the effectiveness of the proposed method with a quantitative principle that goes beyond merely evaluating generative performance or variational gaps. If this is challenging, it is important to show that your method outperforms others in terms of reducing the variational gap or offers advantages in the context of the memory-quality trade-off.
> > > > > > >
> > > > > > > The idea is interesting, but at present, I cannot fully grasp the quantitative advantages of the approach. It seems to position itself as one of several empirical methods, and I am maintaining my current score accordingly.
> > > > > > >
> > > > > > > [1]: Learning Dynamics in Linear VAE: Posterior Collapse Threshold, Superfluous Latent Space Pitfalls, and Speedup with KL Annealing, AISTATS2023
> > > > > > > [2]: Beyond Vanilla Variational Autoencoders: Detecting Posterior Collapse in Conditional and Hierarchical Variational Autoencoders, ICLR2024

---

> > > > > > > > ### Author Response · Authors · 2024-12-02
> > > > > > > >
> > > > > > > > Thank you for engaging in the discussion.
> > > > > > > >
> > > > > > > > While we are aware of the well-documented training issues in VAEs, such as posterior collapse, we do not understand why the reviewer raises this concern, as our results provide no evidence of such behavior. According to [2], *“As a consequence of posterior collapse, the latent variables extracted by the encoder in VAEs preserve less information from the input data and thus fail to produce meaningful representations as input to the reconstruction process in the decoder.”* This does not align with our experimental results, as demonstrated by the following:
> > > > > > > >
> > > > > > > > - **Diversity in Generation**: We show random, uncurated generated images that exhibit diversity in both categories and style (unlike the uncoded model).
> > > > > > > > - **Confident Semantic Accuracy**: We observe an improvement in this metric, confirming that the encoder generates meaningful representations and the mapping to the latent space is neither random nor fixed.
> > > > > > > > - **Posterior Entropy**: The mean entropy of our approximate posterior is significantly lower than the maximum entropy expected in the uniform (prior) case and much better calibrated than the uncoded case.
> > > > > > > > - **Bit Error Rate (BER)**: The significant reduction in BER shows that the coded models successfully recover the original information vector that generated a given image, providing additional evidence of meaningful representations.
> > > > > > > >
> > > > > > > > These results show that our method does not exhibit posterior collapse and instead produces robust and meaningful latent representations. Furthermore, we would like to emphasize that our approach does not rely on a complex variational family. Instead, we demonstrate that inference can be improved even when using a simple independent variational family, avoiding the issues pointed out by the reviewer.
> > > > > > > >
> > > > > > > > At this point, we do not agree with the assessment of our submission. We believe that important aspects of our work have been overlooked (for instance, the evidence that shows that we are not suffering from posterior collapse in our experiments), and the concerns raised by the reviewer are not consistent along different posts. We have already addressed the theoretical foundation of our method and demonstrated its superiority over the well-known IWAE in reducing the variational gap. Additionally, we presented a hierarchical extension that effectively decouples information across discrete latent spaces, clearly showing no evidence of posterior collapse. The claim that our model suffers from posterior collapse is unfounded and lacks any supporting evidence.
> > > > > > > >
> > > > > > > > The reviewer requested that we  *"experimentally demonstrate the effectiveness of the proposed method with a quantitative principle that goes beyond merely evaluating generative performance or variational gaps."* As we have emphasized repeatedly, our experimental evaluation extends beyond these two aspects, and we believe we have provided a thorough and comprehensive analysis of our method (as recognized by several reviewers).
> > > > > > > >
> > > > > > > > To ensure the review process remains constructive, we once again ask the reviewer to suggest specific additional experimental results they think we should include to further evaluate our method.

---

> > > > > > > > > ### Comment · Reviewer_tR1Q · 2024-12-02
> > > > > > > > > **Response**
> > > > > > > > >
> > > > > > > > > Thank you for your detailed response. I understand that your method avoids posterior collapse. However, my main concern lies in the lack of explanation for why these outcomes occur. To address this, I suggest focusing on the following:
> > > > > > > > >
> > > > > > > > > - Why does posterior collapse not occur?: Could you provide a clear explanation, either mathematically or through ablation studies, of the factors that prevent posterior collapse in your method?
> > > > > > > > > - **Why is the variational gap better than IWAE?** Is it possible to formally demonstrate why your approach reduces the variational gap more effectively than IWAE? A mathematical derivation or theoretical insight would strengthen your claims.
> > > > > > > > >
> > > > > > > > > I want to quantitatively understand the mechanisms behind how your method achieves its results. Currently, I am unable to clearly grasp the relationship between your experimental results and the underlying mechanisms. Therefore, at this stage, I regard your method as a heuristic technique for performance improvement.

---

> ### Author Response · Authors · 2024-11-22
> **Response Continuation**
>
> > Even without a theoretical basis, it is essential to experimentally analyze how encoding influence the distribution of the latent space and how this influence enables high accuracy to be achieved with a smaller number of trainable parameters.
>
> We have evaluated how the introduction of ECCs influences the posterior distribution in our experiments. One key result is the evaluation of the word and bit error rates. In this experiment, we sample a number of information vectors $\boldsymbol{m}$, feed them to the decoder NN to obtain the corresponding generated images, and map these images back to the latent space. A better approximation of the posterior distribution leads to reduced error rates [1], which we see in coded models in comparison to uncoded ones.
>
> We further analyzed the posterior distribution over the latent space through semantic accuracy and posterior uncertainty calibration experiments, focusing on two metrics: **Semantic Accuracy** and **Confident Semantic Accuracy**. Semantic Accuracy evaluates whether models successfully reconstruct images belonging to the same class as the original ones. Confident Semantic Accuracy applies the same evaluation but only considers cases where images are projected into a latent vector with a posterior probability exceeding 0.4, excluding errors when the MAP value of $q(\boldsymbol{m}|\boldsymbol{x})$ is below this threshold. In both metrics, coded models consistently outperformed uncoded ones, indicating that the posterior is more likely to map images to latent vectors of the correct semantic class. This improvement is also reflected in better reconstruction performance measured in PSNR.
>
> We also calculated the mean entropy of the posterior distribution over the latent vectors and provided examples of incorrect reconstructions on the FMNIST and MNIST datasets. These examples included the four most probable a posteriori latent vectors alongside their posterior probabilities. Our observations showed that uncoded models often map data to incorrect class vectors with high confidence, whereas coded models exhibit greater uncertainty in such cases. This is further supported by the mean entropy across the dataset: uncoded models present low posterior uncertainty, confidently projecting data into the latent space, while coded models exhibit higher entropy, suggesting that projections are made with greater uncertainty. However, as shown in the Confident Semantic Accuracy results, when coded models project data with lower uncertainty, they are more likely to map it to a vector belonging to the correct semantic class (as evidenced by the consistent improvement of Confident Semantic Accuracy over Semantic Accuracy).
>
> **Final comment**
>
> We look forward to your feedback. If our explanations require further clarification, please let us know—we would be happy to provide further details if needed. We will also be available over the weekend to address any questions.
>
> **References**
>
> [1] Richardson, T., & Urbanke, R. (2008). Modern Coding Theory. Cambridge University Press.

---

> ### Author Response · Authors · 2024-12-03
>
> We would like to remind the reviewer that we have already addressed the connection between our experimental results and the underlying mechanisms by explaining the theoretical foundations and intuition behind our method, which are grounded in well-established results from information theory.
>
> At this stage of the discussion period, it is not feasible to provide additional mathematical derivations or ablation studies; however, we will further elaborate on the intuition addressing the concerns raised by the reviewer.
>
> > Why does posterior collapse not occur?: Could you provide a clear explanation, either mathematically or through ablation studies, of the factors that prevent posterior collapse in your method?
>
> Posterior collapse is a well-studied phenomenon in the VAE literature, though we would like to point out that it has been primarily examined in the context of continuous latent spaces. Several factors are known to contribute to posterior collapse, including the presence of local minima in the training objective and the non-identifiability of the latent space. Of particular relevance to our scenario, [1] highlights how an inexact variational approximation can lead to coding inefficiencies in VAEs, potentially resulting in posterior collapse due to a form of information preference. Building on this intuition, our method mitigates this issue by providing a tighter posterior approximation, leveraging the known structure of the ECC introduced. **Note that our uncertainty calibration results demonstrate that the posterior is more open and exhibits higher variance in cases of poor reconstruction, which is a clear indication that posterior collapse is not occurring.**
>
> > Why is the variational gap better than IWAE? Is it possible to formally demonstrate why your approach reduces the variational gap more effectively than IWAE? A mathematical derivation or theoretical insight would strengthen your claims.
>
> The two approaches are fundamentally different. In IWAE, independent samples are drawn from the variational posterior and passed independently through the generative model to achieve a tighter variational bound on the marginal log-likelihood, which is then optimized. The motivation behind this approach is that the standard ELBO penalizes posterior samples that fail to explain the observations, imposing strict constraints on the model (the posterior distribution must be approximately factorial and predictable with a neural network). By relaxing these constraints, IWAE provides additional flexibility to train generative models whose posterior distributions may deviate from standard VAE assumptions.
>
> In contrast, **our method jointly propagates the output of the ECC encoder through the generative model to produce a single prediction while leveraging the introduced known correlations in the variational posterior approximation**. For repetition codes, the ECC encoder outputs repeated bits (or repeated probabilities in the case of soft encoding). However, our approach extends far beyond repetition codes, paving the way for improved inference in discrete latent variable models. Unlike IWAE, we focus on enhancing the variational proposal while adhering to the VAE assumptions.
>
> Specifically, we utilize the redundancy introduced by repetition codes to correct potential errors made by the encoder via a soft decoding approach. This leads to a more accurate approximation of $p(\boldsymbol{m}∣\boldsymbol{x})$ and an improved sampling proposal. Notably, the performance improvements achieved with our coded DVAE approach cannot be replicated by training the uncoded DVAE with the IWAE objective.
>
> Furthermore, the hierarchical example in Section 6 demonstrates the model's ability to separate features at different levels of granularity: low-level features (e.g., class) are captured in the most protected latent space, while high-level features are represented in the least protected space. This demonstrates how, through a deterministic and low-complexity coding design, we can effectively tackle the design of hierarchical discrete deep generative models—an open challenge in the state-of-the-art.
>
> Finally, we would like to point out that in the initial assessment, the reviewer considered our contribution excellent (4) and that the paper comprehensively evaluates coded and uncoded models. We have thoroughly addressed all the concerns raised by the reviewer, offering detailed explanations of the theoretical foundations of our method and how it promotes overall performance improvements. Therefore, we respectfully believe the current score does not reflect the merit of our work.
>
> **References**
>
> [1]  Chen, X., Kingma, D. P., Salimans, T., Duan, Y., Dhariwal, P., Schulman, J., ... & Abbeel, P. (2017). Variational Lossy Autoencoder. In International Conference on Learning Representations.

---

### Official Review · Reviewer_82yp · 2024-10-29

**Soundness:** 3
**Presentation:** 3
**Contribution:** 3
**Rating:** 6
**Confidence:** 4

**Summary:**

The authors propose to introduce redundancy into the latent representation in a Discrete Variational Autoencoder (DVAE) by employing error-correcting codes (ECC). This would reduce errors in samples drawn from the variational posterior, resulting in a better estimation of the true posterior.

**Strengths:**

- The contribution is novel and interesting.
- The paper is well-written, and mathematics is easy to follow.
- Introducing ECCs results in significant performance improvement in DVAEs across several datasets as demonstrated in the paper.

**Weaknesses:**

While I appreciate the contribution and the presented results in the paper, there are few concerns I have which are listed below:

- Comparison baselines: The baseline used for comparison is just DVAE with a simple independent prior. It would better highlight the effectiveness of the proposed approach if the comparison is also done with DVAEs with Boltzmann machine priors and other discrete VAE models such as VQ-VAE (discussed in the introduction section).
- The ablation study is conducted by adjusting the number of parameters of a neural network. I think it’s more convincing to compare the results of the uncoded and coded DVAEs with the same dimensionality of latent space (let's say D). (The case where the coded DVAE has redundancy included in it (D = ML with only M original message bits) while the uncoded VAE has D original message bits.)
- To directly verify the improvement in variational inference, I believe measuring the variational gap (inference gap) between the uncoded and the coded DVAEs (as done in [1]) would be helpful. This would provide a direct verification of the main claim of the paper.
- Minor: The resolution of Figure 1(b) is poor. Also, the metric used for measuring reconstruction loss in the vertical axis is not stated.

[1] Shu et. al., Amortized Inference Regularization. NeurIPS, 2018.

If the concerns are adequately addressed, I’d be ready to increase my rating for the paper.

**Questions:**

- Can the proposed method be extended to VAE with continuous latent space?

---

> ### Author Response · Authors · 2024-11-15
> **Response**
>
> **Baselines**
>
> Numerous factors affect the tightness of the variational approximation, including the complexity of the true posterior, the selection of the variational family, the architectures used to parameterize the distributions, and the sampling techniques employed to evaluate the ELBO objective. Consequently, various approaches have been proposed in the literature to address these factors and reduce the gap between the variational and true posterior distributions in both continuous and discrete settings.
>
> One popular approach to improve inference in VAEs is to define flexible priors that better align with the posterior, alleviating problems such as posterior collapse or the hole problem [1]. An example of this is the VQ-VAE [2], which fits an autoregressive model to train a prior that aligns with the inferred aggregated posterior; the VampPrior [3], where the prior is defined as a mixture of variational posteriors conditioned on learnable pseudo-inputs; or the Variational Lossy Autoencoder [4], where the authors employ an autoregressive invertible flow as prior. In this work, we introduce an alternative approach that, to the best of our knowledge, has not been previously explored in the literature. This paper aims to demonstrate that incorporating ECCs into the generative pathway can enhance inference in discrete VAEs. Since we do not tailor our solution to any specific prior distribution, this approach is fully compatible with state-of-the-art methods that utilize complex priors, such as those previously mentioned. However, we chose to focus on highlighting the significant performance improvements achievable through the introduction of ECCs alone, using a simple, fixed independent prior, which can also aid in obtaining more interpretable latent spaces. While the coded models demonstrate substantial improvements over their uncoded counterparts, we acknowledge that our results may not be directly comparable to those achieved with VQ-VAE or other advanced methods for discrete latent representations. However, establishing such comparisons is not the primary aim of this work.
>
> We remark that even using an independent prior, the hierarchical code structure outlined in Section 6 naturally decouples information across different latent spaces at various conceptual levels. Specifically, the most relevant information (class label) is encoded by the most protected latent space, while the other space captures fine-grained features. This effect is not straightforward to enforce by direct design of a more complex prior. In fact, the DVAE [5], DVAE++ [6], and DVAE# [7] consider complex and expressive Boltzmann Machine priors coupled with a hierarchy of continuous variables modeled in an autoregressive manner to model global and local factors of variation. However, we consider that conducting an extensive comparison in terms of interpretability or the ability to decouple high- and low-level information is beyond the scope of this paper.
>
> VQ-VAEs [2] stand out as state-of-the-art discrete latent space models, and we believe it would be valuable to highlight the differences between the two approaches. A primary distinction lies in the latent space’s structure and dimensionality. In image modeling, VQ-VAE typically employs a latent matrix where indices correspond to codewords in a codebook. This setup allows each latent code to explain a different patch of the original image, enhancing reconstruction and generation performance (the latter after training an autoregressive prior on the latent representations). However, this structure can complicate interpretability, as each data point is represented by a matrix of embeddings. Instead, our method projects all the information into a single low-dimensional vector $\boldsymbol{m}$, with the consequent information loss. Another important difference is that VQ-VAE is based on a non-probabilistic encoder, where the output is mapped to the nearest latent code based on a distance metric. This deterministic mapping limits the model's ability to quantify uncertainty in the latent space. In contrast, our method uses a fully probabilistic framework that enables uncertainty quantification in the latent representations. Moreover, all operations in our method are differentiable, enabling seamless computation and backpropagation of gradients, allowing for an end-to-end training of the model (in contrast to the two-stage training of the VQ-VAE). This also lets us use the reparameterization trick introduced in the DVAE++ [6], which we found to be more stable than continuous relaxations like the Gumbel-Softmax, used in VQ-VAEs with stochastic quantization [8].

---

> > ### Author Response · Authors · 2024-11-15
> > **Response continuation**
> >
> > **Ablation study on the number of parameters**
> >
> > The dimensionality of the latent space is determined by the number of information bits, corresponding to the dimensionality of the binary latent variable $\boldsymbol{m}$, as this defines the set of possible latent vectors for representing the information. In the coded approach, redundancy is introduced to safeguard the information in $\boldsymbol{m}$, generating a codeword $\boldsymbol{c}$. Note that $\boldsymbol{c}$ carries precisely the same information as $\boldsymbol{m}$ since the encoding process adds redundancy deterministically without increasing flexibility (an injective mapping). Thus, a fair comparison between coded and uncoded models must maintain the same number of information bits (i.e., equal dimensionality of $\\boldsymbol{m}$). For example, a 5/50 coded model and a 50-bit uncoded model differ in that the latter has $2^{50}$ possible codewords to represent information, while the former has only $2^5 = 32$. We included $\\boldsymbol{c}$ in the graphical model to explicitly represent the ECC, although this does not add extra latent variables or increase the latent space dimensionality.
> >
> > A consistent architecture was employed across all experiments. However, the encoder’s output and decoder’s input layers were adjusted to accommodate the dimension of encoded vectors, resulting in an increase in trainable parameters for the coded models compared to the uncoded ones. Since models with more parameters may achieve better performance, we conducted an ablation study on the model’s trainable parameters to confirm that the performance gains from the coded models were not due to this factor (see Appendix J.2). By adjusting the encoder’s last hidden dimension and the decoder’s first hidden dimension, we ensured that both coded and uncoded configurations had approximately the same number of trainable parameters, demonstrating that the performance improvement is independent of this slight increase in parameter count.
> >
> > We additionally trained uncoded models on FMNIST using 50, 100, 200, and 300 bits to assess their performance. The table below presents metrics obtained for these various uncoded configurations, alongside coded models with an equivalent number of trainable parameters (matching encoder output and decoder input dimensions). In each case, we specify the number of possible codewords available to represent the information (# latent vectors). We observe that coded models with 10 information bits achieve performance that is comparable to, or even surpasses, that of uncoded models (as shown by the PSNR and Semantic Accuracy metrics), despite having latent dimensionalities that are 10, 20, or 30 times larger.
> >
> > | Model | BER | WER | PSNR test | Acc. test  | # latent vectors |
> > |----------|--------|--------|----------------|--------------|----------------------|
> > |uncoded 50    | 0.320 | 1.000 | 18.056 | 0.818 | $2^{50}$ |
> > |coded 5/50     | 0.011 | 0.046 | 16.241 | 0.647 | $2^5$ |
> > |uncoded 100  | 0.411 | 1.000 | 17.967 | 0.813 | $2^{100}$ |
> > |coded 5/100   | 0.010 | 0.049 | 16.702 | 0.700 | $2^5$ |
> > |coded 10/100 | 0.040 | 0.321 | 17.694 | 0.790 | $2^{10}$ |
> > |uncoded 200  | 0.457 | 1.000 | 18.015 | 0.808 | $2^{200}$ |
> > |coded 10/200 | 0.044 | 0.341 | 18.009 | 0.814 | $2^{10}$ |
> > |uncoded 300  | 0.472 | 1.000 | 17.861 | 0.805 | $2^{300}$ |
> > |coded 10/300 | 0.045 | 0.349 | 18.111 | 0.817 | $2^{10}$ |
> >
> > We also evaluated the generation performance of the different configurations. The figure available at https://figshare.com/s/ef88903bd693306930c3 shows examples of randomly, uncurated FMNIST images generated with the different configurations. In uncoded models with high-dimensional latent spaces, we observed that not all words in the codebook are specialized during model training, resulting in generation artifacts (i.e. noisy images where different object classes overlap). In contrast, coded models do not exhibit such artifacts, demonstrating more effective use of the latent space. This further supports that the superior performance of coded models is not attributed to the difference in trainable parameters but rather to more efficient utilization of the latent space enabled by improved posterior inference.

---

> > > ### Author Response · Authors · 2024-11-15
> > > **Response continuation**
> > >
> > > **Variational gap**
> > >
> > > In the main text, we have provided marginal log-likelihood estimates for different model configurations via importance sampling (further details are provided in Appendix K), with coded models obtaining better values. The variational gap is calculated as the difference between the marginal log-likelihood and the ELBO. In the table below, we provide these values for models trained on FMNIST, showing that coded models effectively reduce the gap in both training and test partitions.
> > >
> > > | Model | LL train | ELBO train | Gap train| LL test | ELBO test | Gap test |
> > > |----------|------------|----------------|-------------|-----------|---------------|-------------|
> > > | uncoded 5 | -266.157 | -274.525 | 8.368 | -267.703 | -275.416 | 7.713 |
> > > | coded 5/50 | -239.379 | -244.286 | 4.907| -241.882 | -247.024 | 5.142 |
> > > | uncoded 8 | -247.964 | -256.792 | 8.828 | -249.880 | -259.049 | 9.169 |
> > > | coded 8/80 | -227.550 | -230.392| 2.842 | -232.992 | -236.924 | 3.932 |
> > > | uncoded 10 | -242.842 | -252.555 | 9.713 | -244.997 | -254.714 | 9.717 |
> > > | coded 10/100 | -222.011 | -223.970 | 1.959 | -230.772 | -234.206 | 3.434|
> > >
> > > **Can the proposed method be extended to VAE with continuous latent space?**
> > >
> > > In the case of models with continuous latent spaces, such as the vanilla VAE, one could quantize the latent distribution to obtain a discrete representation and then apply our proposed approach. This is an interesting direction for future research.
> > >
> > > **Minor**
> > >
> > > We will enhance the resolution of Figure 1(b). This figure illustrates the evolution of the reconstruction and KL divergence terms of the ELBO during training. Therefore, it does not evaluate any reconstruction metric but shows the evolution of $E_{q_{\boldsymbol{\eta}}(\boldsymbol{m},\boldsymbol{z}|\boldsymbol{x})}\log p_{\boldsymbol{\theta}}(\boldsymbol{x}|\boldsymbol{z})$.
> > >
> > > **Conclusion**
> > >
> > > We respectfully believe that the concerns raised by the reviewer have been addressed and would be glad to provide further clarification if needed. As the reviewer has noted, our work introduces novel contributions to the field of deep generative models. We feel that a marginally below acceptance score may not fully capture the potential impact our work could have in advancing this area.
> > >
> > > **References**
> > >
> > > [1] J. M. Tomczak, Deep Generative Modeling. Springer Cham, 2022.
> > >
> > > [2] Van Den Oord, A., & Vinyals, O. (2017). Neural Discrete Representation Learning. Advances in Neural Information Processing Systems, 30.
> > >
> > > [3] Tomczak, J., & Welling, M. (2018, March). VAE with a VampPrior. In International Conference on Artificial Intelligence and Statistics (pp. 1214-1223). PMLR.
> > >
> > > [4] Chen, X., Kingma, D. P., Salimans, T., Duan, Y., Dhariwal, P., Schulman, J., ... & Abbeel, P. (2017). Variational Lossy Autoencoder. In International Conference on Learning Representations.
> > >
> > > [5] Rolfe, J. T. (2017). Discrete Variational Autoencoders. In International Conference on Learning Representations.
> > >
> > > [6] Vahdat, A., Macready, W., Bian, Z., Khoshaman, A., & Andriyash, E. (2018). Dvae++: Discrete variational autoencoders with overlapping transformations. In International Conference on Machine Learning (pp. 5035-5044). PMLR.
> > >
> > > [7] Vahdat, A., Andriyash, E., & Macready, W. (2018). DVAE#: Discrete Variational Autoencoders with Relaxed Boltzmann Priors. Advances in Neural Information Processing Systems, 31.
> > >
> > > [8] Williams, W., Ringer, S., Ash, T., MacLeod, D., Dougherty, J., & Hughes, J. (2020). Hierarchical Quantized Autoencoders. Advances in Neural Information Processing Systems, 33, 4524-4535.

---

> > > > ### Comment · Reviewer_82yp · 2024-11-24
> > > >
> > > > I would like thank the authors for their detailed response. A couple of my concerns are addressed. However, for the first point regarding the baselines, the authors mention that:
> > > >
> > > > **"Since we do not tailor our solution to any specific prior distribution, this approach is fully compatible with state-of-the-art methods that utilize complex priors, such as those previously mentioned. However, we chose to focus on highlighting the significant performance improvements achievable through the introduction of ECCs alone, using a simple, fixed independent prior"**
> > > >
> > > > I believe conducting experiments with complex priors would have demonstrated the versatility of the proposed approach.
> > > >
> > > > I still acknowledge the contribution and the results presented, and I have increased the rating.

---

> > > > > ### Author Response · Authors · 2024-11-24
> > > > >
> > > > > Thank you for your response and for acknowledging our contribution by increasing your rating. We truly appreciate your constructive feedback.
> > > > >
> > > > > > I believe conducting experiments with complex priors would have demonstrated the versatility of the proposed approach.
> > > > >
> > > > > As noted earlier, we chose to use a simple, fixed independent prior to show the performance gains achieved by incorporating ECCs. Moreover, independent priors help improve the interpretability of the latent space, which adds value to showing performance improvements in this scenario. However, it is worth noting that the hierarchical case we presented can also be seen as a coded VAE with a more complex prior distribution, as the information bits used to construct $\boldsymbol{c}_1$ and $\boldsymbol{c}_2$ are no longer independent.

---

### Official Review · Reviewer_8NtD · 2024-10-31

**Soundness:** 3
**Presentation:** 3
**Contribution:** 2
**Rating:** 6
**Confidence:** 4

**Summary:**

In this paper, the authors propose to improve variational inference in discrete variational autoencoders by utilizing error correcting codes (ECCs) in the latent space. Despite significant advances in VAEs, the ability to learn low-dimensional discrete latent-variable models remains a challenging task. To that end, the authors employ ECCs as borrowed from digital communications in order to add redundance in the posterior samples. During training, the VAE is able to capture meaning in the latent variables, resulting in significantly improved generation quality and more accurate estimates. The authors emphasize that this technique is general and can be used alongside existing VAE techniques, such as tighter importance weighted autoencoders and Hamiltonian Monte Carlo.

**Strengths:**

The authors present a theoretically sound and practically useful enhancement to discrete VAEs, noting its practical utility in digital communication. The paper is well-presented; it is easy to follow the contribution from the problems and weaknesses of contemporary approaches to the proposed solution. The experiments validate the theoretical claims, including the claim that the proposed method can be used alongside existing enhancements like IWAEs.

**Weaknesses:**

Although the particular enhancement presented is novel, the motivation behind has been thoroughly studied within the context of VAEs in previous papers, specifically in relation to lossy/lossless compression and communication [1, 2]. The paper mentions Vector Quantized VAEs as an existing approach only briefly, failing to discuss the significance of it in relation to the aforementioned overlapping subjects. It would be helpful to discuss the general relationship between compression and variational inference and how certain VAE implementations exploit this close relationship to their advantage for certain applications, such as lossy compression [2, 3] and lossless compression [4].

References

[1] Aaron van den Oord, Oriol Vinyals, & Koray Kavukcuoglu. (2018). Neural Discrete Representation Learning.

[2] Will Williams, Sam Ringer, Tom Ash, John Hughes, David MacLeod, & Jamie Dougherty. (2020). Hierarchical Quantized Autoencoders.

[3] Xi Chen, Diederik P. Kingma, Tim Salimans, Yan Duan, Prafulla Dhariwal, John Schulman, Ilya Sutskever, & Pieter Abbeel. (2017). Variational Lossy Autoencoder.

[4] James Townsend, Tom Bird, & David Barber. (2019). Practical Lossless Compression with Latent Variables using Bits Back Coding.

**Questions:**

My suggestion to the authors is to read the papers referenced in 'Weaknesses' and write a small section in the introduction describing the relationship between variational inference and Shannon's rate-distortion theory of lossy compression. This will substantially reinforce the validity of the proposed approach, as there is theoretical justification for utilizing a tool originally used in digital communications. Alternatively, the overlap between digital communications and variational inference is made more explicit, so that the unfamiliar reader can make sense of why such a tool is useful.

---

> ### Author Response · Authors · 2024-11-15
> **Response**
>
> Effective learning of low-dimensional discrete latent representations is a technically challenging problem. In this work, we propose a novel method to improve inference in discrete VAEs within a fully probabilistic framework, introducing a new perspective on the inference problem. We understand the connection that the reviewer suggests between our work and papers in the literature that analyze VAEs using rate-distortion (RD) theory. However, our approach stems from a different perspective, that is indeed compatible with all those works.
>
> While VAEs are often viewed as lossy compression models (e.g. VAEs as source coding methods),  our contribution is best understood from a generative perspective. We conceptualize the process of inference via $q(\boldsymbol{m}|\boldsymbol{x})$ as decoding the latent variable $\boldsymbol{m}$ given the observed data $\boldsymbol{x}$. We sample a vector $\boldsymbol{m}$, generate an image $\boldsymbol{x}$, and seek to minimize the error rate in recovering $\boldsymbol{m}$ from $\boldsymbol{x}$. Achieving this requires the variational approximation $q(\boldsymbol{m}|\boldsymbol{x})$ to closely align with the true posterior. Estimating $\boldsymbol{m}$ from $\boldsymbol{x}$ thus involves approximating the true posterior $p(\boldsymbol{m}|\boldsymbol{x})$ with $q(\boldsymbol{m}|\boldsymbol{x})$. We show that introducing redundancy in generation reduces error rates in estimating $\boldsymbol{m}$ and, accordingly, constructs better approximations to the latent posterior distribution. This is reflected in higher log-likelihoods, a smaller variational gap (see response to Reviewer 82yp), and enhanced generation performance.
>
> VQ-VAEs [1] are notable for effectively learning discrete representations of data, and we believe it is useful to highlight key differences between VQ-VAEs and our approach. A primary distinction lies in the latent space’s structure and dimensionality. In image modeling, VQ-VAE typically employs a latent matrix where indices correspond to codewords in a codebook, allowing each codeword to capture specific patches of the original image, which enhances reconstruction and generation (the latter after training an autoregressive prior on the latent representations). However, this matrix representation complicates interpretability, as each data point is represented by a grid of embeddings. In our approach, the latent space encodes the whole image, instead of specific patches. Another important difference is that VQ-VAE is based on a non-probabilistic encoder, where the output is mapped to the nearest latent code based on a distance metric. This deterministic mapping limits the model's ability to quantify uncertainty in the latent space. In contrast, our method uses a fully probabilistic framework that enables uncertainty quantification in the latent representations. Moreover, all operations in our method are differentiable, enabling seamless computation and backpropagation of gradients, allowing for an end-to-end training of the model. This also lets us use the reparameterization trick introduced in the DVAE++ [2], which we found to be more stable than continuous relaxations like the Gumbel-Softmax, used in VQ-VAEs with stochastic quantization [3].
>
> In RD theory, the focus is on compression within the latent space, typically analyzed from an encoder/decoder and reconstruction (distortion) perspective. RD theory sets theoretical limits on achievable compression rates and describes how practical models may diverge from these limits. Using RD practical compression methods, [4] demonstrates how asymmetric numeral systems (ANS) can be integrated with a VAE to improve its compression rate by jointly encoding sequences of data points, bringing performance closer to RD theoretical limits. Similarly, [5] shows that a complex prior distribution in a VAE using an autoregressive invertible flow narrows the gap between the approximate posterior and the true posterior, thereby enhancing the overall performance of the VAE. We remark that even using an independent prior, the hierarchical code structure outlined in Section 6 naturally decouples information across different latent spaces at various conceptual levels. Specifically, the most relevant information (class label) is encoded by the most protected latent space, while the other space captures fine-grained features. This effect is not straightforward to enforce by direct design of a more complex prior.

---

> > ### Author Response · Authors · 2024-11-15
> > **Response continuation**
> >
> > While both papers are relevant to our work (we appreciate the suggestion to include them),  they do not directly align with our approach. Instead, our method can complement these works by introducing redundancy in the generation pathway to improve variational inference, constructing more accurate latent posterior approximations. Notably, even with a complex prior distribution, error-correcting codes could still be used to improve variational inference and overall model performance. In this paper, we demonstrate that even with a simple, fixed independent prior, our approach produces more robust models, evidenced by improved log-likelihood, generation quality, and reconstruction metrics. Furthermore, in Section 6, we show that combining a hierarchical ECC with the same independent prior further improves the results.
> >
> > We stress again that our method could be combined with the techniques in [4], and [5] to refine the posterior or explore compression rates further. Ultimately, our approach could also be paired with RD analysis for VAEs to examine the extent to which prior improvement (as explored in [5], for example) would be necessary when incorporating the enhanced inference enabled by our method. This raises an intriguing question: which is more effective, optimizing the prior design or achieving a more robust inference? However, this issue lies beyond the scope of our current paper.
> >
> > **Conclusion**
> >
> > In light of the reviewer’s comments, we acknowledge that the suggested discussion would indeed enhance the clarity and context of our contribution. We are therefore committed to updating the paper accordingly. However, we respectfully disagree with the rejection score, as we feel it does not adequately reflect the novelty and significance of our proposed method, underscoring its impact and value to the field. As noted by the reviewer, our approach has been rigorously validated across multiple datasets using a comprehensive range of metrics and perspectives.
> >
> > **References**
> >
> > [1] Van Den Oord, A., & Vinyals, O. (2017). Neural Discrete Representation Learning. Advances in Neural Information Processing Systems, 30.
> >
> > [2] Vahdat, A., Macready, W., Bian, Z., Khoshaman, A., & Andriyash, E. (2018). Dvae++: Discrete Variational Autoencoders with Overlapping Transformations. In International Conference on Machine Learning (pp. 5035-5044). PMLR.
> >
> > [3] Williams, W., Ringer, S., Ash, T., MacLeod, D., Dougherty, J., & Hughes, J. (2020). Hierarchical Quantized Autoencoders. Advances in Neural Information Processing Systems, 33, 4524-4535.
> >
> > [4] Townsend, J., Bird, T., & Barber, D. (2019) Practical lossless compression with latent variables using bits back coding. In International Conference on Learning Representations.
> >
> > [5] Chen, X., Kingma, D. P., Salimans, T., Duan, Y., Dhariwal, P., Schulman, J., ... & Abbeel, P. (2017). Variational Lossy Autoencoder. In International Conference on Learning Representations.

---

> > > ### Comment · Reviewer_8NtD · 2024-12-02
> > > **Thanks for the clarification**
> > >
> > > I have updated my score accordingly.

---

### Official Review · Reviewer_NFQm · 2024-11-02

**Soundness:** 2
**Presentation:** 2
**Contribution:** 2
**Rating:** 5
**Confidence:** 3

**Summary:**

The paper introduces the method of Error-Correcting Codes (ECC) into the problem of DVAE. The goal is to safeguard the latent variables to add redundancy to the latent representations.  In this way, the proposed method utilizes the redundancy introduced by the ECC to constrain the solution space of $q(c|x)$, given that each bit from $m$ is repeated L times (through the known G) to create c. Empirical result demonstrated the reconstruction signal to noise ratio of the VAE with ECC is improved than that without ECC.

**Strengths:**

Novelty: The paper is novel in it borrows the method of Error-Correcting Codes (ECC) and introduces it into the problem of DVAE. The motivation is to safeguard the latent variables by adding redundancy to the latent representations through the error correction method in information theory.  In this way, the proposed method claims that it is able to utilizes the redundancy introduced by the ECC to constrain the solution space of $q(c|x)$, therefore leading to a better reconstruction signal to noise ratio of the DVAE than that without ECC. To me, this seems to be the first paper that introduces this method.

Motivation and Significance: The paper aims to improve the reconstruction of the DVAE by introducing the ECC method from information theory. This is a good example of how the techniques in information theory and communication techniques can help improve generative models. The motivation of the work is good and interesting.

**Weaknesses:**

Clarity and rigour: I am a lost in section 4, where how the $m$ is sampled and encoded into the required $c$ is introduced. It seems to me that c is now a linear transformation of $m$ through the known $G$. However, how c is sampled and backpropogated through the DVAE is unclear to me. This section also says: `When comparing uncoded vs. coded DVAEs, the structure of the decoder NN is equal in both cases except for the first Multilayer Perceptron (MLP) layer that attacks the input z'. This is very confusing to me. What does the ``attack'' mean here? Where did the attack come from? In the earlier section of the paper, I did not assume there is any noise injection or attacks existing in the generative model. Also, as the c is linear transformation of the original m. it is unclear to me why such redundancy can improve the signal to noise ratio of the reconstruction. For communication channels, yes, because there is loss during the signal transmission in the channels, but I do not understand why the c=mG can help better constrain the sampling space of $m$ and why this is relevant in achieving a better generative model. I guess better instantiation of the the method with further theoretical support and an algorithmic flow can better support the idea of the paper.

**Questions:**

1. Please clarify what is the gain of introducing ECC into the encoding of $m$ is beneficial intuitively. Where does the noise come from (analogously to communication channels), and why ECC helps reducing the reconstruction error through this encoding.

2. If possible, please use theoretical support to prove that the encoded DVAE can achieve a better SNR (signal to noise ratio) through the proposed encoding of latent variable.

3. I appreciate if any algorithmic flow is further provided in the appendix or somewhere to better clarify the sampling process of c and how it is backpropogated.

---

> ### Author Response · Authors · 2024-11-15
> **Response**
>
> **Clarification on Section 4**
>
> The coded DVAE extends the DVAE by introducing an ECC to encode the binary variable $\boldsymbol{m}$ (the information vector). The resulting codeword $\boldsymbol{c}$ is indeed a linear transformation of $\boldsymbol{m}$ as you note, computed by multiplying $\boldsymbol{m}$ with a known generator matrix $G$. This approach augments the dimensionality of the binary latent variable in a controlled and deterministic manner. During generation, we sample $\boldsymbol{m}$ and transform it into $\boldsymbol{c}$ using $G$ (refer to Fig.39 in Appendix N for the scheme). This process, known as hard encoding, is the method we use to generate new samples once the model is trained, and all presented generation results were obtained with this approach. The key aspect of our proposal is that the variational posterior approximation exploits this knowledge (i.e. certain elements of the vector must be identical) to improve inference and enhance the training of the whole model. See the “Intuition behind our approach” section below.
>
> However, during training, soft encoding is required for reparameterization. To compute the reconstruction term of the ELBO, given a data point $\boldsymbol{x}$, a marginal posterior probability $q(m_i)$ is obtained for each latent bit $m_i$, with $i=1,..., M$. To evaluate the datum log-likelihood under this posterior, soft encoding produces a marginal probability $q(c_j)$ for each bit in the codeword, $ j=1,..., M/R$, incorporating the structure of the ECC. These coded marginals are combined with the reparameterization trick in Eq.(5). In this way, marginal posterior probabilities $q(c_j)$ are directly used to sample $\boldsymbol{z}$, just as in the uncoded scenario. This method allows us to avoid the instability of alternative approaches such as the Gumbel-Softmax relaxation or the REINFORCE gradient estimator. Since the operations in the reparameterization trick and computation of the marginal posteriors are differentiable, the model can be trained via SGD.
>
> When we say, "When comparing uncoded vs. coded DVAEs, the structure of the decoder NN is equal in both cases except for the first Multilayer Perceptron (MLP) layer that attacks the input $\boldsymbol{z}$," we mean that the decoder architecture is identical in both cases, except for the input layer, which processes the input $\boldsymbol{z}$ and therefore needs to match its dimensionality. The misunderstanding stems from wording; we’ll use clearer terminology to prevent confusion.

---

> > ### Author Response · Authors · 2024-11-15
> > **Response continuation**
> >
> > **The intuition behind our approach (answer to Q1 and Q2)**
> >
> > In fields where accurate data transmission or storage is essential, introducing error-correcting techniques is a well-established approach to reduce the error rate when estimating a discrete source $\boldsymbol{m}$ that is to be sent through a noisy channel (in our context, the VAE decoder). The reason that this works, intuitively, is that the resulting codewords now form an $m$-dimensional subspace of the larger $n$-dimensional space in which any two codewords $\boldsymbol{c}$ are now separated by at least some minimum distance (i.e., it is a sphere packing). This structure, where $2^{n-m}$ of $n$-tuples are not valid codewords, allows a decoder to tolerate errors and attempt to identify the most likely codeword. Given an observation $\boldsymbol{x}$, decoding algorithms can efficiently (attempt to) find the codeword $\boldsymbol{c}$ to maximize the probability of $p(\boldsymbol{c}|\boldsymbol{x})$ over all $2^m$ codewords and thereby maximize the probability $p(\boldsymbol{m}|\\boldsymbol{x})$ of the estimate of $\boldsymbol{m}$, as $\boldsymbol{c}$ and $\boldsymbol{m}$ are deterministically related. The more redundancy we add, the larger the minimum distance between codewords becomes and the less likely it becomes to pick the incorrect estimate of $\boldsymbol{m}$.
> >
> > The same mechanism helps to improve the estimate of the posterior $q(\boldsymbol{m}|\boldsymbol{x})$ in our proposal. Estimating $\boldsymbol{m}$ from $\boldsymbol{x}$ involves approximating the true posterior $p(\boldsymbol{m}|\boldsymbol{x})$ distribution with $q(\boldsymbol{m}|\boldsymbol{x})$, our variational proposal. The closer these distributions are, the fewer errors we make when comparing $\boldsymbol{m}$ with samples from $q(\boldsymbol{m}|\boldsymbol{x})$. The gap between $q(\boldsymbol{m}|\boldsymbol{x})$ and $p(\boldsymbol{m}|\boldsymbol{x})$ is precisely the variational bound and can be considered as the “noise”. The structured redundancy introduced by the ECC is exploited to improve the posterior $q(\boldsymbol{m}|\boldsymbol{x})$ and therefore reduce the variational gap to the true posterior. Our results confirm that this approach reduces error rates (see table included in Figure 4, for example), improves log-likelihoods (refer to the table in Figure 4), narrows the variational gap (see the response to Reviewer 82yp), and enhances overall model performance.
> >
> > Specifically, for repetition codes, error detection and correction can be achieved in a simple way by majority voting. Since each information bit is repeated L times, the original value can be recovered by choosing the most common value among the repetitions. This approach allows one to recover the original information vector, even if some bits in the transmitted encoded word are corrupted. In our method, we apply a soft majority voting strategy to compute the posterior probability of each information bit by evaluating the all-ones and all-zeros probabilities for the L repeated bits. This soft decoding strategy exploits the known correlations introduced by the linear code to correct potential errors introduced by the encoder when mapping an image to latent space.
> >
> > **Algorithm (answer to Q3)**
> >
> > In Appendix B of the original submission, we provided the training pseudo-code for the coded model using repetition codes. In step 4, we apply a soft decoding method to obtain $q(m_i|x)$ given $q^{u}(c_j|\boldsymbol{x})$ for all coded bits. This step is where potential errors from the encoder NN $g_{\boldsymbol{\eta}}(\boldsymbol{x})$, which estimates $q^{u}(c_j|\boldsymbol{x})$ from $\boldsymbol{x}$, are corrected, thereby improving the approximation of the posterior $q(\boldsymbol{m}|\boldsymbol{x})$. Steps 5 and 6 describe the sampling procedure: after obtaining the posterior $q(\boldsymbol{m}|\boldsymbol{x})$ via soft decoding in step 4, we replicate the marginal posterior probabilities $q(m_i|\boldsymbol{x})$ across copies of the same information bit to derive the encoded marginals $q(c_j|\mathbf{x})$ (soft encoding, step 5). These marginals are then used to sample $\boldsymbol{z}$ through the reparameterization trick in Eq. (5), similar to the uncoded case (step 6). Since the discrete variable $\boldsymbol{c}$ is marginalized and $\boldsymbol{z}$ is sampled using reparameterization, it is possible to compute gradients and optimize the ELBO with respect to the model parameters.
> >
> > **Conclusion**
> >
> > Please let us know if our explanations provide sufficient clarity; we would be glad to elaborate further. As you have acknowledged, this work represents the first contribution to introduce error-correcting techniques as viable, low-complexity methods for enhancing variational inference. In light of this, if our responses have adequately addressed your concerns, we respectfully believe that a ‘marginally below acceptance threshold’ score would not fully reflect the merit of this contribution.

---

> > > ### Comment · Reviewer_NFQm · 2024-11-25
> > > **Thanks for your feedback**
> > >
> > > Thanks for the detailed response to my questions. However, after reading other reviewers comments, I agree to many of their points. Similar to Reviewer tR1Q's concern "a closer match between the true posterior and the variational posterior does not necessarily guarantee a reduction in generalization error [Reviewer tR1Q]", I am also looking for some more theoretical support behind this error correction technique, in that how this helps generalization. Although I fully understand in terms of communication techniques, this method is guaranteed to improve the signal to noise ratio, but still, its connection to the generalization remains unclear. I am afraid I need to keep my score by taking this into account.

---

> > > > ### Author Response · Authors · 2024-11-28
> > > >
> > > > Thank you for your response! We kindly direct you to the general comment we have posted, as well as our response to Reviewer tR1Q, since your concerns are aligned.
> > > >
> > > > With the discussion period still ongoing, we would sincerely appreciate it if you could suggest specific actions we could take to address your concerns and encourage you to consider raising your score after reviewing our responses.

---

### Author Response · Authors · 2024-11-15

We would like to express our gratitude to the reviewers for their valuable feedback, and for recognizing the novelty of our contribution. We sincerely appreciate the time they dedicated to thoroughly reviewing our paper. We believe the received feedback led to valuable discussions and motivated us to enhance our work by addressing all the specific concerns raised. We have provided detailed responses to each reviewer’s comments to ensure all points are thoroughly addressed.

---

### Author Response · Authors · 2024-11-19

Dear Reviewers,

We would like to kindly remind you that we have addressed all the specific concerns raised by the Reviewers by Friday. Please let us know if our explanations are clear enough; we are happy to provide further details if needed, as we have time to do so. We have made an effort to respond promptly to encourage discussion given your positive feedback on the novelty of our work and its potential impact.

Once again, we sincerely thank you for the feedback, which has led to valuable discussions and inspired us to improve our work by addressing all the concerns raised.

---

### Author Response · Authors · 2024-11-20

Dear Reviewers,

We would like to inform you that we have uploaded a revised version of the paper, addressing minor issues such as the resolution of Figure 1(b) and incorporating a discussion on prior work regarding VAEs as source coding methods in Appendix L. Once again, we sincerely thank you for your thoughtful reviews and the positive feedback on our work. Please let us know if our explanations are clear and if we have addressed all your concerns. We would be happy to provide further clarifications if needed.

---

### Author Response · Authors · 2024-11-28
**General Response**

At this stage, we believe it may be helpful to provide a general response addressing the theoretical foundations of the paper and the merit of the proposed approach, as this topic has been raised in multiple reviewer discussions. As previously mentioned, several factors influence how closely the variational approximation matches the true posterior, which determines the variational gap. These include the complexity of the true posterior, the choice of variational family, the architecture used to model the distributions, and the sampling method for estimating the ELBO. While the standard VAE setup often performs well in simpler scenarios, it can face challenges in representing more complex data structures. Consequently, various approaches have been proposed over the years to **reduce the gap between the variational and true posterior distributions, which is a common goal of the VAE framework and VAE literature**.

A common strategy to enhance inference in VAEs is to design more flexible priors, which helps address challenges like posterior collapse and the hole problem. One approach is using mixture distributions to introduce multimodality, as seen in VampPrior [1], where the prior is an approximation of the aggregated posterior given by a mixture of variational posteriors conditioned on learnable pseudo-inputs (optimized through backpropagation). Another approach is to define a hierarchical latent space, composed of multiple stochastic layers of latent variables with conditional dependencies. An example of this is DVAE++ [2], which uses a Boltzmann Machine prior along with a hierarchy of continuous latent variables. Similarly, flexible autoregressive priors can be employed. For instance, VQ-VAE [3], though not affected by over-regularization due to its non-probabilistic encoder, uses an autoregressive model to train a prior that aligns with the learned aggregated posterior, enabling the generation of new samples. More recently, Flow-based [4] and Diffusion-based [5] priors have been proposed following the same intuition.

The choice of the variational family also impacts the flexibility of the model and the tightness of the variational bound. Similar approaches to the ones presented for the prior have been proposed to define more flexible variational approximations, including hierarchical structures [6] and Flow-based posteriors [7].

While inference can be improved by defining more flexible prior and approximate posterior distributions, another approach is to use tighter variational bounds as training objectives. One example is the Importance Weighted Autoencoder (IWAE) bound [8], which represents an importance-weighted estimate of the log-likelihood. Increasing the number of samples used in the estimate makes the bound tighter, and optimizing these tighter bounds is generally (**empirically**) associated with better performance in terms of marginal likelihood on unseen data. **It is worth noting that in our experiments, we trained the uncoded model using the IWAE bound, yet its performance was still significantly below that achieved with our proposed approach based on ECCs.**

While all these outstanding works significantly advanced the state-of-the-art in VAEs, it is important to note that they do not provide theoretical proof of reducing the gap to capacity. Instead, they introduce innovative methods to effectively address existing challenges in training VAEs and demonstrate the effectiveness of their approaches through **comprehensive empirical evaluations**.

In this work, we introduce a novel perspective on the inference problem, which, to the best of our knowledge, has not been previously explored in the literature. We propose a new method that significantly enhances inference, and that is **supported by both theoretical foundations** (as discussed in our response to Reviewer tR1Q) **and empirical evidence demonstrating its effectiveness, similar to many of the impactful works previously mentioned**. While we are not claiming that our method outperforms the approaches discussed earlier, **we present it as a novel alternative for achieving the same goal, introducing a novel perspective on the inference problem that could inspire future research. The key contribution lies in offering a novel approach that is inherently compatible with these methods.** For instance, consider the **hierarchical extension of our method through the integration of polar codes** presented in Section 6.

---

> ### Author Response · Authors · 2024-11-28
> **General Response Continuation**
>
> **We have conducted a comprehensive empirical evaluation of our method**, demonstrating that the tighter bound observed during training is not a result of overfitting. While we do not claim that our method directly improves generalizability or provides implicit regularization, we assert that it enhances inference narrowing the gap between the true and approximate posterior, which in turn leads to an overall improvement in performance. This is consistent with the general trends observed in the VAE literature (we would appreciate it if the Reviewers could provide any references that suggest otherwise). Additionally, we observe improvements in both the training and test sets (with training metrics provided in the Appendix), which indicate an overall enhancement in model fitting. We have evaluated not only the model's reconstruction accuracy (using metrics such as PSNR, Semantic Accuracy, and Confident Semantic Accuracy) but also its generation performance (through visual examples and FID) and the tightness of the variational posterior (measured by the variational gap, log-likelihood, BER, and WER). **All results are consistent across different datasets and model configurations, providing empirical evidence that clearly supports our claims (as the Reviewers have acknowledged).**
>
> **We believe this contribution is valuable and worth sharing with the community. This seminal work (supported by theoretical foundations and empirical evidence) opens the door to further theoretical analysis, which is beyond the scope of this paper, and offers a novel perspective on the inference problem that could inspire future research.** In light of this, we respectfully believe that a score of ‘marginally below acceptance threshold’ would not adequately reflect the merit of this contribution, as it would likely result in the rejection of our submission.
>
> Since there is still time until the discussion period ends, we would appreciate it if the reviewers could specify what actions we could take to encourage them to consider raising their scores.
>
> **References**
>
> [1] Tomczak, J., & Welling, M. (2018, March). VAE with a VampPrior. In International Conference on Artificial Intelligence and Statistics (pp. 1214-1223). PMLR.
>
> [2] Vahdat, A., Macready, W., Bian, Z., Khoshaman, A., & Andriyash, E. (2018, July). Dvae++: Discrete Variational Autoencoders with Overlapping Transformations. In International Conference on Machine Learning (pp. 5035-5044). PMLR.
>
> [3] Van Den Oord, A., & Vinyals, O. (2017). Neural Discrete Representation Learning. Advances in Neural Information Processing Systems, 30.
>
> [4] Gatopoulos, I., & Tomczak, J. M. (2021). Self-Supervised Variational Auto-Encoders. Entropy, 23(6), 747.
>
> [5] Wehenkel, A., & Louppe, G. (2021). Diffusion Priors In Variational Autoencoders. In ICML Workshop on Invertible Neural Networks, Normalizing Flows, and Explicit Likelihood Models.
>
> [6] Sønderby, C. K., Raiko, T., Maaløe, L., Sønderby, S. K., & Winther, O. (2016). Ladder Variational autoencoders. Advances in Neural Information Processing Systems, 29.
>
> [7] Rezende, D., & Mohamed, S. (2015, June). Variational Inference with Normalizing Flows. In International Conference on Machine Learning (pp. 1530-1538). PMLR.
>
> [8] Yuri Burda, Roger Grosse, and Ruslan Salakhutdinov (2016). Importance Weighted Autoencoders. In 4th International Conference on Learning Representations (ICLR).

---

### Note · Authors · 2025-09-11

**Comment:**

A revised version of this paper has been accepted at the Forty-first Conference on Uncertainty in Artificial Intelligence (UAI 2025). Therefore, we kindly request the withdrawal of this version of the article.

**Withdrawal Confirmation:**

I have read and agree with the venue's withdrawal policy on behalf of myself and my co-authors.

---

### Meta-Review · Area_Chair_jNiP · 2024-12-18

**Metareview:**

This paper proposes a novel variational inference framework using Error-Correcting Codes (ECC) to improve the performance of VAEs with discrete latent variables. Specifically, it models discrete variables with repetition codes based on ECC and employs an appropriate variational posterior distribution, demonstrating significant performance improvements across various tasks.
The main weakness of this study is the unclear scope of its applicability. As pointed out by multiple reviewers, the validity of the proposed method has been evaluated solely through numerical experiments. While performance improvements are observed under a simple prior using independent Bernoulli distributions, it remains uncertain whether the method is effective under other, more realistic model settings. Thus, as several reviewers have noted, it would be desirable to strengthen the numerical evidence supporting the study's contributions and to provide numerical or theoretical insights into the settings where this method is most applicable or less effective.
On the other hand, all reviewers have recognized the novelty of the idea, suggesting that the research direction holds sufficient significance. However, in its current form, the study is not sufficient for publication, and I recommend a rejection at this time.

**Additional Comments On Reviewer Discussion:**

All reviewers raised questions about why the use of ECC leads to performance improvements. The authors addressed these concerns primarily from a qualitative perspective, explaining the improvement in terms of the variational gap. However, these explanations do not necessarily guarantee the experimental performance, prompting Reviewer tR1Q and Reviewer NFQm to request a theoretical explanation of the method, particularly in terms of SNR and generalization ability. The authors did not provide a direct response to these requests, instead offering further qualitative explanations.
Additionally, Reviewer 8NtD and others raised concerns about the relationship with existing studies, particularly VQ-VAE. These concerns were adequately addressed with sufficient explanations provided by the authors.

---

### Decision · Program_Chairs · 2025-01-22

Reject